# DOT1L provides transcriptional memory through PRC1.1 antagonism

Daniel Neville[1,8], Daniel T. Ferguson[1,8], Emily B. Heikamp[2], Zhihao Lai[3], Graham W. Magor[1,4], Charlene Lam[1], Olivia G. Dobbs[1], Vita Levina[3], Kathy Knezevic[5], James J. The[1], Shania Alex[1], Stephen C. Suits[1], Bradon Rumler[1], Michael Uckelmann[3], Laure Talarmain[5], Enid Y. N. Lam [5,6], Andrew C. Perkins [1], Scott A. Armstrong [2], Charles C. Bell [3], Chen Davidovich [7] ✉ & Omer Gilan [1] ✉

DOT1L and Menin are essential cofactors for the oncogenic activity of MLL fusion proteins (MLL-FPs) in leukaemia. However, the mechanisms underpinning the therapeutic effects of their inhibitors remain unclear. Here we identify a critical role for the non-canonical Polycomb repressive complex 1.1 (PRC1.1) in mediating the cellular responses to DOT1L and Menin inhibitors. Menin inhibition induces PRC1.1-dependent deposition of H2AK119ub to silence a subset of MLL-FP targets, whereas DOT1L inhibition results in a genome-wide increase in H2AK119ub. We show that enhanced PRC1.1 activity arises specifically from the progressive loss of DOT1L-mediated H3K79 methylation, independent of MLL-FP displacement or transcriptional repression. This regulatory crosstalk is conserved across cell types and is driven by direct biochemical antagonism between H3K79 methylation and PRC1 activity. Together, our findings establish DOT1L as a component of transcriptional memory co-opted in leukaemia and suggest it serves as the missing link balancing the opposing forces of the MLL–Polycomb axis.

Gene expression is regulated by the interplay of activating and repressive chromatin factors that fine-tune transcription and chromatin state[1–4]. A key example of such antagonism occurs between Mixed Lineage Leukaemia (MLL) and Polycomb complexes, which coordinate developmental gene expression in embryonic and haematopoietic lineages[5]. MLL1/2 COMPASS-like complexes promote transcriptional activation by depositing tri-methylation on histone H3 at lysine 4 (H3K4me3) at developmental gene promoters through Menin-dependent mechanisms that oppose Polycomb-mediated repression[6–8].

Polycomb repressive complexes 1 and 2 (PRC1 and PRC2) repress transcription through distinct mechanisms: PRC1 catalyses mono-ubiquitination of histone H2A at lysine 119 (H2AK119ub), while PRC2 deposits methylation on histone H3 at lysine 27 (H3K27me1/2/3)[9–11]. H2AK119ub is a dynamic mark essential for gene repression[12,13], whereas H3K27me3 maintains heritable silencing[14,15]. Together, MLL and Polycomb regulate chromatin states essential for cell differentiation and, consequently, mutations within these complexes are common drivers of cancer development[5,16].

[1]Australian Centre for Blood Diseases, School of Translational Medicine, Monash University, Melbourne, Victoria, Australia. [2]Department of Pediatric Oncology, Dana-Farber Cancer Institute, Division of Hematology/Oncology, Boston Children's Hospital and Harvard Medical School, Boston, MA, USA. [3]Mater Research Institute, University of Queensland, Woolloongabba, Queensland, Australia. [4]Queensland Institute of Medical Research, Brisbane, Queensland, Australia. [5]Peter MacCallum Cancer Centre and the University of Melbourne, Parkville, Victoria, Australia. [6]Sir Peter MacCallum Department of Oncology, The University of Melbourne, Parkville, Victoria, Australia. [7]Department of Biochemistry and Molecular Biology, Biomedicine Discovery Institute, Monash University, Clayton, Victoria, Australia. [8]These authors contributed equally: Daniel Neville, Daniel T. Ferguson. ✉e-mail: chen.davidovich@monash.edu; omer.gilan@monash.edu

Chromosomal rearrangements involving the MLL1 gene generate oncogenic MLL fusion proteins (MLL-FPs), which retain Menin-mediated chromatin recruitment but lack the C-terminal MLL1 SET domain[7,8,17]. MLL-FPs aberrantly recruit the super elongation complex and the disruptor of telomeric silencing 1 (DOT1L) complex to MLL1 target genes, which are thought to enhance transcriptional elongation at genes associated with leukaemic self-renewal[17–20]. DOT1L is a histone methyltransferase that methylates histone H3 at lysine 79 (H3K79me1/2/3), a histone modification that is enriched at MLL-FP target genes and is linked with transcriptional output[21]. As H3K79 methylation lacks an active demethylase, its turnover is slow, which suggests functions beyond elongation kinetics[22].

MLL-FPs remain undruggable but therapies that indirectly target their function include Menin or DOT1L inhibitors[7,17,23,24]. Menin inhibitors show promise in clinical trials with recent FDA approval in relapse settings[25–27], while DOT1L inhibitors have been less effective[28]. Although DOT1L inhibitors have suffered from poor pharmacokinetic/pharmacodynamic properties, the differences in clinical success between the two drugs probably also reflects their distinct mechanism of action[26,28–30]. Menin inhibitors immediately impact transcription, whereas DOT1L inhibition shows delayed gene downregulation owing to slow H3K79me turnover[31,32]. DOT1L inhibition has also been shown to promote H3K27me3 accumulation at HOXA genes, further suggesting additional roles beyond elongation[33]. Here, we sought to address these gaps in our understanding and provide insight into the distinct functions of DOT1L and Menin in MLL-FP leukaemia and the MLL–Polycomb regulatory axis.

## Results

### Depletion of the PRC1.1 complex components, BCOR and PCGF1, confers resistance to Menin and DOT1L inhibition in MLL-leukaemia cells

To identify the genes required for the efficacy of DOT1L and Menin inhibition, we performed CRISPR survival screens in human and murine leukaemia cells using selective inhibitors of DOT1L (SGC0946;DOT1Li) and Menin (VTP50469;MENi) (Fig. 1a). We initially undertook a genome-wide CRISPR screen in human (MV4;11) cells treated with DOT1Li to broadly uncover the mediators of DOT1Li efficacy, which identified several hits that function together in epigenetic complexes (Fig. 1b and Supplementary Tables 1 and 2). These include BCOR and PCGF1, components of the non-canonical PRC1.1 complex, KMT2D and NCOA6, members of the MLL3/MLL4 COMPASS complex, and EED, EZH2 and SUZ12, components of the PRC2 complex (Fig. 1c and Extended Data Fig. 1a). Given the enrichment of genes with a chromatin-based function, we subsequently performed additional

screens using a bespoke epigenetics-focused library in murine MLLAF9 cells using both DOT1Li and MENi, which confirmed the requirement of PRC1.1 and MLL3/4 complex subunits for the efficacy of both compounds (Fig. 1d, Extended Data Fig. 1b,c and Supplementary Tables 3 and 4).

Validation using independent single guide RNAs (sgRNAs) confirmed that depletion of BCOR, PCGF1, RING1B, UTX or KMT2D conferred a growth advantage in the presence of DOT1Li or MENi in both human and murine leukaemia cells (Fig. 1e and Extended Data Fig. 1d–g). Despite previous reports, BCOR or PCGF1 knockout (KO) cells were still able to proliferate under standard culture conditions as well as in drug treatment, albeit to a lesser extent[34,35] (Fig. 1f and Extended Data Fig. 1h–j). Importantly, reconstituting PRC1 function by the expression of PCGF1 cDNA in PCGF1 KO cells restored sensitivity to both Menin and DOT1L inhibition (Extended Data Fig. 1i). Unlike the PRC1.1 components, KO of the PRC2 subunits, EZH2 and SUZ12, did not enrich in the CRISPR screens performed in the murine MLLAF9 cells or at a later timepoint in the human MV4;11 screen, probably owing to their essential role in cell survival, supported by public datasets[36] (Fig. 1a and Extended Data Fig. 1k–m).

The consistent requirement of PRC1.1 subunits for DOT1L and Menin inhibitor efficacy across all tested leukaemia cell lines suggests that Polycomb repression is critical for silencing MLL-FP target genes upon inhibitor treatment. Importantly, genetic depletion of DOT1L in PCGF1 or BCOR KO cells confirmed that resistance is not due to residual DOT1L activity, which is further supported by the fact that H3K79me2 loss after DOT1Li remained unaffected in PRC1.1 depleted cells (Fig. 1g,h and Extended Data Fig. 1n). PCGF1 KO cells exhibited markedly reduced apoptosis upon prolonged DOT1Li or MENi treatment, and dose–response assays confirmed resistance to both high and extended treatment doses, without cross-resistance to other therapies such as IBET-151 or ABT-199 (Fig. 1i,j and Extended Data Fig. 1o–q). To further probe PRC1.1 subunit interaction, double KO of BCOR and PCGF1 in MOLM13 cells showed greater resistance and increased proliferation, indicating non-redundant roles (Fig. 1k). Collectively, these results suggest that PRC1.1 suppresses cell proliferation and identify a critical role for PRC1.1 subunits in selectively mediating the efficacy of Menin and DOT1L inhibition in MLL-leukaemia.

### PRC1.1 depletion blunts the transcriptional response to Menin and DOT1L inhibition

Having established that PRC1.1 components are required for the efficacy of DOT1L and Menin inhibitors, we next examined whether PRC1.1 is necessary for the transcriptional response to these treatments. We profiled global gene expression changes after DOT1Li and MENi treatment in

---

**Fig. 1 | Depletion of the PRC1.1 complex components, BCOR and PCGF1, confers resistance to Menin and DOT1L inhibition in MLL-leukaemia cells.** **a**, A schematic of the CRISPR survival screens. MV4;11 Cas9 or murine MLLAF9 Cas9 cells were infected with either a genome-wide or epigenetics-focused (1,134 genes) sgRNA CRISPR library. Cells were treated with DMSO, the Menin inhibitor VTP50469 (MENi, 100 nM) or the DOT1L inhibitor SGC0946 (DOT1Li, 5 µM). Samples were collected at days 12 and 24 (MV4;11 cells) and day 14 (murine MLLAF9 cells). **b**, A bubble plot showing the genes required for the efficacy of SGC0946 (day 24) in MV4;11 cells from the whole-genome CRISPR screen. *P* values were calculated using the MAGECK algorithm and adjusted for multiple testing. **c**, A schematic overview of the complexes of interest identified in the CRISPR screen: PRC1.1 and MLL3/4 with the enriched components highlighted in colour. **d**, A Venn diagram of the top 20 most significant hits from the murine MLLAF9 SGC0946 and VTP50469 survival screens performed using an epigenetics-focused CRISPR library. **e**, sgRNA negative selection competition assay in MV4;11 Cas9 cells transduced with non-silencing sgRNA (control) or two independent sgRNAs against BCOR or PCGF1. The percentage of sgRNA positive cells remaining over time. Data represent the mean ± s.d. from *n* = 3 independent experiments. **f**, A proliferation assay in control (ctrl) or two independent PCGF1 (left) or BCOR (right) sgRNAs in MV4;11 Cas9 cells treated

with DMSO or SGC0946 5 µM as indicated. Data represent the mean ± s.d. from *n* = 3 independent replicates. **g**, A schematic of the layered DOT1L sgRNA negative selection competition assay. Two independent DOT1L sgRNAs were layered on top of cells already expressing BFP-control, PCGF1 or BCOR sgRNAs. These double KOs were then mixed with single KOs of each and the percentage of GFP⁺ was measured by flow cytometry. **h**, sgRNA negative competition assay using MOLM13 Cas9 control, PCGF1 or BCOR KO-BFP cells transduced with two independent sgRNA against DOT1L or non-silencing control sgRNA linked with GFP. The percentage of sgRNA positive cells remaining over time are shown. Data represent the mean ± s.d. from *n* = 3 independent experiments. **i**, Bar plots of the percentage of Annexin V positive (apoptotic) control or PCGF1 KO murine MLLAF9 cells treated with either DMSO, SGC0946 (3 µM) or VTP50469 (300 nM) for 4 or 6 days. Data represent mean ± s.d. from *n* = 3 independent experiments. **j**, Proliferation assays with control or PCGF1/BCOR KO MOLM13 cells treated with ABT-199 (100 nM) or I-BET151 (1 µM). Data represent mean ± s.d. from *n* = 3 independent experiments. **k**, A proliferation assay in control or PCGF1/BCOR KO MOLM13 cells treated with DMSO or combined treatment with SGC0946 (5 µM) and VTP50469 (500 nM) as indicated. Data represent *n* = 3 independent experiments.

three leukaemia cell lines with or without PCGF1 or BCOR depletion using 3' RNA-sequencing (RNA-seq) (Fig. 2a and Extended Data Fig. 2a–c). As expected from their shared role in PRC1.1, depletion of PCGF1 and BCOR produced highly correlated transcriptional profiles

(Extended Data Fig. 2d). Given the established role of PRC1.1 in gene repression, we first assessed baseline gene expression before treatment. Although most genes normally downregulated by Menin or DOT1L inhibition were unchanged in PCGF1 or BCOR KO cells, specific

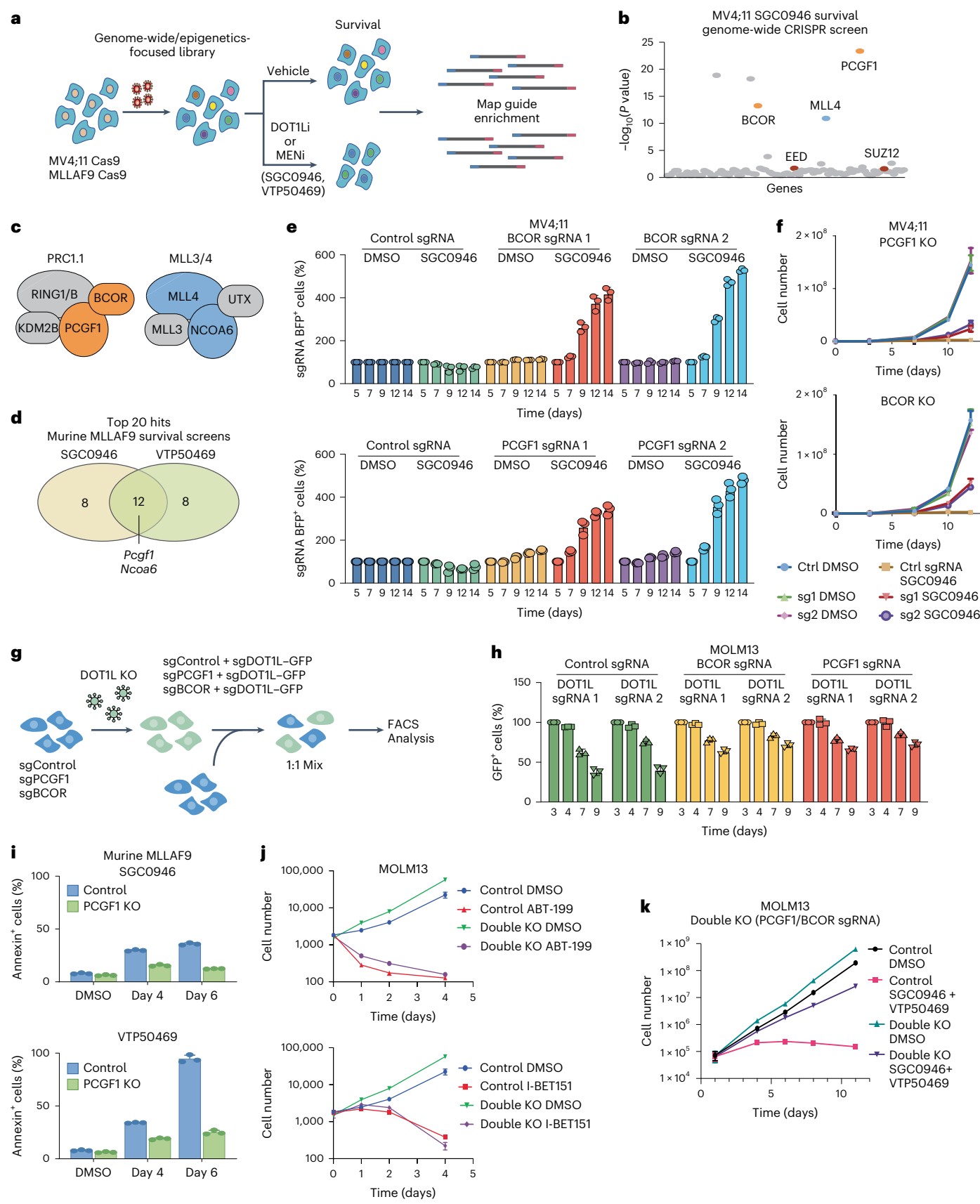

MLL-FP targets (for example, *Meis1*) showed modest de-repression (Extended Data Fig. 2e). Upon inhibitor treatment, however, loss of PCGF1 or BCOR broadly blunted both up- and downregulation of target genes (Fig. 2b,c). Across all three leukaemia cell lines, PRC1.1 depletion produced widespread rescue of genes normally repressed or activated by DOT1Li or MENi treatment (Fig. 2d and Extended Data Fig. 2f–j). While some genes remained similarly downregulated in control and KO cells, key MLL-FP target genes (for example, *Meis1*) were strongly rescued in the absence of PRC1.1 (Fig. 2e and Extended Data Fig. 2h). Overexpression of PCGF1 in KO cells restored *Meis1* repression, confirming specificity (Extended Data Fig. 2i). These results demonstrate that PRC1.1 is selectively required for repression of MLL-FP target genes following disruption of MLL-FP chromatin function.

Although Menin and DOT1L regulate overlapping transcriptional programmes, each inhibitor also affects unique targets[7,31]. In murine MLLAF9 cells, DOT1Li-specific downregulated genes included *Hoxa9/11*, *Tsc22d2* and *Zeb2*, while MENi uniquely affected *Lpxn* and *Foxp1* (Fig. 2f,g). Across all three leukaemia models, only two genes (*Meis1* and *Emb*) were consistently rescued by PRC1.1 loss under both treatments (Fig. 2h).

Since *Meis1* repression is central to Menin inhibitor activity, we analysed its downregulation kinetics by quantitative PCR with reverse transcription (RT–qPCR)[7] (Fig. 2i). While early responses were similar, *Meis1* expression continued to decline progressively in control cells, whereas its expression stabilized in PCGF1 KO cells (Fig. 2i). Collectively, these data implicate PRC1.1 in the robust shutdown of critical MLL-FP targets upon exposure to differentiation therapies.

### PRC1.1 is required for efficient DOT1L and Menin inhibitor-induced differentiation in vitro and in vivo

Menin and DOT1L inhibitors act as differentiation therapies that can restore normal developmental trajectories in leukaemia models and in the clinic[26,28]. Given that PRC1.1 depletion reduced responsiveness to these inhibitors, we investigated whether it also impaired leukaemia cell differentiation. Using CellRadar RNA-seq analysis, which compares gene sets to published lineage-associated expression programmes, we found that genes upregulated by Menin or DOT1L inhibition were enriched for differentiated granulocytic and monocytic states (Fig. 3a). By contrast, PCGF1 depletion upregulated genes associated with a haematopoietic stem and progenitor cell programme, consistent with modest de-repression of MLL-FP target genes and a more immature phenotype (Fig. 3a,b and Extended Data Fig. 3a).

Importantly, PRC1.1 loss attenuated the induction of differentiated gene expression programmes in response to Menin and DOT1L inhibition (Extended Data Fig. 3b). Flow cytometry and morphological analyses supported this, showing reduced cell size and GR-1 expression

and diminished induction of Ly6G, GR-1 and CD11b in PRC1.1-deficient cells (Fig. 3c–e and Extended Data Fig. 3c,d).

To test whether PRC1.1 is required for Menin inhibitor efficacy in vivo, NSG mice were transplanted with human MLLAF9 cells (MOLM13) expressing control or PCGF1/BCOR KO sgRNAs. Mice were treated with control chow or chow formulated with 0.1% SNDX-5613, a clinical formulation of the Menin inhibitor VTP50469, that was well tolerated across all cohorts (Extended Data Fig. 3e). Control mice responded to SNDX-5613 with a ~3-week survival benefit, whereas the PRC1.1-deficient leukaemia showed no significant improvement (Fig. 3f). High levels of human CD45+ cells in bone marrow, spleen and peripheral blood further confirmed treatment resistance (Fig. 3g and Extended Data Fig. 3f). In addition, PRC1.1 loss impaired SNDX-5613-induced differentiation in vivo (Fig. 3h). RNA-seq of CD34+ cells collected at day 18 showed that in vivo PRC1.1-deficient MOLM13 cells exhibited a more immature signature than their in vitro counterparts (Fig. 3i). Importantly, analysis of drug-induced transcriptional changes strongly overlapped with direct targets of MLLAF9 in a primary human model of CD34+ MLLAF9 transformed cells[32] (Fig. 3i and Extended Data Fig. 3g). Overall, PRC1.1 is required for leukaemia cell differentiation driven by Menin and DOT1L inhibition and is essential for the therapeutic efficacy of Menin inhibition in vivo.

### Menin and DOT1L inhibition induce repressive chromatin modifications at MLL-FP target genes

Since PRC1.1 is thought to regulate the initiation of gene repression, we hypothesized that it may mediate responses to Menin and DOT1L inhibition by directly silencing MLL-FP target genes[37]. Repressive modifications are frequently deposited following the loss of active modifications, so we first defined MLL-FP target genes by assessing the changes in active marks. Using our Flag-tagged murine MLLAF9 model, we performed chromatin immunoprecipitation sequencing (ChIP–seq) after DOT1Li (72 h) or MENi (48 h) to profile MLLAF9, endogenous MLL1 and active chromatin marks associated with MLL-FP function (H3K79me2, H3K27ac, H3K9ac, H3K4me3 and H2BK120ub)[31,38].

As expected, MLLAF9 and MLL1 occupancy decreased after both treatments and H3K79me2 was specifically reduced by DOT1Li (Fig. 4a)[7,31]. While H3K27ac, H2BK120ub and H3K9ac were not globally affected, they were markedly reduced at a subset of direct 'MLLAF9 target genes' (top 100 MLLAF9-bound genes that were also significantly downregulated by both inhibitors, $n = 38$). Loss of these active modifications was concordant with the strong reduction in expression of these genes (Fig. 4b,c and Extended Data Fig. 4a,b).

We then assessed whether PRC1.1 directly represses these targets upon Menin or DOT1L inhibition. ChIP–seq for KDM2B (PRC1.1 subunit with DNA-binding capacity)[34], BCOR, H2AK119ub and

---

**Fig. 2 | PRC1.1 depletion blunts the transcriptional response to Menin and DOT1L inhibition. a**, A schematic of the 3′ RNA-seq experiments. **b,c**, Violin plots of the significantly up- and downregulated genes (false discovery rate (FDR) < 0.05) in non-silencing sgRNA (control) cells treated with SGC0946 for 72 h (**b**) or VTP50469 for 48 h (**c**) presented as the logFC relative to DMSO. The cell lines shown are murine MLLAF9 with non-silencing (control) or PCGF1 sgRNA (3 μM SGC0946 or 200 nM VTP50469), MOLM13 with non-silencing (control), BCOR or PCGF1 sgRNA (5 μM SGC0946 or 500 nM VTP50469) and MV4;11 with non-silencing (control), BCOR or PCGF1 sgRNA (5 μM SGC0946 or 500 nM VTP50469). Control cells are shown in blue, BCOR KO in green and PCGF1 KO in red. **d**, Box plots (the centre line represents the median, limits represent upper and lower quartiles and whiskers represent the maxima and minima if not exceeding 1.5× the interquartile range (IQR)) showing the subset of genes downregulated (logFC <−0.5, FDR <0.05 and baseline CPM >2) in control cells upon treatment with SGC0946 or VTP50469 for control and PCGF1 KO murine MLLAF9 (SGC0946 560 genes: $P = 6.6 \times 10^{-24}$; VTP50469 1,143 genes: $P = 1.6 \times 10^{-17}$), MOLM13 (SGC0946 255 genes: $P = 1.1 \times 10^{-32}$; VTP50469 391 genes: $P = 1.6 \times 10^{-17}$) and MV4;11 (SGC0946 263 genes: $P = 1.1 \times 10^{-19}$; VTP50469 720 genes: $P = 1.4 \times 10^{-15}$). $P$ values were calculated using a two-sided Welch's $t$-test

and adjusted using the Holm method. Data are from three biological replicates. **e**, Correlation plots of the logFC of SGC0946 (top, $R = 0.793$, $P = 1 \times 10^{-700}$) or VTP50469 (bottom, $R = 0.869$, $P = 1 \times 10^{-994}$) treatment relative to DMSO for control versus PCGF1 KO murine MLLAF9 cells. Genes highlighted in red are direct MLLAF9 target genes. The association was assessed using the Pearson method. **f**, A Venn diagram of significantly downregulated genes (logFC <−1 and FDR <0.1) in murine MLLAF9 cells treated with VTP50469 or SGC0946. Rescued genes (pale) were defined as those where the logFC downregulation was >0.5 more in control than PCGF1 KO cells (for example, if control logFC = −2 then KO LFC >−1.5). Highlighted genes are either direct MLLAF9 targets or associated with a stem-like expression programme. **g**, A heat map of murine MLLAF9 control or PCGF1 KO cells treated with DMSO, SGC0946 (3 μM for 48 h) or VTP50469 (200 nM for 48 h). The data are presented as the $z$-score normalized CPM. **h**, Venn diagrams of genes commonly downregulated in MOLM13, MV4;11 and murine MLLAF9 cells following treatment with either SGC0946 or VTP50469 relative to DMSO (logFC <−1 and FDR <0.05), which were rescued in the PCGF1 KO cells. **i**, RT–qPCR analysis of *Meis1* expression in MOLM13 and murine MLLAF9 cells treated with a VTP50469 time course (300 nM) relative to DMSO. Data represent mean ± s.d. from $n = 3$ experiments.

H3K27me3 revealed that KDM2B and H2AK119ub were enriched at active, H3K4me3-marked genes before treatment[39] (Fig. 4d and Extended Data Fig. 4c). Following treatment, global levels of these modifications were largely unaffected, with the notable exception of H2AK119ub, which displayed a modest global increase upon DOT1L inhibition, confirmed using spike-in normalized ChIP–seq (ChIP–Rx) (Extended Data Fig. 4d). Notably, following treatment, MLLAF9 target

genes acquired strong KDM2B and BCOR occupancy and displayed selective induction or spreading of H2AK119ub across regions previously bound by MLLAF9 (Fig. 4e–g and Extended Data Fig. 4e).

For a subset of PRC1.1-rescued genes (n = 11; for example, *Meis1*, *Jmjd1c* and *Six1*), increased H2AK119ub was accompanied by local H3K27me3 deposition, particularly with combined DOT1L and Menin inhibition (Fig. 4f,g). MOLM13 cells (human) also showed similar

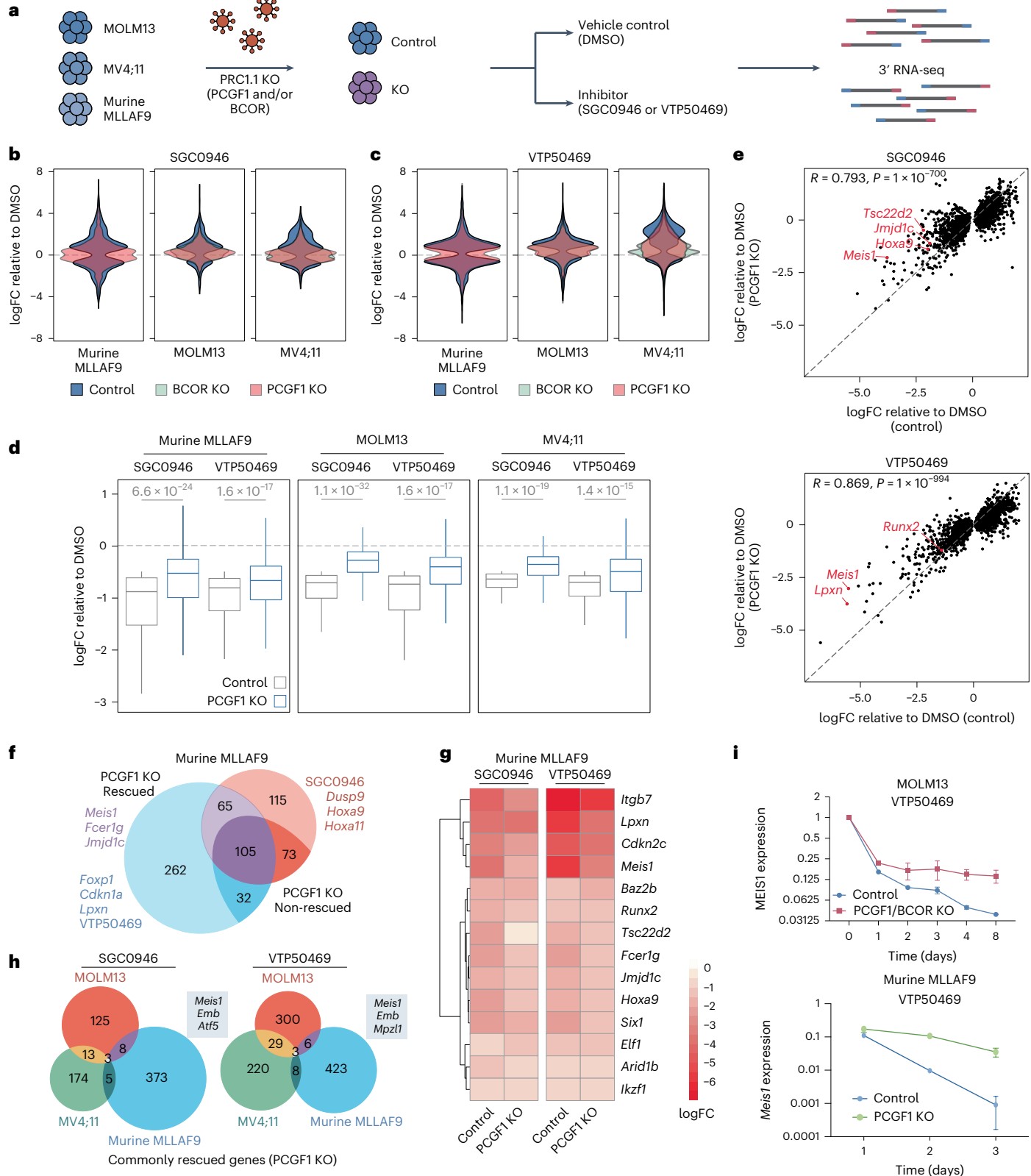

H2AK119ub induction and restricted H3K27me3 gains at key targets (Extended Data Fig. 4f). Together, these data show that MLLAF9 target genes undergo direct, selective Polycomb-mediated repression following Menin and DOT1L inhibition.

## H3K79me2 protects genes from PRC1.1-induced H2AK119ub to control the kinetics of gene repression following Menin inhibition

The requirement for PRC1 and PRC2 in directly silencing MLL-FP target genes suggests that MLL-FPs function, at least in part, by preventing Polycomb-mediated silencing. We proposed three potential mechanisms by which MLL-FPs could impede PRC1.1 activity: (1) Menin-dependent MLL occupancy may sterically block PRC1.1 binding; (2) transcriptional activity driven by MLL-FP may inhibit PRC1 function; or (3) H3K79me2, deposited by DOT1L, may have a previously unrecognized role in antagonizing PRC1 repression. Since Menin and DOT1L inhibitors disrupt distinct aspects of MLL-FP function, we leveraged these mechanistic differences to identify which process underlies this antagonism.

We first compared changes in PRC1.1 activity (measured through H2AK119ub induction) with changes in MLL-FP binding, MLL1 occupancy, transcriptional output and H3K79me2 across genes significantly downregulated by both inhibitors (log2FC <−0.5 and FDR <0.05, $n = 176$). Quantification of chromatin changes at the transcription start site (TSS) ± 5 kb revealed that neither MLLAF9 displacement, MLL1 loss nor transcriptional reduction reliably predicted increases in H2AK119ub. Instead, DOT1Li consistently triggered stronger H2AK119ub induction than MENi, despite both treatments causing similar transcriptional repression and MENi causing more extensive MLLAF9 eviction (Fig. 5a,b and Extended Data Fig. 5a).

We noticed a clear pattern emerging whereby loss of H3K79me2 correlated strongly with increased H2AK119ub. Consistent with this, global analysis revealed that DOT1Li treatment, which globally reduces H3K79me2, induced H2AK119ub (log2FC >0.5) at ~10 times more genes than MENi treatment (Fig. 5c). Remarkably, either H3K79me2 loss or baseline levels were well correlated ($r = 0.51$ and $r = 0.66$, respectively) with H2AK119ub induction after DOT1Li, whereas H3K27me3 was not (Fig. 5d and Extended Data Fig. 5b). Under MENi, only genes with both high MLLAF9 occupancy and substantial H3K79me2 loss displayed meaningful H2AK119ub induction (Fig. 5e and Extended Data Fig. 5c). Other active marks (H3K9ac and H2BK120ub) or transcription showed very weak correlation with H2AK119ub induction and therefore could not account for PRC1.1 activation (Extended Data Figs. 5d and 6a,b). Owing to the variability of H3K79me2 and H2AK119ub within a gene body, we next analysed their relationship genome wide using 5 kb bins (Fig. 5f). This higher-resolution approach confirmed a strong inverse correlation between H3K79me2 and H2AK119ub at baseline (Extended Data Fig. 5e). It also revealed that MENi only increased H2AK119ub at genomic regions where H3K79me2 was profoundly lost, whereas DOT1Li produced widespread H2AK119ub induction, including in regions where MENi had little effect due to global elimination of H3K79me2 (Fig. 5g and Extended Data Fig. 5f). Interestingly, even within individual genes, MENi sometimes caused uneven loss of H3K79me2, and only those subregions with strong depletion accumulated H2AK119ub (Fig. 5g). This further indicated that the presence or absence of H3K79me2, rather than MLL-FP binding or transcription, dictates PRC1.1 activation (Fig. 5g and Extended Data Fig. 5g,h).

We then examined whether the opposite (MENi-mediated gain of H3K79me2) could lead to reduced PRC1.1 activity. Indeed, genes with increased H3K79me2 after MENi (log2 fold change (logFC) >1, $n = 206$) showed significant decreases in H2AK119ub, strongly reinforcing the antagonistic relationship (Extended Data Fig. 5i).

To determine whether H3K79me2 affects PRC1.1 recruitment or its enzymatic activity, we performed ChIP–seq for RING1B (core enzymatic subunit of PRC1). Both MENi and DOT1Li increased RING1B occupancy at major MLL-FP targets, probably reflecting MLL1/MLL-FP eviction (Extended Data Fig. 6c). However, global changes in RING1B and BCOR binding did not correlate with baseline H3K79me2 levels, suggesting that H3K79me2 mainly regulates PRC1.1 activity rather than its recruitment. Interestingly, a subset of genes showed especially high RING1B recruitment accompanied by H3K27me3 gain, consistent with cooperative recruitment of canonical PRC1 following PRC2 activation[40,41] (Extended Data Fig. 6d,e).

Since no known H3K79 demethylases exists, H3K79me2 is thought to only be removed passively through nucleosome turnover during cell division[42]. This relative stability suggested that H3K79me2 might preserve a transcriptionally active 'memory' state, reminiscent of the role of H3K27me3 in repression[43]. This model predicts that MENi should cause delayed PRC1.1 activation as H3K79me2 decays slowly. Indeed, we found that PRC1.1 is not required for the early phase of transcriptional downregulation but is essential for later, sustained repression (Fig. 5h).

Temporal profiling after MENi showed that H3K79me2 levels declined gradually, consistent with the 12-h doubling time of our murine MLLAF9 cells, while H2AK119ub induction lagged behind this loss and preceded H3K27me3 accumulation (Fig. 5i,j and Extended Data Fig. 6g,h). These observations are consistent with the non-canonical recruitment model of PRC1.1 followed by PRC2 and suggest that H3K79me2 may function to delay Polycomb repression after eviction of Menin and MLL1 (ref. 41). Together, the data implicate DOT1L as a key antagonist of PRC1.1-mediated repression and a major

---

**Fig. 3 | PRC1.1 is required for efficient DOT1L and Menin inhibitor-induced differentiation in vitro and in vivo. a**, A CellRadar plot of the top 100 upregulated genes (FDR <0.05) after SGC0946 or VTP50469 in control (cyan and green, respectively) or PCGF1 KO (orange and red, respectively) murine MLLAF9 cells. Also, the top 100 upregulated genes following PCGF1 KO relative to control cells (blue). LT-HSC, long-term hematopoietic stem cell; ST-HSC, short-term hematopoietic stem cell; LMPP, lymphoid-primed multipotent progenitor; GM, granulocyte–monocyte; GMP, granulocyte–monocyte progenitor; CLP, common lymphoid progenitor; ETP, early T-cell progenitor; NK, natural killer cell; ProE, proerythroblast; CFUE, colony-forming unit–erythroid; MkP, megakaryocyte progenitor; MkE, megakaryocyte–erythroid progenitor; HPC, hematopoietic progenitor cell; CMP, common myeloid progenitor; MEP, megakaryocyte–erythroid progenitor; ProM, promonocyte; mDC, myeloid dendritic cell; pDC, plasmacytoid dendritic cell. **b**, Scatter plots of the CPM of selected genes (*MEIS1*, *HOXA13*, *SOCS2*, *HOXA10* and *MEF2C*) from RNA-seq data in untreated control, PCGF1 KO or BCOR KO MOLM13 and MV4;11 cells. Data are from three biological replicates. **c**, A histogram of the forward scatter (FSC) and side scatter (SSC) of control and PCGF1 KO murine MLLAF9 cells, normalized to mode. **d**, A histogram of Ly6G (APC) staining in control and PCGF1 KO murine MLLAF9 cells following treatment with DMSO, VTP50469 (500 nM) or SGC0946 (5 μM), normalized to mode (left). A scatter plot of the Ly6G mean fluorescence intensity (MFI) of data from $n = 2$ independent biological replicates (right). **e**, Bar plots of CD11b⁺ expression normalized to DMSO in control or PCGF1/BCOR KO MOLM13 cells treated with SGC0946, VTP50469 or SGC0946 + VTP50469 (5 μM + 500 nM). Mean represents ± s.d. of $n = 3$ biological replicates. **f**, A Kaplan–Meier survival curve of mice engrafted with MOLM13 control cells fed either control chow ($n = 5$) or 0.1% SNDX chow ($n = 5$) or PCGF1/BCOR KO cells fed either control chow ($n = 5$) or 0.1% SNDX chow ($n = 6$). **g**, Disease burden in the bone marrow and spleen (NT control $n = 12$, NT SNDX $n = 8$, KO control $n = 11$ and KO SNDX $n = 13$) and peripheral blood (NT control $n = 10$, NT SNDX $n = 7$, KO control $n = 9$ and KO SNDX $n = 10$) of mice transplanted with leukaemia cells and assessed by the percentage of human CD45⁺ cells. **h**, CD11b MFI expression of leukaemia cells from the indicated samples taken from the bone marrow at the end point (NT control $n = 4$, NT SNDX $n = 7$, KO control $n = 4$ and KO SNDX $n = 12$). The results are presented as mean ± s.d. **i**, A CellRadar plot of the top 100 upregulated genes in PCGF1/BCOR KO cells relative to control isolated from bone marrow as human CD45⁺ cells (left). A CellRadar plot showing the overlap of the downregulated genes in engrafted control MOLM13 cells treated with 0.1% SNDX chow (red) and human CD34⁺ MLLAF9 (brown) cells treated with Menin inhibitor (right).

barrier to Polycomb-mediated silencing in MLL-leukaemia, explaining why MLL-FPs exploit DOT1L enzymatic activity.

## Menin inhibition induces irreversible Polycomb-mediated repression in MLL-rearranged and NPM1c-mutant leukaemia

PRC2-mediated H3K27me3 is thought to confer stable, heritable gene repression downstream of PRC1 activity[39]. Since H3K79 methylation is relatively stable, it may delay the onset of PRC1.1-mediated silencing following the loss of transcriptional activators, potentially preventing premature stable repression by PRC2. To determine whether failure to deposit H2AK119ub impacts subsequent PRC2-mediated H3K27me3, we performed ChIP–seq for H3K27me3, H2AK119ub, H3K79me2 and MLLAF9 in control and PCGF1 KO cells treated with DMSO, DOT1Li or MENi. Genes were grouped into high, medium and low clusters of baseline H3K79me2, and changes in H2AK119ub and H3K27me3 were analysed. While DOT1Li still induced H2AK119ub at genes with high H3K79me2 in PCGF1 KO cells, the levels were substantially lower than in control (Fig. 6a and Extended Data Fig. 7a). This observation

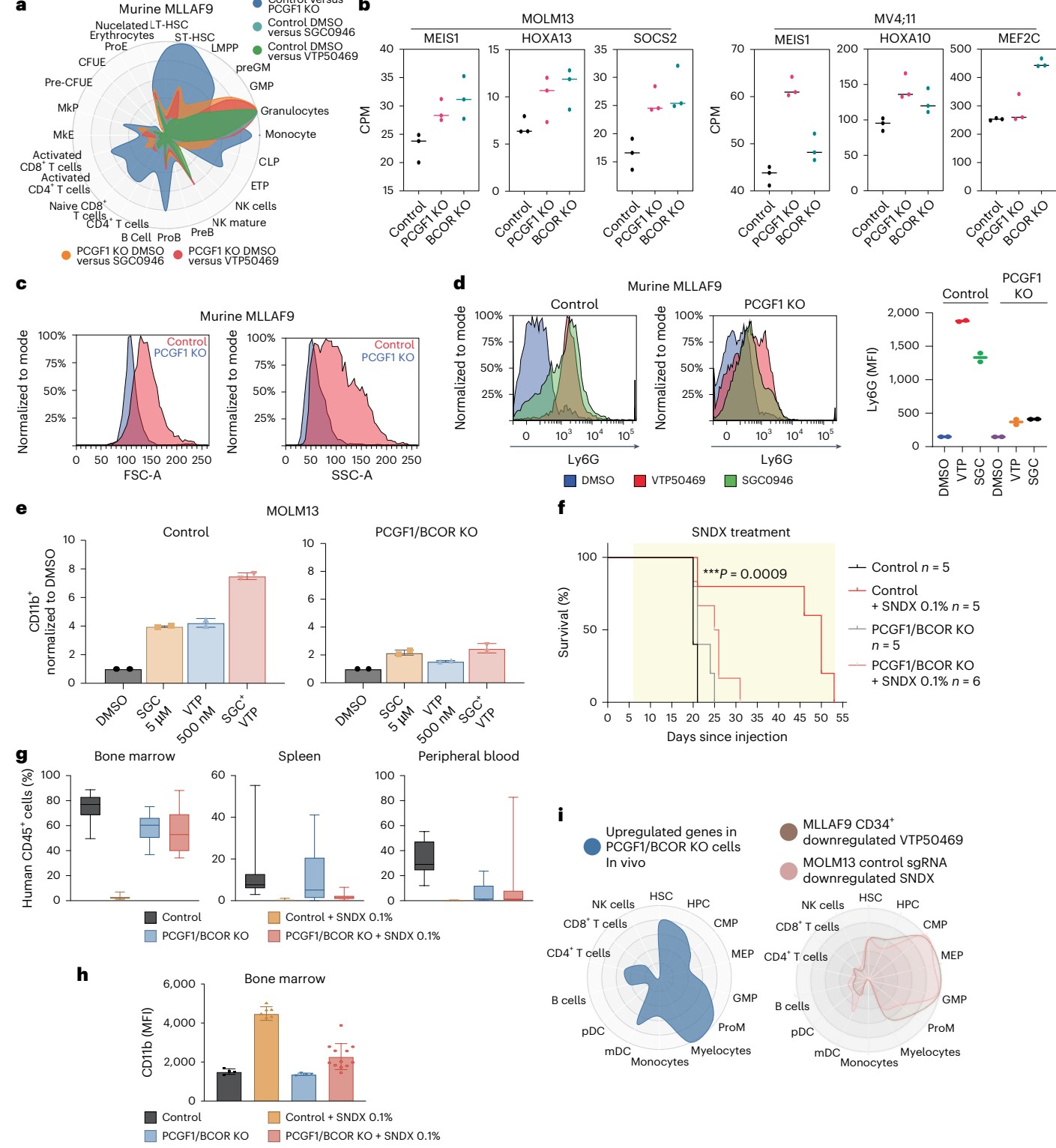

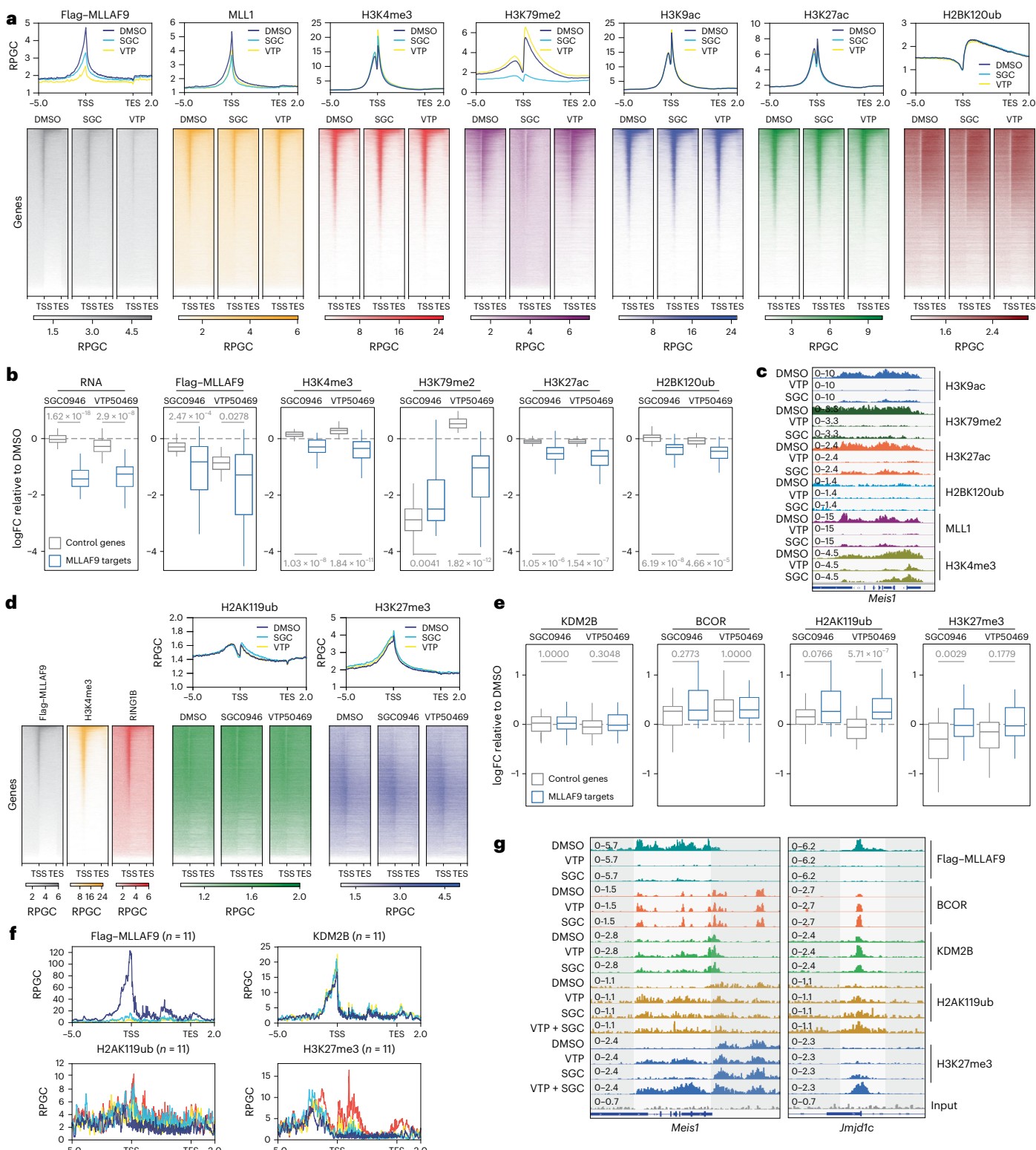

could be attributable to the function of other PRC1 complexes or owing to incomplete inactivation of PRC1.1. In support of the latter, double KO of PCGF1/BCOR in MOLM13 cells resulted in reduced baseline levels and the nearly abolished induction of H2AK119ub after DOT1L inhibition (Extended Data Fig. 6a). Importantly, rescue of PCGF1 expression in the KO cells restored the DOT1L-induced H2AK119ub (Extended Data Fig. 7b). In the murine MLLAF9 PCGF1 KO cells, H2AK119ub induction was strongly suppressed at target genes preventing subsequent deposition of H3K27me3 (Fig. 6b,c and

Extended Data Fig. 7c,d). Finally, global H3K79me2 was equally reduced by western blot and ChIP–seq in control and KO cells after DOT1Li (Extended Data Figs. 1m and 7e). In addition, both treatments also caused similar levels of MLLAF9 eviction in control and PCGF1 KO cells, which together confirm that PRC1.1 acts downstream of H3K79me2 loss and MLLAF9 displacement (Extended Data Fig. 7f).

Owing to the long half-life of H3K79 methylation and rapid kinetics of Menin inhibition[31], MENi should cause reversible gene expression changes early (owing to protection by H3K79me2), but irreversible

**Fig. 4 | Menin and DOT1L inhibition induce repressive chromatin modifications at MLL-FP target genes. a**, Heat maps and average profile plots of Flag−MLLAF9, MLL1, H3K4me3, H3K79me2, H3K9ac, H3K27ac and H2BK120ub ChIP−seq data for all genes in murine MLLAF9 cells treated with DMSO, SGC0946 (5 μM for 48 h) or VTP50469 (500 nM for 48 h). Genes are ranked from highest to lowest by MLLAF9 occupancy in DMSO. **b**, Box plots of RNA-seq and ChIP-seq data for Flag−MLLAF9, H3K4me3, H3K79me2, H3K27ac and H2BK120ub (from 5 kb upstream of the transcription start site (TSS) to 2 kb downstream of the transcription end site (TES)) in murine MLLAF9 cells. Data are plotted as the logFC of SGC0946 or VTP50469 relative to DMSO for each variable. The control genes (grey, $n = 48$) are the most highly expressed genes that are not significantly affected by SGC0946 and VTP50469 treatment. The MLLAF9 target genes (blue, $n = 38$) are those that are significantly downregulated (logFC <−0.5 and FDR <0.1) by both SGC0946 and VTP50469 and in the top 100 most highly bound genes by Flag−MLLAF9 ChIP−seq. $P = 1.62 \times 10^{-18}$, $P = 2.91 \times 10^{-8}$, $P = 2.47 \times 10^{-4}$, $P = 0.0278$, $P = 1.03 \times 10^{-8}$, $P = 1.84 \times 10^{-11}$, $P = 0.0041$, $P = 1.82 \times 10^{-12}$, $P = 1.05 \times 10^{-6}$, $P = 1.54 \times 10^{-7}$, $P = 6.19 \times 10^{-8}$ and $P = 4.66 \times 10^{-5}$ for control versus MLLAF9 target genes for RNA, Flag−MLLAF9, H3K4me3, H3K79me2, H3K27ac and H2BK120ub, respectively. **c**, Genome browser snapshot of canonical MLLAF9 target gene *Meis1* with the indicated samples and treatments. **d**, Heat maps and average profile plots of Flag−MLLAF9, H3K4me3, RING1B, H2AK119ub and H3K27me3 ChIP−seq data for all genes in murine MLLAF9 cells treated with either DMSO,

SGC0946 (5 μM for 48 h) or VTP50469 (500 nM for 48 h). Genes are ranked from highest to lowest by MLLAF9 occupancy in DMSO. **e**, Box plots of KDM2B, BCOR, H2AK119ub and H3K27me3 ChIP−seq data in murine MLLAF9 cells treated with DMSO, SGC0946 or VTP50469. Data are plotted as the logFC of treatment relative to DMSO for each condition. Gene subsets are the same as in **b**. $P = 1$, $P = 0.3048$, $P = 0.2773$, $P = 1$, $P = 0.0766$, $P = 5.71 \times 10^{-7}$, $P = 0.0029$ and $P = 0.17779$ for control versus MLLAF9 target genes for KDM2B, BCOR, H2AK119ub and H3K27me3, respectively. **f**, Average profile plots of Flag−MLLAF9, KDM2B, H2AK119ub and H3K27me3 across the same MLLAF9 target genes as in **b** and **e** that were rescued by PCGF1 KO ($n = 11$: *Man1a, Ssbp2, Meis1, Itgb5, Stau2, Ptprc, Tsc22d2, Sgip1, Jmjd1c, Six1* and *Msi2*) in murine MLLAF9 cells treated with DMSO, SGC0946 (5 μM for 48 h) or VTP50469 (500 nM for 48 h). The combination treatment (SGC0946 and VTP50469) is also shown for H2AK119ub and H3K27me3. **g**, Genome browser snapshots of *Meis1* and *Jmjd1c* in murine MLLAF9 cells with the indicated samples and treatments. The highlighted region shows where H2AK119ub and H3K27me3 are specifically induced after treatment with SGC0946 and VTP50469. For all box plots, the centre line represents the median, limits represent upper and lower quartiles and whiskers represent the maxima and minima if not exceeding 1.5× the IQR and *P* values were calculated using a two-sided Welch's *t*-test with adjustment for multiple comparisons using the Holm method.

repression once H3K27me3 is established. To test this, we treated control and PCGF1/BCOR KO MOLM13 cells with MENi for 8 h (before H3K27me3 establishment) or 96 h (after H3K27me3 establishment) and then washed out the drug. Importantly, at these timepoints where drug was withdrawn, both the control and PCGF1/BCOR KO cells showed a similar phenotype with cells remaining viable (Extended Data Fig. 7g). RT-qPCR analysis of *MEIS1* revealed full recovery after brief treatment in both conditions, whereas prolonged exposure before washout led to irreversible repression in control cells that reverted in KO cells (Fig. 6d). Stable *MEIS1* repression was also observed in OCIAML3 cells, suggesting that the switch to durable Polycomb-mediated silencing following Menin inhibition may be broadly relevant beyond MLL-rearranged leukaemia (Fig. 6e).

These findings were confirmed globally using 3′ RNA-seq (Fig. 6f and Extended Data Fig. 7h). Short MENi exposure (8 h) transiently downregulated target genes, with expression restored after drug

withdrawal in both control and PCGF1 KO murine MLLAF9 cells (Fig. 6g). By contrast, prolonged treatment (48 h) caused irreversible repression in control cells, whereas many target genes were rescued in PCGF1 KO cells (Fig. 6g–i and Extended Data Fig. 7i). Functionally, MENi treatment followed by withdrawal inhibited proliferation to a similar extent as continuous drug exposure, demonstrating that Menin inhibition can trigger potent anti-leukaemic effects without prolonged treatment (Fig. 6j and Extended Data Fig. 7j). Strikingly, while brief treatment with MENi caused minimal differentiation at the point of washout, it was sufficient to induce further differentiation following drug withdrawal, similar to what was seen with chronic drug exposure (Extended Data Fig. 7k).

Together, these results demonstrate that DOT1L, PRC1.1 and PRC2 act in a coordinated epigenetic network to control the timing and reversibility of gene repression. Stable H3K79 methylation protects genes from premature Polycomb-mediated silencing, and PRC1.1 is

**Fig. 5 | H3K79me2 protects genes from PRC1.1-induced H2AK119ub to control the kinetics of gene repression following Menin inhibition. a**, A schematic of the regions used for analysis of the ChIP−seq data in **a**–**e**: 5 kb upstream of the TSS to 5 kb downstream of the TSS. An example gene, *Jmjd1c*, is used to show the signal of H2AK119ub and Flag−MLLAF9 across the defined region. **b**, Box plots of the logFC of RNA-seq ($P = 0.1$), Flag−MLLAF9 ($P = 3.9 \times 10^{-9}$), H3K79me2 ($P = 1.53 \times 10^{-55}$) and H2AK119ub ($P = 3.8 \times 10^{-6}$) in SGC0946 (5 μM for 48 h) versus VTP50469 (500 nM for 48 h) treatment relative to DMSO in murine MLLAF9 cells. The regions plotted are those with a baseline H3K79me2 reads per million (RPM) >10 that are assigned to genes with a baseline counts per million (CPM) >2 and a logFC <−0.5 and FDR <0.05 after treatment with both SGC0946 and VTP50469 ($n = 176$) represented in grey and blue, respectively. **c**, A Venn diagram of the number of genes that contain regions with induction of H2AK119ub (logFC >0.5) after treatment with SGC0946 or VTP50469, with a baseline H3K79me2 RPM >10. **d**, Correlation plots of the logFC of H2AK119ub versus logFC of H3K79me2 (left, $R = -0.51$, $P = 1 \times 10^{-963}$) or baseline H3K79me2 RPM (right, $R = 0.66$, $P = 1 \times 10^{-1,806}$). Data are presented as the logFC of SGC0946 treatment relative to DMSO. **e**, The same as in **d** but data are presented as the logFC of VTP50469 treatment relative to DMSO. $R = -0.3$, $P = 1 \times 10^{-297}$ for logFC H3K79me2 and $R = 0.23$, $P = 1 \times 10^{-175}$ for baseline H3K79me2 RPM. Correlation was tested using the Pearson method. **f**, A schematic of tiling the genome into 5 kb bins. **g**, Top: Genome browser snapshots of canonical MLLAF9 target genes *Six1* and *Elf1* as well as a non-MLLAF9 target, *Brd4*. Bottom: box plots of regions (5 kb bins) where either both SGC0946 and VTP50469 induced a loss of H3K79me2 (logFC <−2, left) or where VTP50469 induced a loss of MLL1 and MLLAF9 (logFC <−1) without affecting H3K79me2 (right) plotted for H2AK119ub, H3K79me2 and MLL1 in cells treated

with either SGC0946 (grey) or VTP50469 (blue). $P = 0.76$, $P = 9.86 \times 10^{-5}$ and $P = 0.0188$ for H2AK119ub, H3K79me2 and MLL1, respectively, in genes where SGC0946 and VTP50469 induce loss of H3K79me2. $P = 1.6 \times 10^{-59}$, $P = 1 \times 10^{-799}$ and $P = 3.5 \times 10^{-10}$ for H2AK119ub, H3K79me2 and MLL1, respectively, in genes where only SGC0946 induces loss of H3K79me2. **h**, Box plots of the logFC of VTP50469 (300 nM) treatment relative to DMSO for 8 h ($P = 0.59$) or 48 h ($P = 9.6 \times 10^{-16}$) in control (grey) or PCGF1 KO (blue) murine MLLAF9 cells. *P* values were calculated using a two-sided Welch's *t*-test with no adjustment for multiple comparisons. Data are from three biological replicates. **i**, Box plots of the logFC of H3K79me2, H2AK119ub and H3K27me3 relative to DMSO in murine MLLAF9 cells treated with VTP50469 for 8 h (grey), 24 h (blue) or 48 h (red). The regions analysed ($n = 45$) had a baseline H3K79me2 RPM >25, logFC of H3K79me2 <−1 after VTP50469 treatment (500 nM for 48 h) and were assigned to genes that were significantly downregulated (logFC <−0.5 and FDR <0.05) by VTP50469 treatment (200 nM for 48 h). Data are from one representative replicate of two biological replicates with similar results. $P = 2.4 \times 10^{-14}$, $P = 8.2 \times 10^{-23}$ and $P = 2.2 \times 10^{-4}$ for H3K79me2 at 8 h versus 24 h, 8 h versus 48 h and 24 h versus 48 h, respectively. $P = 0.2$, $P = 5.19 \times 10^{-9}$ and $P = 9.9 \times 10^{-6}$ for H2AK119ub at 8 h versus 24 h, 8 h versus 48 h and 24 h versus 48 h, respectively. $P = 0.114$, $P = 6.6 \times 10^{-3}$ and $P = 0.177$ for H3K27me3 at 8 h versus 24 h, 8 h versus 48 h and 24 h versus 48 h, respectively. **j**, Genome browser snapshots of *Meis1* and *Jmjd1c* in murine MLLAF9 cells for time course ChIP−seq analysis of the indicated samples and tracks. For all box plots, the centre line represents median, the limits represent upper and lower quartiles and whiskers represent the maxima and minima if not exceeding 1.5× the IQR and *P* values were calculated using a two-sided Welch's *t*-test with adjustment for multiple comparisons using the Holm method unless stated otherwise.

essential for the transition from reversible to irreversible repression after Menin inhibition. The data also support a role of DOT1L in regulating transcriptional memory and highlights the therapeutic potential of transient epigenetic targeting.

## DOT1L–PRC1 antagonism is conserved across different cell types

The stable nature of H3K79me2 and its antagonism of PRC1.1-mediated H2AK119ub suggested that DOT1L may act more broadly in providing a memory of an active transcriptional state. To assess whether this mechanism is conserved beyond MLL-leukaemia, we profiled H2AK119ub and H3K79me2 after DOT1L inhibition in three wild-type MLL1 contexts: K562 cells, human erythroid progenitors (HuDEP-2) and mouse embryonic stem cells (mouse ES cells).

As expected, DOT1Li caused global loss of H3K79me2 across all cell types (Extended Data Fig. 8a). Strikingly, we also observed a modest but reproducible increase in H2AK119ub in each cell line (Fig. 7a–c). Clustering analyses showed that genes with high baseline H3K79me2

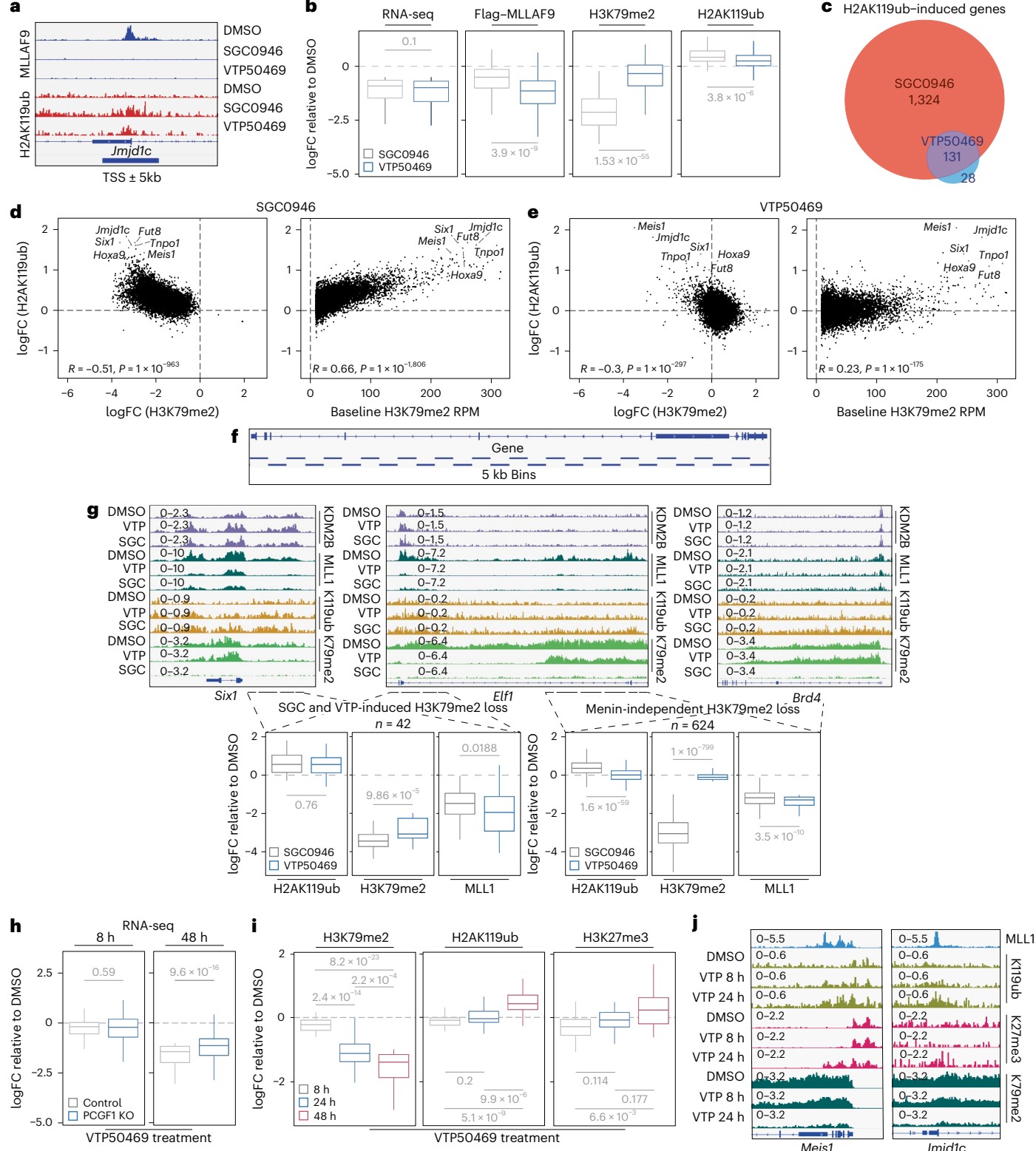

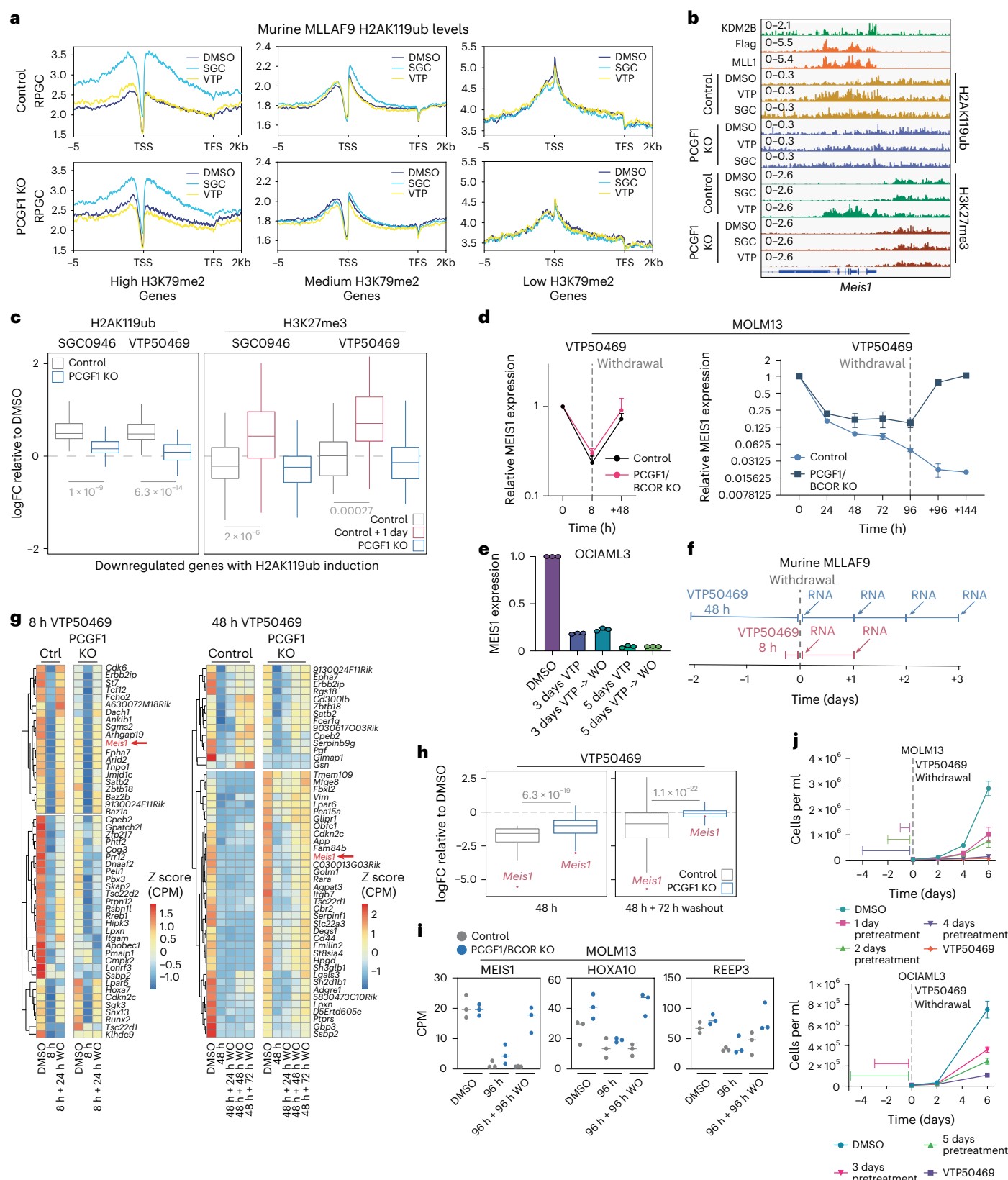

gained H2AK119ub (cluster 1), whereas genes with low H3K79me2 (already enriched for H2AK119ub) remained unchanged (cluster 2) (Fig. 7d–f). To confirm that this effect was not driven by loss of MLL binding, we treated K562 cells with a Menin inhibitor. MENi treatment did not affect H3K79me2 or induce H2AK119ub, despite being known to efficiently evict MLL1 in these cells (Extended Data Fig. 8b).

Thus, as in MLL-leukaemia cells, the increase in H2AK119ub reflects loss of H3K79me2 rather than changes in MLL occupancy (Fig. 7g,h and Extended Data Fig. 8c–e). Notably, H2AK119ub gains frequently spread across gene bodies and mirrored the pattern of baseline H3K79me2 while intergenic regions were unaffected, indicating a tight coupling between local H3K79me2 loss and PRC1 activity.

**Fig. 6 | Menin inhibition induces irreversible Polycomb-mediated repression in MLL-rearranged and NPM1c-mutant leukaemia. a**, Average profile plots of H2AK119ub in control and PCGF1 KO murine MLLAF9 cells treated with DMSO, SGC0946 (5 μM for 48 h) or VTP50469 (500 nM for 48 h). Plots were divided into three clusters (high, medium and low) by k-means clustering of control cell H3K79me2 levels in DMSO. **b**, A Genome browser snapshot of *Meis1* for the indicated samples and treatments in control or PCGF1 KO murine MLLAF9 cells. **c**, Box plots of H2AK119ub and H3K27me3 at H2AK119ub-induced genes in control or PCGF1 KO murine MLLAF9 cells treated with DMSO, SGC0946 or VTP50469. Induced genes (n = 54) were significantly downregulated (logFC <−1 and FDR <0.05) after VTP50469 treatment (200 nM for 48 h) and had TSS ±5 kb regions with a baseline H3K79me2 RPM >10 that saw an induction of H2AK119ub (logFC >0.5). Control and PCGF1 KO cells were treated for 48 h with both drugs; however, there is an extra timepoint for control cell H3K27me3 (red), which were treated for 72 h. $P = 1 \times 10^{-9}$ and $P = 6.3 \times 10^{-14}$ for H2AK119ub in control versus PCGF1 KO cells treated with SGC0946 and VTP50469, respectively. $P = 2 \times 10^{-6}$ and $P = 2.7 \times 10^{-3}$ for H3K27me3 in control cells treated with SGC0946 for 48 h (grey) versus 72 h (red) or with VTP50469 for 48 h (grey) versus 72 h (red), respectively. **d**, RT–qPCR analysis of relative *MEIS1* levels after treatment with VTP50469 (100 nM) for 8 h (left) or a time course (right) and drug withdrawal (indicated by the dotted vertical line) relative to DMSO-treated MOLM13 cells. Data represent mean ± s.d. from n = 3 independent experiments. **e**, RT–qPCR analysis of relative

*MEIS1* levels after VTP50469 treatment for 3 or 5 days and drug withdrawal as indicated in OCIAML3 (NPM1 mutant cell line). Data represent mean ± s.d. from n = 3 independent experiments. WO, washout; VTP, VTP50469 **f**, A schematic overview of the drug withdrawal RNA-seq experiment in murine MLLAF9 cells. RNA was collected at the indicated timepoints following withdrawal. **g**, A heat map of the top 50 downregulated genes in control cells treated with 300 nM VTP50469 for 8 h or 48 h plotted for control and PCGF1 KO murine MLLAF9 cells treated with VTP50469 and subsequent drug withdrawal. Data are presented as the z-score CPM. **h**, Box plots of the logFC in RNA of VTP50469 treatment for 48 h relative to DMSO ($P = 6.3 \times 10^{-19}$) or 48 h plus 72 h washout ($P = 1.1 \times 10^{-22}$) in control and PCGF1 KO murine MLLAF9 cells. P values were calculated using a two-sided Welch's t-test with no adjustment for multiple comparisons. Data are from three biological replicates. **i**, Scatter plots of CPM for selected genes from RNA-seq data in control or PCGF1/BCOR KO MOLM13 cells treated with VTP50469 for 96 h or 96 h + 96 h post-withdrawal. Data from three biological replicates. **j**, A proliferation assay of MOLM13 and OCIAML3 cells pretreated with VTP50469 (500 nM), constant DMSO or constant VTP50469 (500 nM). Data points represent mean ± s.d. from n = 3 independent experiments. For all box plots, the centre line represents the median, the limits represent upper and lower quartiles and the whiskers represent the maxima and minima if not exceeding 1.5× the IQR and P values were calculated using a two-sided Welch's t-test with adjustment for multiple comparisons using the Holm method unless stated otherwise.

---

Since transcriptional inhibition itself can rapidly recruit PRC2 to chromatin, we next asked whether potent transcriptional shutdown may be sufficient to induce repressive histone modifications[15]. In mouse ES cells treated with the transcription elongation inhibitor (DRB for 8 h) SUZ12 was robustly recruited to chromatin, confirming global PRC2 engagement (Extended Data Fig. 8f,g). However, neither H3K27me3 nor H2AK119ub levels changed, and SUZ12 accumulation occurred even at previously lowly expressed genes suggesting that local gene activity may not be a primary factor in antagonizing the catalytic activity of PRC2 (Extended Data Fig. 8f,g). These results imply that transcriptional status alone does not dictate PRC1 and PRC2 activity and that PRC2 recruitment does not immediately translate into stable repressive chromatin modification. Consistent with this, DOT1Li did not broadly reduce transcriptional output in K562 or HuDEP-2 cells despite increasing H2AK119ub (Fig. 7i and Extended Data Fig. 8h).

Rather than resulting in the broad silencing of transcription, we reasoned that DOT1L inhibition may instead prime cells for future

repression when exposed to an additional differentiation cue. To test this, we pretreated HuDEP-2 cells with DOT1Li or DMSO for 7 days, then induced erythroid differentiation (Fig. 7j). Before differentiation, both conditions were indistinguishable. However, DOT1Li-pretreated cells underwent accelerated differentiation, marked by earlier loss of the CD49d surface marker (Fig. 7k,l and Extended Data Fig. 8i,j). Transcriptomic profiling showed that a subset of genes normally downregulated during erythroid maturation became more strongly repressed following DOT1Li pretreatment (Extended Data Fig. 8k). Importantly, these same genes had gained H2AK119ub during the pretreatment phase, linking DOT1L loss, PRC1 activation and enhanced differentiation associated gene silencing.

## H3K79me2/3 is sufficient to antagonize the histone ubiquitin ligase activity of PRC1

Having ruled out a transcription-dependent mechanism for inhibiting PRC1 activity, we next investigated whether H3K79 methylation directly

---

**Fig. 7 | DOT1L–PRC1 antagonism is conserved across different cell types. a–c**, Global average profile plots of H2AK119ub and H3K79me2 ChIP–seq in K562 (**a**), HuDEP-2 (**b**) and mouse ES cells (**c**) treated with DMSO or SGC0946 5 μM for 7 days. **d–f**, Top: heat maps of the same ChIP–seq data as in **a–c**, respectively, separated into two clusters by k-means clustering. Bottom: average profile plots of H2AK119ub occupancy in clusters 1 and 2 from **d–f**, respectively. **g**, Genome browser snapshots of exemplar gene *CD2AP* in K562 cells treated with DMSO, VTP50469 500 nM or SGC0946 5 μM. **h**, A Genome browser snapshot of exemplar gene *ZBTB7A* in HuDEP-2 cells treated with DMSO or SGC0946 5 μM. **i**, Box plots (the centre line represents the median, the limits represent the upper and lower quartiles and the whiskers represent the maxima and minima if not exceeding 1.5× the IQR) of the logFC of SGC0946 treatment relative to DMSO for H3K79me2, H2AK119ub and RNA in K562 cells for genes with 'high' H2AK119ub (grey) or 'high' H3K79me2 (blue). 'High' is defined as the top 500 genes by baseline RPM. P values were calculated using a two-sided Welch's t-test with no adjustment for multiple comparisons. Data are from three biological replicates. $P = 2.1 \times 10^{-132}$, $P = 9.8 \times 10^{-130}$ and $P = 2.9 \times 10^{-7}$ for H2AK119ub, H3K79me2 and RNA, respectively, in genes with high H2AK119ub versus high H3K79me2. **j**, A schematic of the HuDEP-2 erythroid differentiation experiment. Flow cytometry was performed on days 0, 3, 4, 5, 6 and 7 and RNA-seq on day 2. **k**, A histogram of CD49d in the DMSO and SGC0946 pretreated HuDEP-2 cells over the course of differentiation. **l**, The percentage of CD49d+ cells (relative to day 0) in DMSO and SGC0946 pretreated HuDEP-2 cells over the course of differentiation. Data are from two technical replicates representative of two biological replicates with similar results. **m**, A schematic representation of the in vitro histone ubiquitination assay. E1 and E2 represent UBA1 and UbcH5c, respectively, E3 represents the PRC1.1 or PRC1.4 protein complex dimers consisting of PCGF1–RING1B or

PCGF4–RING1B, respectively. **n**, Western blots of the in vitro ubiquitination assay comparing the activity of RING1B–PCGF1 (PRC1.1) or RING1B–PCGF4 (PRC1.4) on either unmodified nucleosome core particles (H3K79me0) or nucleosome core particles that were reconstituted using octamers that contain an H3 di- or tri-methyl-lysine analogue (MLA) at position 79 (H3K79me2 MLA or H3K79me3 MLA, respectively, as indicated). Black triangles represent twofold serial dilutions of the substrate nucleosomes, starting from 1,000 nM. PRC1 concentration is indicated. Ubiquitination was detected by immunoblotting using anti-H2AK119ub antibodies. H3 and MBP immunoblots were used as loading controls to detect the nucleosome substrate and MBP-tagged PCGF1/4 proteins, respectively. **o**, Ubiquitination activity results from **n** are shown as a bar plot. The means represent the relative ubiquitination activities presented in the form of H2AK119ub densitometry values normalized to the average signal of the same blot. The error bars indicate the s.d. over three independent replicates of ubiquitination reactions that were carried out on three different days and were then subjected to immunoblotting simultaneously. Values for each replicate are shown as dots. **p**, A schematic of the chromatin switch from active to stable repression at MLL-FP target genes following Menin inhibition. Early after eviction of the MLL-FP, H3K79me2 is retained and opposes PRC1.1-mediated H2AK119ub deposition. Following prolonged Menin inhibition, H3K79me2 is turned over, enabling increased PRC1.1 activity. Consequently, increased H2AK119ub is then recognized by PRC2, which deposits H3K27me3 resulting in stable epigenetic repression. **q**, A schematic representation of epigenetic marks associated with gene activation (blue) or repression (red). We propose that H3K79me2 serves as a memory device for active transcription, balancing the long-lived memory of repression provided by H3K27me3. The antagonistic relationship between H3K79me2 and H2AK119ub is denoted by a red arrow.

antagonizes PRC1. Although no high-resolution PRC1.1–nucleosome structure exists, PRC1.4 structures show PCGF4 positioned in close proximity to H3K79 (Extended Data Fig. 9a). PCGF4 and PCGF1 are paralogues that define the PRC1.4 and PRC1.1 complexes, respectively, and are required for the histone ubiquitin ligase activity of PRC1[44].

This supported the possibility that H3K79 methylation directly interferes with PRC1 function.

Although H3K79me2 is the predominant cellular readout of DOT1L activity, H3K79me3 is also deposited at active genes and reflects longer DOT1L residence time, making both modifications

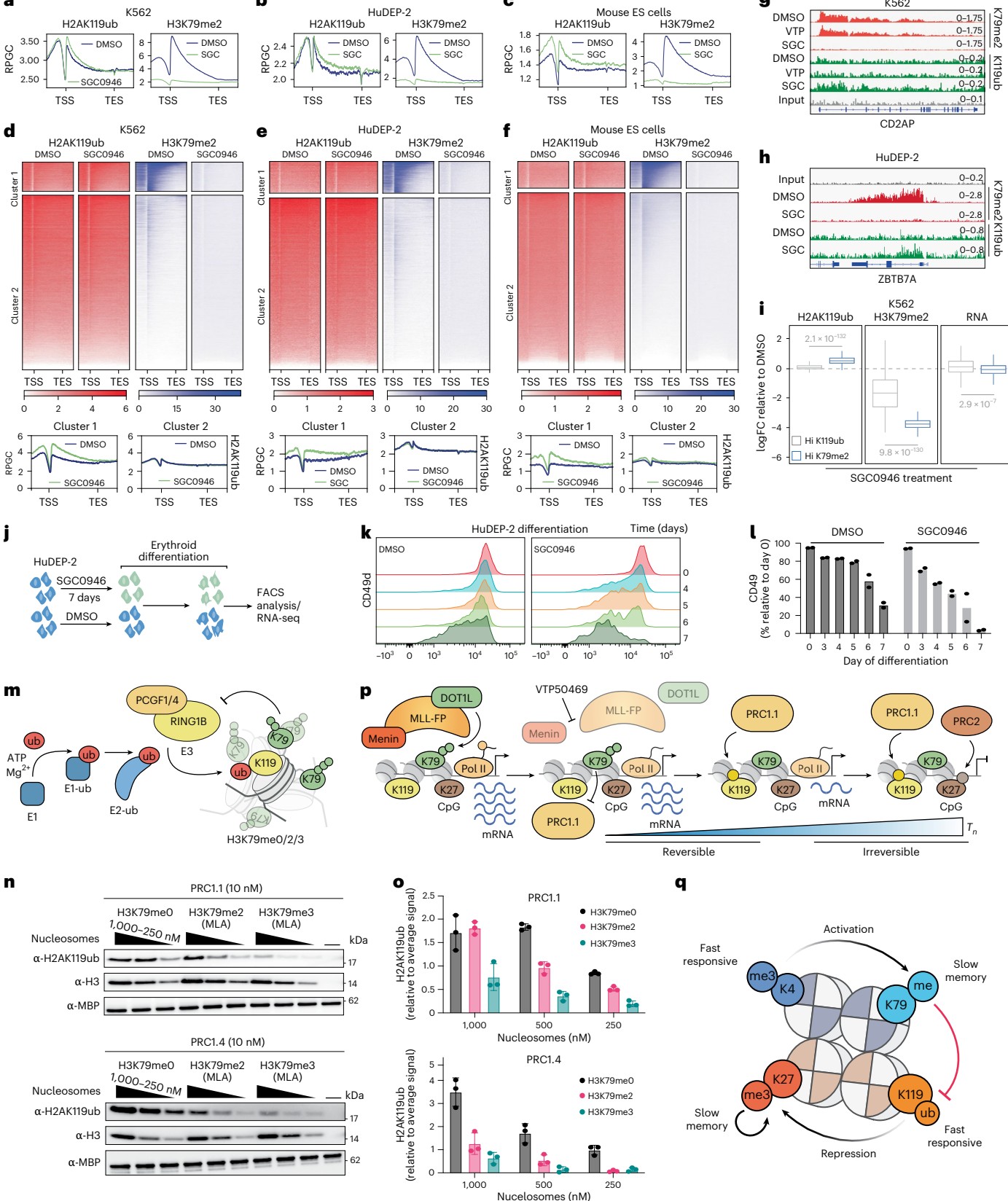

biologically relevant[45]. To test whether H3K79me2/3-modified nucleosomes are sufficient to reduce PRC1 activity, we performed in vitro ubiquitination assays with recombinant human PRC1.1 and PRC1.4 complexes (RING1B–PCGF1 and RING1B–PCGF4) (Fig. 7m and Extended Data Fig. 9b,c).

As substrates, we used either unmodified nucleosome core particles (H3K79me0) or nucleosomes reconstituted with octamers containing MLAs at H3K79 in either the di- or tri-methyl state (H3K79me2 MLA or H3K79me3 MLA, respectively; Extended Data Fig. 9d,g,h).

Substrate titration experiments revealed that ubiquitination of H3K79me2-modified nucleosomes is inhibited to approximately half the rate observed with unmodified nucleosomes (Fig. 7n,o). This inhibition was even stronger with H3K79me3 nucleosomes, which showed an approximately four-fold reduction in the ubiquitination rate. A time course assay confirmed linear reaction progression during the chosen 30-min window (30 min; Extended Data Fig. 9e,f). Importantly, these effects were observed for both PRC1.1 and the core PRC1.4 complexes, indicating that H3K79me2/3 directly antagonizes the ubiquitination activity of the RING1B–PCGF catalytic module (Fig. 7n,o). Together, these biochemical data demonstrate that H3K79 methylation is sufficient to inhibit PRC1 activity, providing a mechanistic explanation for the DOT1L–PRC1 antagonism observed in cells.

## Discussion

The balance between active and repressive chromatin factors underpins the regulatory potential of the epigenome, enabling cells to fine-tune gene expression, control transcriptional kinetics and establish epigenetic memory[5,46–50]. In MLL-leukaemia, MLL-FPs disrupt the normal function of MLL1 in part through aberrant recruitment of DOT1L and deposition of H3K79 methylation. Yet the mechanistic basis for how DOT1L regulates transcription has remained incompletely defined[51].

Using functional genomics, epigenomics and biochemical reconstitution, we have uncovered a fundamental role for DOT1L in preventing Polycomb repression mediated by PRC1.1 and, subsequently, by PRC2. Although PRC2 is directly antagonized by the active chromatin marks H3K4me3 and H3K36me3[52], no activating modification has previously been shown to inhibit PRC1 activity. Our data demonstrate that H3K79me2/3 is sufficient to antagonize PRC1 catalytic activity, revealing a previously unrecognized mechanism of histone crosstalk (Fig. 7m–o). The reduced activity of PRC1 on H3K79me2/3-marked nucleosomes also provides a mechanistic explanation for the synergy recently reported between H3K79 and H3K36 methyltransferases[53], with the former blocking PRC1 and the latter blocking PRC2. This suggests that inhibition of Polycomb complexes by active chromatin may represent a broad principle preventing the deposition of repressive marks on actively transcribed regions.

MLL-FPs appear to hijack this conserved mechanism to further shift the MLL–Polycomb balance towards activation. This insight highlights a therapeutic opportunity: Menin inhibitors, by displacing MLL-FP and reducing aberrant DOT1L recruitment, may enable PRC1.1 to initiate stable repression at target genes essential for leukemogenesis (similar observations by us and others have been made in AML cells harbouring wild-type MLL1)[54]. Our findings also help reconcile prior observations that were difficult to explain solely through altered transcriptional elongation. Although acute degradation of MLLAF9 causes rapid transcriptional downregulation, the initial effects on target genes are modest relative to later timepoints[31,32]. Similarly, genetic de-induction of MLLAF9 leads to progressive repression of *Meis1* over several days[55]. Our results reveal an additional layer of MLL-FP function: to protect target genes from Polycomb repression by maintaining high H3K79 methylation.

Menin and DOT1L inhibition both induce PRC1.1 activity at MLL-FP target genes, yet PCGF1 deletion does not completely prevent H2AK119ub induction. This probably reflects compensation by other PRC1 complexes or residual PRC1.1 activity. A more intriguing observation is the difference between global H2AK119ub induction after DOT1L inhibition versus the limited response to Menin inhibition. Since Menin inhibitors evict MLLAF9, they are remove the oncogenic pool of DOT1L recruited by the fusion protein. However, native DOT1L–MLLT10 complexes remain unaffected and continue depositing H3K79 methylation[31]. Thus, the extent to which a gene becomes PRC1.1 repressed following Menin inhibition depends on the ratio of oncogenic to native DOT1L activity[31]. Genes retaining high H3K79me2/3 that is deposited by the native DOT1L complex remain protected from PRC1.1 and do not gain H2AK119ub upon Menin inhibition. Consistent with this idea, global loss of H3K79me2/3 by DOT1L inhibition leads to a widespread induction of H2AK119ub across multiple cell types (Fig. 7a–c). Importantly, while our biochemical experiments show that H3K79me2/3 alone is sufficient to inhibit the activity of PRC1, future structural studies will be essential to define the molecular basis for this antagonism.

The unusually long half-life of H3K79me2/3, arising from the absence of demethylases[22], provides a buffering mechanism that delays Polycomb-mediated repression following transcriptional downregulation. This temporary protection may prevent premature stable repression caused by transient transcriptional fluctuations[56]. The stability of H3K79 methylation may also slow transcriptional shut down, generating a graded response to stimuli[11]. We propose that H3K79me2 endows active genes with a memory of prior activation, analogous to the repressive memory encoded by H3K27me3 (ref. 57)This memory is linked to cell division (when H3K79 methylation is diluted), as dilution during replication couples the pace of epigenetic change to proliferation. Such coupling provides a likely mechanistic rationale for the accelerated differentiation of erythroid progenitors after DOT1L inhibition (Fig. 7k–m). It is tempting to speculate that these insights may also explain roles of DOT1L in cellular plasticity, cancer evolution and induced pluripotent stem cell reprogramming[58–60].

Furthermore, such a core function of DOT1L in maintaining cellular stability may account for toxicities observed with DOT1L inhibitors, supporting the rationale for Menin inhibitors as a more selective alternative. Indeed, recent first-in-human clinical trials of relapsed refractory AML patients with NPM1c mutant or MLL rearrangements has resulted in FDA approval and the potential to transform the natural history of this disease[26].

Taken together, we have shown that MLL-leukaemias aberrantly recruit DOT1L to exploit a fundamental epigenetic mechanism: the antagonism of Polycomb-mediated repression via H3K79 methylation. Consequently, transient Menin inhibition is sufficient to erode this protection enabling Polycomb to trigger irreversible differentiation and cell death. These findings suggest that shorter, more intense dosing schedules may maximize therapeutic benefit with reduced toxicity. Finally, our work provides an explanation for oncogenic transformation by MLL-FPs (Fig. 7p) and a unifying model in which the MLL–Polycomb axis encodes a balanced memory of activation and repression (Fig. 7q).

## Online content

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

## Methods

All animal experiments were performed in compliance with the ethical regulations of the institutional Animal Care and use Committee at the Dana-Farber Cancer Institute (protocol no. 16-021). All antibodies used in this study are commercially available polyclonal antibodies that have been validated per manufacturer's websites. All oligonucleotide sequences are provided and were sourced from Integrated DNA Technologies (IDT).

### Cell lines

Murine MLLAF9 leukaemia cells were generated by magnetic bead selection (Miltenyi Biotec) of c-KIT+ cells from whole female mouse bone marrow and retrovirally transduced with a Flag−MLLAF9 construct. MOLM13 (ACC 554), MV4;11 (CRL9591), K562 (CCL-243), OCIAML3 (ACC 582) and HEK293T (CRL-3216) were obtained from ATCC or DSMZ. Mouse ES and HuDEP-2 cells were obtained from collaborators. All cell lines were routinely subjected to regular mycoplasma testing and underwent short tandem repeat profiling. K562, MV4;11, OCIAML3 and MOLM13 cells were cultured in RPMI-1640 supplemented with 2 mM GlutaMAX, 100 IU ml⁻¹ penicillin, 100 µg ml⁻¹ streptomycin and 10% foetal bovine serum (FBS; Bovogen). HEK293T cells were cultured in DMEM supplemented with 2 mM GlutaMAX, 100 IU ml⁻¹ penicillin, 100 µg ml⁻¹ streptomycin and 10% FBS. MLLAF9 cells were cultured in RPMI-1640 supplemented with 2 mM GlutaMAX, 100 IU ml⁻¹ penicillin, 100 µg ml⁻¹ streptomycin, 20% FBS, and IL-3 10 ng ml⁻¹.

All cell lines were cultured in 5% $CO_2$ at 37 °C with the exception of HEK293T cells which were cultured in 10% $CO_2$. Although all the cell lines used corroborated the findings, MOLM13 cells showed a stronger phenotype and were thus used for most of the subsequent validation. Murine MLLAF9 cells were engineered to accurately assess MLLAF9 chromatin occupancy and were therefore used for most of the experiments characterizing chromatin changes using ChIP−seq. Unless otherwise specified, doses used for SGC0946 and VTP50469 were 5 µM for 72 h and 500 nM for 48 h, respectively, in MOLM13 and MV4;11 cells, and 3 µM for 72 h and 300 nM for 48 h for murine MLLAF9 cells.

### CRISPR sgRNA library

The targeted sgRNA CRISPR library is a custom-built library containing ~7,239 human and ~7,395 mouse sgRNAs targeting ~1,144 chromatin regulators (6 sgRNAs per gene), as well as non-targeting and safe targeting control sgRNAs. Library sgRNAs were expressed in a modified pKLV-U6gRNA(BbsI)-EF1apuro2ABFP lentiviral sgRNA expression vector that encodes puromycin and BFP selection markers obtained from Addgene no. 50946, a gift from Kosuke Yusa[61].

The whole genome Bassik Human CRISPR Knockout Library was obtained from Addgene. This library contains 212821 unique guides targeting a total of 20,549 genes in the pMCB320 backbone (without Cas9).

### CRISPR−Cas9-mediated gene disruption

sgRNA oligonucleotides (IDT) were cloned into lentiviral expression vectors pKLV−U6gRNA(BbsI)−PGKpuro2ABFP (Addgene no. 50946, a gift from Kosuke Yusa). Oligonucleotide sequences are listed in Supplementary Table 2. For CRISPR−Cas9-mediated gene disruption, cells were first transduced with FUCas9Cherry (a gift from Marco Herold, Addgene no. 70182) and sorted for mCherry expression. To generate polyclonal populations with targeted gene disruption, cells were subsequently transduced with pKLV-gRNA-PGKpuro2ABFP encoding either gene-specific sgRNAs or with a control sgRNA targeting a 'safe' genomic location with no annotated function.

Cells infected with pKLV-gRNA-PGKpuro2ABFP were selected with 2 µg ml⁻¹ puromycin for 72 h, commencing 48 h after transduction. Efficient functional CRISPR−Cas9-mediated gene disruption of target genes was confirmed by immunoblot or Sanger sequencing using tracking of indels by decomposition analysis.

### Virus production and transduction

The lentivirus was produced by triple transfection of HEK-293ET cells with a lentiviral transfer vector and the packaging plasmids psPAX2 and pCMV-VSV-G at a 0.5:0.35:0.15 ratio. All transfections were performed using polyethylenimine. Viral supernatants were collected 48 h following transfection, filtered through a 0.45 µm filter and added to target cells.

### sgRNA competition assays

Murine MLLAF9, MV4;11 or MOLM13 Cas9 cells were transduced with lentivirus expressing a gene-specific sgRNA. The percentage of BFP-positive cells was measured following infection as indicated in the figures and normalized to the percentage of BFP at day 3 or 5 post infection, as indicated in the figures.

### Cell proliferation and dose−response assays

Cells were seeded at a consistent density before treatment in duplicate or triplicate and treated with DMSO, VTP50469 (MENi) or SGC0946 (DOT1Li) over the indicated time period. Cell number was calculated using the Northern Lights Flow Cytometer (Cytek). Dose response assays were performed using CellTitre-Blue (Promega) and measured using the FLUOstar Omega microplate reader (BMG Labtech).

### Rescue experiments

PCGF1 wild-type cDNA sequences were ordered from IDT as geneBlocks and included sgRNA-resistant mutations and an N-terminal Flag tag. Sequences were cloned into the HIV1-GFP lentiviral vector using the BAMHI and XHOI restriction sites. Lentiviral infection of PCGF1 or empty vector were performed in either PCGF1 KO or parental MOLM13 cells and subjected to various assays.

### ChIP−seq

Briefly, for each ChIP, 20 million cells were crosslinked for 15 min with 1% formaldehyde. Crosslinked material was sonicated to ~200–1,000 bp using the Covaris Ultrasonicator e220. Sonicated material was incubated overnight with each antibody, then incubated for 3 h with Protein A magnetic beads. Beads were washed with low- and high-salt wash buffers and TE, before being eluted and de-crosslinked overnight. DNA was purified using Qiagen Minelute columns. Then 5 µg of antibody was used per ChIP except for H3K27ac (4 µg) and H3K79me2 (3 µg). Sequencing libraries were prepared from eluted DNA using the NEBNext Ultra II DNA Library Prep Kit for Illumina kit and the NEBNext Multiplex Oligos for Illumina (96 unique dual-index primer pairs) for indexing. Libraries were sequenced on the NovaSeq 6000. For a more detailed description, please see ref. 62.

### ChIP−seq analysis

Reads were aligned to GRCh38 for human cell lines (MOLM13, MV4;11, K562 and HuDEP-2) and GRCm38 for murine cell lines (MLLAF9 and mouse ES cells). Duplicate reads were filtered by Picard before alignment using Bowtie2. BAM files were converted to TDFs using igvtools (count) for genome browser snapshots in IGV. BAM files were converted to reads per genome content (RPGC) normalized bigwig files using BamCoverage. Heat maps and profile plots were made using Deeptools computeMatrix scale regions for the region from 5 kb upstream of the transcription start site (TSS) to 2 kb downstream of the transcription end site (TES). The reference BED files for the plots were generated using the GenomicFeatures::genes() command to extract TSS, TES and chromosome information from the TxDb.Mmusculus.UCSC. mm10.ensGene annotation. This was then subset further for more specific analyses. For Fig. 4 specifically, the box plots for ChIP−seq data were created by reading computeMatrix scale-region matrices into R with profileplyR. Matrices were imported with importdeepToolsMat() followed by profileplyr::summarize(data, fun = rowMeans, output = 'long') to get the data ready to plot as a box plot. The TSS ±5 kb

regions file was generated using the GenomicFeatures::promoters(upstream = 5000, downstream = 5000) command on the TxDb.Mmusculus. UCSC.mm10.ensGene object. Similarly, the GRChm38 was divided into 5 kb bins using the GenomicFeatures::tileGenome (tilewidth = 5000) command on the BSgenome.Mmusculus.UCSC.mm10 object. The ChIP signal across these regions was read using Rsubread::featureCounts() and then normalized to sequencing depth as reads per million (RPM). The RPMs were then used in subsequent visualizations with ggplot2. Region analyses used a DMSO-treated control cell H3K79me2 RPM cut off of 10–25 and an induction logFC cut off of 0.5–1.

## Immunoblotting
Cells were lysed in buffer containing 2% SDS (Sigma), 0.5 mM EDTA (Sigma), 20 mM HEPES (Sigma) with a 1 in 50 dilution of complete Protease Inhibitor (Sigma) added fresh.

Lysates were heated to 70 °C in SDS sample buffer with 50 mM dithiothreitol (DTT) for 10 min, separated by SDS–polyacrylamide gel electrophoresis and transferred to polyvinylidene fluoride membrane (Millipore). Membranes were blocked in 5% milk in TBS-T and incubated with indicated antibodies (1 in 1,000 dilution in 5% milk in TBS-T) overnight at 4 °C. Blots were imaged with ECL Prime using a Gel Doc instrument.

## RNA-seq
RNA was extracted using the Bioline Isolate II RNA Mini kit. The RNA concentration was quantified using a Qubit fluorometer (Thermo Fisher Scientific). Libraries were prepared using the 3′Pool-seq method[63]. First-strand cDNA synthesis was performed by first mixing 200 ng of RNA (diluted to 5 µl with water) with 1 µl indexed reverse transcription (RT) primer (10 µM) and 1 µl dNTPs and incubating at 72 °C for 3 min. Then 10 µl of RT master mix (3.6 µl SuperScript 5× buffer, 0.25 µl water, 0.25 µl DTT 100 mM, 2 µl betaine 5 M, 0.9 µl MgCl$_2$ 100 mM, 2.5 µl template switching oligo and 0.5 µl SuperScript II reverse transcriptase) was added to each sample and then incubated at 42 °C for 90 min followed by 10 cycles (50 °C for 2 min, 42 °C for 2 min) and 70 °C for 15 min. Samples with unique RT indexes were then pooled to a total volume of 20 µl for Exonuclease1 treatment by adding 1 µl of Exo1 and incubating at 37 °C for 45 min followed by 92 °C for 15 min. Exo1-treated pools were then cleaned using SPRI Select magnetic beads at a 0.6× ratio according to manufacturer's instructions and eluted in 10 µl. Next, 15 µl of cDNA amplification master mix (12.5 µl KAPA HotStart Mix, 1.25 µl enrichment primer A 20 µM and 1.25 µl enrichment primer B 20 µM) was added to each pool before touch-up PCR: 95 °C for 3 min, 4 cycles (98 °C 20 s, 65 °C 45 s, 72 °C 3 min), 9 cycles (98 °C 20 s, 67 °C 20 s, 72 °C 3 min) and 72 °C for 5 min. Amplified cDNA was cleaned as before but eluted in 20 µl and then diluted to 0.25 ng µl$^{-1}$. This was then subjected to tagmentation and PCR following the manufacturer's instructions with the following changes: 2 µl of diluted cDNA was added to 4 µl of TD buffer followed by 2 µl of ATM buffer. For the amplification, a master mix was prepared (2 µl indexed i5 primer 2 µM, 2 µl enrichment primer A 2 µM and 6 µl NPM) and added to each tagmented pool. PCR reaction was carried out with 13 cycles. The pools were then cleaned as before and then run on a D5000 Tapestation tape for pooling before sequencing on the Illumina NovaSeq 6000.

## RNA-seq analysis
Bcl2fastq (Illumina) was used to perform sample demultiplexing and to convert BCL files generated from the sequencing instrument into Fastq files. Reads were trimmed using bbmap ver38.81 and aligned to GRCh38 for human cell lines (MOLM13, MV4;11, K562 and HUDEP-2) and GRCm38 for murine cell lines (MLLAF9) using STAR ver2.7.9a (–quantMode GeneCounts). The counts file was then plugged into Degust for differential expression analysis. All visualizations were created using ggplot2 in R. P values for box plots were calculated by Wilcoxon or Welch's t tests comparing means using ggpubr::stat_compare_means.

CellRadar plots were made using the CellRadar website (https://karlssong.github.io/cellradar/). Analyses used a count per million (CPM) cut off of 2 in at least 2 samples, significance false discovery rate (FDR) cut off of 0.1 and logFC cut off for induction of 0.5.

## RT–qPCR
Messenger RNA was prepared with a Bioline Isolate II RNA Mini Kit, and complementary DNA synthesis was performed with a LunaScript RT SuperMix Kit (New England BioLabs) per the manufacturers' instructions. qPCR analysis was performed on a Thermo Fisher QuantStudio 6 or Roche LightCycler 480 Real-Time PCR System with SYBR Green reagents. All samples were assayed in triplicate. Relative expression levels were determined with the ΔCt method and normalized to GAPDH or HPRT. RT–qPCR primers are listed in Source Data for Fig. 1.

## Flow cytometry
Cells were washed in PBS/2% FBS, resuspended in PBS with 2% FBS and filtered. Data were acquired on a BD LSR/LSRII LSRFortessa using DiVA V6.3.1 and analysed in FlowJo or sorted using a BD FACSFusion sorter. Data were also analysed using Flowlogic software 7.2.1 (Inivai Technologies).

## Animal studies
All animal experiments were performed with the approval of Dana-Farber Cancer Institute's Institutional Animal Care and Use Committee (IACUC) protocol no. 16-021. For in vivo experiments using CRISPR–Cas9 edited MOLM13 cells, female NOG mice (NOD.Cg-Prkdc$^{scid}$Il2rg$^{tm1Wjl}$/SzJ stock#005557) mice at 6–8 weeks of age were purchased form the Jackson Laboratory and were housed in a pathogen-free (BSL-2) animal facility at the Dana-Farber Cancer Institute under a 12-h light/dark cycle with controlled ambient temperature (20–24 °C) and humidity (40–60%).

## Assessment of in vivo menin inhibitor resistance using CRISPR–Cas9-edited MOLM13 cells
In vivo studies were conducted in NOD.Cg-Prkdc$^{scid}$Il2rg$^{tm1Wjl}$/SzJ (NOG) mice (Jackson Laboratory) aged 7–9 weeks. MOLM13 cells stably expressing the CRISPR–Cas9 enzyme had been engineered to stably express sgRNAs targeting PCGF1 and BCOR (sgPCGF1/sgBCOR), as described above. MOLM-13 CRISPR–Cas9 cells expressing a non-targeting sgRNA (NT sgRNA) were used as a control. Briefly, $5 × 10^4$ MOLM-13 cells were intravenously transplanted into NOG mice who had been sublethally irradiated with 450 Gy. On day 5 following transplantation, mice from each group (NT sgRNA control or sgPCGF1/sgBCOR sgRNA) were randomly assigned to receive either control chow or 0.1% SNDX-5613 spiked-in chow (n = 5 mice per group; 4 groups total). Mice were monitored clinically for signs of illness such as hunched posture or hindlimb paralysis and animals were weighed weekly. Mice were killed if they lost >15% body weight, as dictated by humane end points of our animal protocol. When mice developed signs of leukaemia, the bone marrow, spleen and peripheral blood were collected and analysed using flow cytometry to evaluate human CD45$^+$ cells (anti-human CD45, APC-Cy7 conjugated BioLegend, 304014) and the differentiation marker CD11b (anti-mouse/anti-human CD11b, APC conjugated, 301310).

Sample sizes were chosen based on power calculations using G*Power 3.1. Timepoints and expected effect size were based on previously published work on MOLM-13 xenografts[64]. At a predefined cut-point of 20 days, there is 84% power to distinguish survival of 10% in the control arm (control vehicle) versus 80% in the three treatment arms (control SNDX 0.1%, PCGF1/BCOR vehicle and PCGF1/BCOR SNDX 0.1%) with an alpha error probability of 0.017 using one-sided Fisher's exact test with Bonferroni correction accounting for three tests using G*Power 3.1. No mice were excluded from the analysis and no data were

censored. Following engraftment of human leukaemia xenografts, mice were randomized into treatment cohorts to receive control chow or 0.1% SNDX-5613 chow to ensure an equal distribution of weights within each treatment cohort. Standardized clinical grading criteria and objective data such as weight loss was used to assess clinical illness in compliance with study end points per the IACUC protocol no. 16-021. Blinding was not used.

## HuDEP-2 erythroid differentiation

HuDEP-2 cells were plated in duplicate at a consistent density in expansion medium supplemented with dexamethasone (1 μM), doxycycline (1 μg ml⁻¹), human stem cell factor (SCF) (50 ng ml⁻¹) and erythropoietin (EPO) (3 U ml⁻¹). Cells were pretreated with DMSO or DOT1Li 5 μM (SGC0946) for 7 days before being subjected to a 7-day erythroid differentiation protocol. Cells were differentiated in IMDM containing human AB serum (3%), FBS (2%), glutamax (1×), ITS-G (1×), heparin (3 U ml⁻¹), holo-transferrin (330 μg ml⁻¹), EPO (3 U ml⁻¹), doxycycline (1 μg ml⁻¹), SCF (10 ng ml⁻¹) and IL-3 (1 ng ml⁻¹). On day 5 of maturation, doxycycline was removed to promote enucleation. To monitor erythroid differentiation, flow cytometry was performed on days 0, 3, 4, 5, 6 and 7 by staining for the early erythroid progenitor marker CD49d (integrin α4) and late erythroid marker CD235a (glycophorin A) as well as propidium iodide for the determination of cell viability.

## Protein expression and purification

For the expression of PRC1.1, maltose binding protein (MBP) N-terminally tagged PCGF1 (MBP-PCGF1) and Strep N-terminally tagged RING1B (Strep-RING1B) were cloned into the same multigene expression vector with a pFastBac1-derived backbone using the MoClo Baculo toolkit (AddGene kit no. 1000000256)[65]. For the expression of PRC1.4, maltose binding protein (MBP) N-terminally tagged PCGF4 (MBP-PCGF4) and Strep N-terminally tagged RING1B (Strep-RING1B) were cloned into a second multigene vector using the same strategy. Baculovirus stocks were generated using the FuGENE HD Transfection Reagent (Promega, E2311) according to the instructions of the manufacturer, and baculovirus amplification was carried out similarly to previous descriptions[66].

Infected Hi5 cells were collected by centrifugation at 1,500*g* and flash-frozen in liquid nitrogen as cell pellets. For purification, the pellets were lysed on ice in TBSL buffer (50 mM Tris pH 8 at 25 °C, 300 mM NaCl, 15% glycerol, NP-40 Alternative (Millipore, 492016), 1 mM TCEP, 1 mM PMSF (Sigma), 80 μM Aprotinin (Abcam, AB146286), 4 μM Bestatin (Sigma, 200484), 1.4 μM E64 (Abcam, AB141418), 2.1 μM Leupeptin hemisulfate (Sigma, L2884) and 1.5 μM Pepstatin A (Millipore, 516481)). The lysate was clarified by centrifugation at 30,000*g* for 30 min using an F14-6x250y rotor. The supernatant was incubated with amylose resin (NEB, E8021) for 40 min at 4 °C with gentle rotation. The resin was batch-washed sequentially with 20 bead volumes of TBSL lacking protease inhibitors, 20 bead volumes of TBS500 (50 mM Tris pH 8 at 25 °C and 500 mM NaCl) and 10 bead volumes of TBS150 (50 mM Tris pH 8 at 25 °C and 150 mM NaCl). Bound proteins were eluted in four bead volumes of TBS150 supplemented with 1 mM TCEP and 10 mM maltose. Heparin affinity purification using a HiTrap Heparin HP affinity column (Cytiva, 17040703) was performed as described previously[65] using TBS150 as buffer A and TBS2000 (50 mM Tris pH 8 at 25 °C and 2 M NaCl) as buffer B. Fractions containing the target protein were pooled and concentrated.

PRC1.1 and PRC1.4 were further purified by Superdex 200 Increase 10/30 or HiLoad Superdex 16/600 (Cytiva), respectively, in 20 mM HEPES−NaOH pH 7.5 and 150 mM NaCl. Desired fractions were pooled and supplemented with 1 mM TCEP and then frozen in liquid nitrogen as single-use aliquots. Human UBA1, UbcH5c and ubiquitin were purified as described previously[67]. Constructs made for this study are available upon request.

## Production of MLA histones and nucleosome reconstitution

Expression and purification of human histones and their mutants for MLA were performed as described previously[66]. MLA alkylation of histone H3.1 that included the point mutations C96S, C110A and K79C was done as in the protocol published by Simon et al.[68,69]. After the reaction, both H3K79me2 and H3K79me3 MLA histones were dialysed in a >1,000-fold reaction volume of MilliQ water with 1 mM DTT. Then MLA histones were loaded on a ZORBAX 300SB-C3 5 μm, 9.4 × 250 mm semi-preparative HPLC column (Agilent, 880995-209) on a Shimadzu HPLC system. Alkylated and unreacted products were separated over a 55 min gradient, from 10% to 70% B, using 0.1% formic acid in acetonitrile as the organic phase (B) and 0.1% formic acid in MilliQ water as the aqueous phase (A). Purified histones were frozen at −80 °C and lyophilized overnight. For quality assurance, peak fractions from the MLA histone preparations were analysed by liquid chromatography–mass spectrometry using the same gradient over a ZORBAX RRHD 300SB-C3 2.1 × 100 mm 1.8 μm analytical column in line with an Impact II mass spectrometer (Bruker Daltonics). Reconstitution of H3K79me2- and H3K79me3-modified MLA or non-MLA nucleosomes was performed using the gradient dialysis method, as described previously[66], and was stored at 4–8 °C until use.

## In vitro ubiquitination assay

The ubiquitination assay was performed using purified UBA1 as the E1 (ref. [69]), UbcH5c as the E2, ubiquitin, PRC1.1 or PRC1.4, and reconstituted nucleosomes as indicated. Before starting the reaction, 3× E1/2-ub mix was prepared as 300 nM UBA1, 1.5 μM UbcH5c, 150 μM ubiquitin and 9 mM ATP in assay buffer containing 20 mM HEPES−NaOH pH 7.5, 150 mM NaCl, 3 mM MgCl₂ and 1 mM DTT. The mixture was incubated at 37 °C for 30 min to allow loading of ubiquitin onto E1 and E2 and was then kept on ice until use. PRC1 enzyme−either PRC1.1 or PRC1.4−was diluted to 3× its indicated assay concentration in dilution buffer containing 20 mM HEPES−NaOH pH 7.5, 150 mM NaCl, 6 mM MgCl₂, 1 mM mDTT and 0.2 mg ml⁻¹ BSA. For preparing the 3× nucleosome solution, nucleosomes were dialysed in 20 mM Tris pH 7.5, 2.5 mM KCl and 1 mM DTT, and were then diluted to 3× their indicated assay concentration in 20 mM HEPES−NaOH pH 7.5, 1 mM DTT. To initiate the histone ubiquitination reaction, equal volumes of the 3× E1/2-ub and 3× nucleosome solutions were mixed first, and the reaction was started by the addition of another equal volume of the 3× PRC1 enzyme solution. By doing so, the NaCl concentration in the final reaction is close to 100 mM. For substrate titrations, the reaction was incubated at 30 °C for 30 min. For progress curves, the reaction stopped after 10, 30 or 90 min. The reactions were stopped by cooling on a prechilled metal block on ice for 3 min and the addition of 0.39 reaction volume of 3.6× NuPAGE LDS Sample Buffer (Thermo Fisher, NP0008) with 40 mM EDTA and 1% beta-mercaptoethanol. The reference sample was prepared by incubating 1.1 μM of wild-type nucleosomes, 10 nM PRC1.1 or PRC1.4, 100 nM UBA1, 0.5 μM UbcH5c, 50 μM ubiquitin and 3 mM ATP following the same method described above. The reaction was incubated at 30 °C for 40 min before being stopped with NuPAGE LDS Sample Buffer (Thermo Fisher, NP0008). Samples were stored at −20 °C before being loaded on SDS−PAGE for immunoblotting.

For immunoblotting, Novex Tris-Glycine Mini Protein Gels, 8–16%, 1.0 mm, WedgeWell format gels (Thermo, XP08162BOX) were used, and gels were run for 30–35 min at 160 V in MES running buffer (Thermo, NP0002) and were then transferred using the iBlot3 transfer system (Thermo) in an iBlot 3 Transfer Stack, mini, nitrocellulose (Thermo) using a one-step 5 min program at 25 V with medium cooling. Alternatively, Novex Tris-Glycine Midi Protein Gels, 4–20% were used, and gels were run for 45 min at 160 V in MES running buffer (Thermo, NP0002). The transfer was done similarly, with the exception that a one-step 6.5 min program at 25 V with medium cooling was used. Then 18 μl of sample was used for immunoblotting using anti-H2AK119ub antibodies and 4 μl of sample was used for immunoblotting using anti-MBP

# Article

and anti-H3 antibodies. Next 18 µl of normalization control (NC) was loaded on each gel for blotting using anti-H2AK119ub antibodies. For the progress curves of PRC1.1, Ponceau stain was carried out on a gel with 18 µl loading while the MBP and H2AK119ub blots were gel with 4 µl loading. While H2AK119ub blots for PRC1.4 were performed on gels with 18 µl loading.

For PRC1.1 activity assays, H2AK119ub blots were blocked in StartingBlock (Thermo, 37538) for 1 h at room temperature and incubated with H2AK119ub (CST, D27C4) antibody at 1:1,000 for 1 h at room temperature followed by 1 h incubation in 1:5,000 HRP-conjugated goat anti-rabbit (Abcam, A0545) antibodies at room temperature. For PRC1.4 activity assays, H2AK119ub blots were carried out the same as for PRC1.1, except that the anti-H2AK119ub antibodies were diluted 1:500 and incubated with the blots overnight at 4 °C.

For immunoblotting using anti-H3 and anti-MBP antibodies, 4 µl of sample or NC were loaded for immunoblotting, and electrophoresis and transfer were carried out as above. The anti-H3 and anti-MBP blots resulting from a PRC1.1 activity assay were blocked for 1 h at room temperature, while the corresponding blots from a PRC1.4 activity assay were blocked overnight in StartingBlock (Thermo, 37538). Rabbit anti-H3 (1:50,000; Abcam, ab1791) antibody and mouse anti-MBP (1:5,000; NEB, E8032L) antibodies were used as primary antibodies. For the H3 blots, the secondary antibody was HRP-conjugated goat anti-rabbit antibodies (1:5,000; Abcam, A0545). For the MBP blots, HRP-conjugated donkey anti-mouse antibodies (1:5,000; Jackson Immuno Research, 715-035-150) were used as secondary antibodies. Both the anti-H3 and anti-MBP primary antibodies and the secondary antibodies were incubated with the blots for 1 h at room temperature.

All antibodies were diluted in StartingBlock (Thermo, 37538) and were not reused. The blots were washed five times in Tris-buffered saline with Tween-20 between primary and secondary antibody incubation, and before imaging by chemiluminescence substrate (Thermo, 34580) using a ChemiDoc (Bio-Rad). Densitometry for H2AK119ub was carried out using Image Lab (Bio-Rad).

## Statistics and reproducibility

Statistical analysis was carried out using GraphPad Prism 9 and RStudio::ggpubr. Details of statistical analysis performed and significance values are provided in the figure legends. Non-significant data were not annotated. Data were reported as mean ± s.d., and independent replicates shown as individual data points. Proliferation assays were reproduced at least three times. No statistical method was used to predetermine sample size, no data were excluded from the analyses for in vitro experiments. For in vivo experiments, sample sizes were chosen based on power calculations using G*Power 3.1. Timepoints and expected effect size were based on previously published work on MOLM13 xenografts[64]. At a predefined cut-point of 20 days, there is 84% power to distinguish survival of 10% in the control arm (control vehicle) versus 80% in the three treatment arms (control SNDX 0.1%, PCGF1/BCOR vehicle or PCGF1/BCOR SNDX 0.1%) with α error probability of 0.017 using one-sided Fisher's exact test with Bonferroni correction accounting for three tests using G*Power 3.1. No mice were excluded from the analysis and no data were censored. Following engraftment of human leukaemia xenografts, mice were randomized into treatment cohorts to receive control chow or 0.1% SNDX-5613 chow to ensure an equal distribution of weights within each treatment cohort. Standardized clinical grading criteria and objective data such as weight loss were used to assess clinical illness in compliance with study end points per the IACUC protocol no. 16-021. Blinding was not used.

## Reporting summary

Further information on research design is available in the Nature Portfolio Reporting Summary linked to this article.

## Data availability

The raw and processed RNA-seq and ChIP–seq data supporting Figs. 2–7 has been deposited in the NCBI Gene Expression Omnibus (GEO) and are accessible via accession numbers GSE260456 and GSE260742, respectively. Published datasets used in this article include ChIP–seq for MLL1 in K562 cells treated with Menin inhibitor, which is accessible from GSE181829. Source data are provided with this paper.

## Code availability

This study did not generate original code or algorithms. All software tools used are publicly available online. Unless otherwise specified in the Methods, all data analyses were performed using default parameters. The specific scripts used for analysis of the data reported in this Article are available upon request.

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

## Acknowledgements

We thank D. Steer from the Monash Proteomics and Metabolomics Platform and the ARAFlowcore Flow facility and molecular genomics facility at the Australian Centre for Blood Diseases, Monash University. We thank the following funders for fellowship, scholarship and grant support: VCA Mid-Career Research Fellowship MCRF19040 (to O.G.); NHMRC Project grant nos. 1146192 and 2021063 (to O.G.); Cancer Council Victoria Grants-in-aid (to O.G.).; NHMRC Investigator grant no. 2027164 (to O.G.). This work is supported by a National Cancer Institute (NCI) Cancer Center Support Grant P30 CA06516 (to S.A.A. and E.B.H.), to the Dana-Farber/Harvard Cancer Center and The V Foundation for Cancer Research (to S.A.A. and E.B.H.). E.B.H. is supported by grants from NCI K08 CA279891, Hyundai Hope on Wheels, the Children's Cancer Research Fund, the Charles H. Hood Foundation, the American Society of Hematology and CURE Childhood Cancer. S.A.A. was supported by grants from the NIH (CA066996) and St. Jude Children's Research Hospital. C.D. was supported by National Health and Medical Research Council (NHMRC) grant nos. APP1162921, APP1184637, APP2011767 and APP 2020900. The funders had no role in the study design, data collection and analysis, decision to publish or preparation of the manuscript.

## Author contributions

O.G. and C.D. designed and supervised the research and wrote the manuscript. D.N. and D.T.F. designed the research, conducted experiments, analysed data and helped write the manuscript. E.B.H. and Z.L. contributed equally. E.B.H. designed and conducted in vivo animal experiments and analysed data. Z.L. designed, conducted and analysed the in vitro biochemical ubiquitination assays. G.W.M. performed bioinformatic analysis. C.L., O.G.D., V.L., K.K., J.J.T., S.A., S.C.S. and B.R. conducted experiments and analysed data. L.T. and E.Y.N.L. provided bioinformatic support. A.C.P., S.A.A. and M.U. provided expertise and/or reagents. C.C.B. conducted experiments, provided critical expertise in data interpretation and helped write the manuscript. All authors reviewed and edited the manuscript.

## Competing interests

S.A.A. has been a consultant and/or shareholder for Neomorph, Cyteir Therapeutics, C4 Therapeutics, Accent Therapeutics, Hyku Therapeutics and Stelexis Therapeutics. S.A.A. has received research support from Janssen, Novartis and Syndax. S.A.A. is an inventor on a patent related to MENIN inhibition WO/2017/132398A1. The other authors declare no competing financial interests.

## Additional information

**Extended data** is available for this paper at https://doi.org/10.1038/s41556-025-01859-8.

**Correspondence and requests for materials** should be addressed to Chen Davidovich or Omer Gilan.

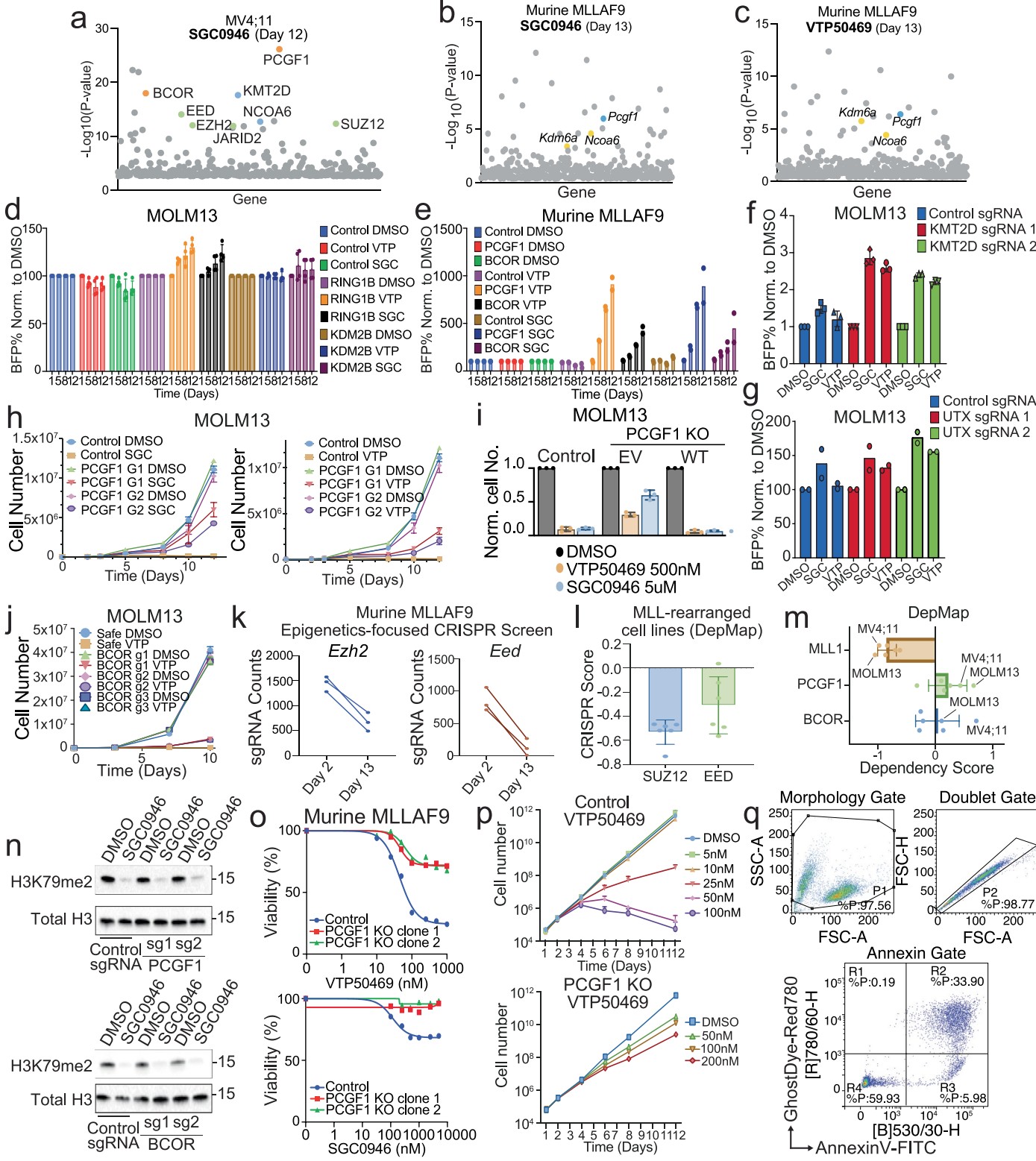

**Extended Data Fig. 1 | See next page for caption.**

**Extended Data Fig. 1 | Depletion of the PRC1.1 complex components, BCOR and PCGF1, confers resistance to Menin and DOT1L inhibition in MLL-leukaemia cells. a-c**. Bubble plots of -Log10(p-value) for: (**a**) MV4;11 SGC0946 genome-wide CRISPR screen at day 12, (**b**) murine MLLAF9 SGC0946 epigenetic-focused screen and (**c**) murine MLLAF9 VTP50469 epigenetic-focused screen at day 13. P-values calculated using the MAGECK algorithm and adjusted for multiple testing. **d**. Competition assay in MOLM13 control, RING1B, or KDM2B sgRNA cells treated with SGC0946 or VTP50469. Data represents mean ± SD from n = 4 independent experiments. **e**. Competition assay in Murine MLLAF9 control, BCOR, or PCGF1 sgRNA cells treated with SGC0946 or VTP50469. Data represents mean ± SD from n = 3 independent experiments. **f-g**. Competition assay in MOLM13 control, KMT2D sgRNA1/2 cells (**f**, data represents mean ± SD from n = 3 independent experiments), or UTX sgRNA1/2 (**g**, data from 2 independent experiments) treated with SGC0946 or VTP50469 for 7 days. **h**. Proliferation of MOLM13 control or PCGF1 sgRNA1/2 cells treated with SGC0946 or VTP50469. Data represents mean ± SD from n = 3 independent experiments. **i**. Proliferation of control or PCGF1 KO MOLM13 with empty vector (EV) or wildtype PCGF1 (WT) treated with SGC0946 or VTP50469. Data represents mean ± SD from n = 3 independent replicates. **j**. Proliferation of MOLM13 control or BCOR sgRNA1/2 cells treated with VTP50469. Data represents mean ± SD from n = 3 independent experiments. **k**. *Ezh2* and *Eed* sgRNA counts at day 2 and 13 from (c). **l**. DepMap scores for *Ezh2* and *Eed*. Data represents mean ± SD from n = 6 MLL-rearranged cell lines. **m**. DepMap gene effect scores for MLL1, PCGF1, and BCOR. Data represents mean ± SD from n = 6 MLL-rearranged cell lines. **n**. Western blot of H3K79me2 and histone H3 in MOLM13 control, PCGF1 sgRNA1/2, and BCOR sgRNA1/2 cells treated with SGC0946 for 4 days. **o**. Dose response assay in murine MLLAF9 control or PCGF1 sgRNA1/2 cells treated with SGC0946 or VTP50469. **p**. Proliferation assay in control or PCGF1 KO murine MLLAF9 cells treated with VTP50469. Data represents mean ± SD from n = 3 independent experiments. **q**. Representative Annexin V gating strategy. Competition assay used BFP+ sgRNA cells with data as BFP + % normalised to DMSO. Source data available for **d-f**, **h-o**, and **p**.

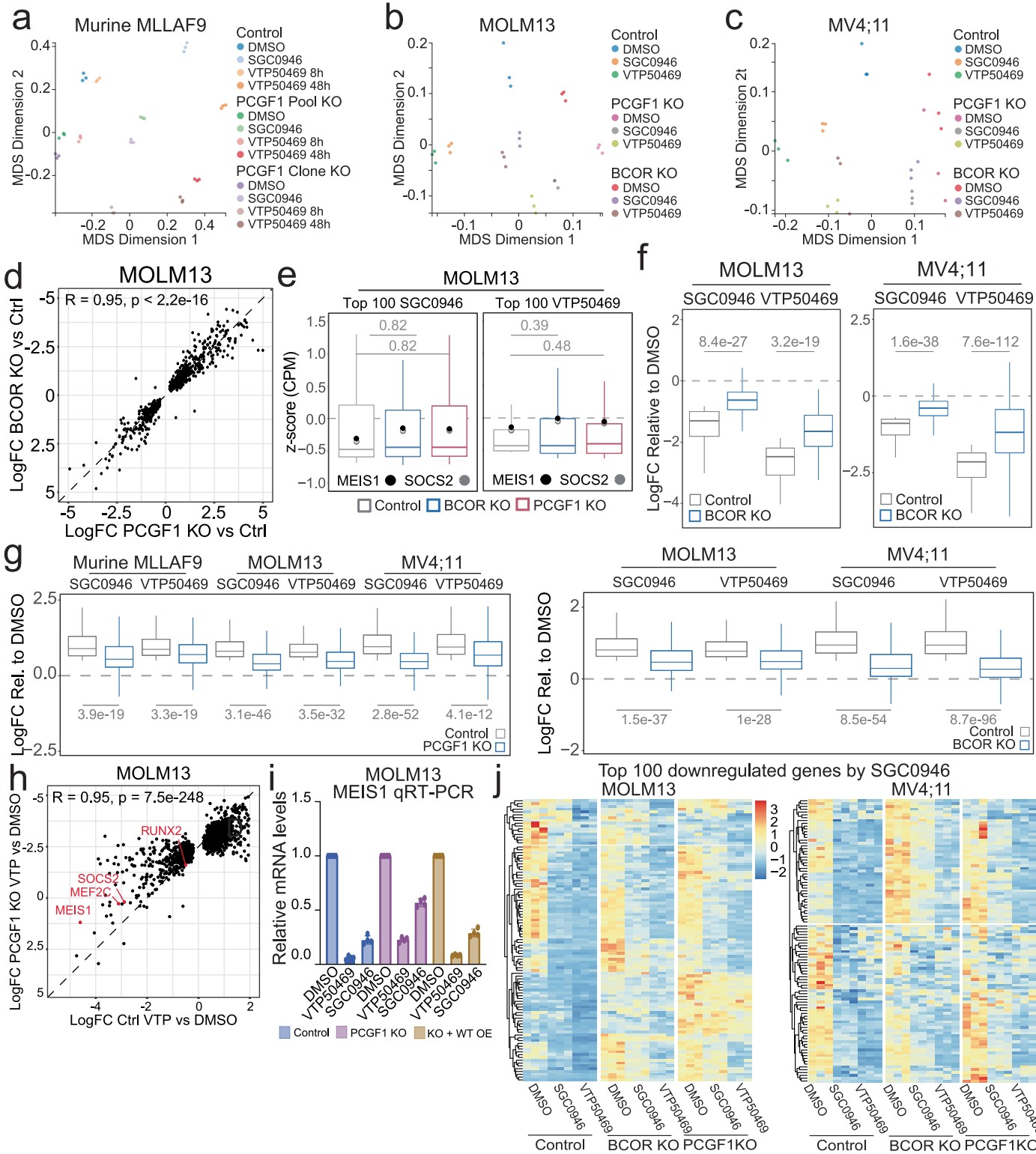

**Extended Data Fig. 2 | See next page for caption.**

**Extended Data Fig. 2 | PRC1.1 depletion blunts the transcriptional response to Menin and DOT1L inhibition. a-c**. MDS plots of RNA-seq in (**a**) murine MLLAF9 (**b**) MOLM13 and (**c**) MV4;11 cells. **d**. Correlation plot of the PCGF1 KO or BCOR KO relative to MOLM13 control cells. **e**. Box plots of the baseline expression of the top 100 SGC0946- or VTP50469-downregulated genes in MOLM13. Data presented as z-score CPM. *MEIS1* represented by a black dot and *SOCS2* by a grey dot. P = 0.82 and p = 0.82 for SGC0946 and P = 0.39 and p = 0.48 for VTP50469 (control vs BCOR or PCGF1 KO). **f**. Box plots of downregulated genes in control MOLM13 and MV4;11 treated with SGC0946 or VTP50469 relative to control. P = 8.4e-27 and p = 3.2e-19 for MOLM13 and p = 1.6e-38 and p = 7.6e-112 for MV4;11 cells treated with SGC0946 and VTP50469, respectively (BCOR KO vs control). **g. (**Left) Box plots equivalent to Fig. 2d for the genes upregulated by SGC0946 or VTP50469 treatment in control and PCGF1 KO. P = 3.9e-19, p = 3.3e-19, p = 3.1e-46, p = 3.5e-32, p = 2.8e-52, and p = 4.1e-12 for murine MLLAF9, MOLM13, and MV4;11 treated with SGC0946 and VTP50469, respectively (PCGF1 KO vs control). (Right)

Box and whiskers plot equivalent to (**f**) for the genes upregulated by SGC0946 or VTP50469 treatment. P = 1.5e-37, p = 1e-28, p = 8.5e-54, and p = 8.7e-96 for MOLM13 and MV4;11 treated with SGC0946 and VTP50469, respectively (BCOR KO vs control). **h**. Correlation plot of VTP50469 treatment relative to DMSO in MOLM13 control vs PCGF1 KO cells (R = 0.95 p = 7.5e-248). **i**. qRT-PCR analysis of *MEIS1* in control, PCGF1 KO, or PCGF1 KO expressing WT PCGF1 MOLM13 cells treated with DMSO, VTP50469 or SGC0946. *GAPDH* used as control. Data represents mean ± SD from n = 4 independent experiments. **j**. Heatmap of SGC0946-treated MOLM13 and MV4;11. Data presented as z-score normalised CPM. Correlation coefficients were calculated using the Pearson method. For all box plots centre line represents median, limits represent upper and lower quartiles, whiskers represent the maxima and minima if not exceeding 1.5*IQR. Box plot p-values were calculated using two-sided Welch's t-tests with no adjustment for multiple comparisons. RNA-seq data from 3 biological replicates. Source data available for **i**.

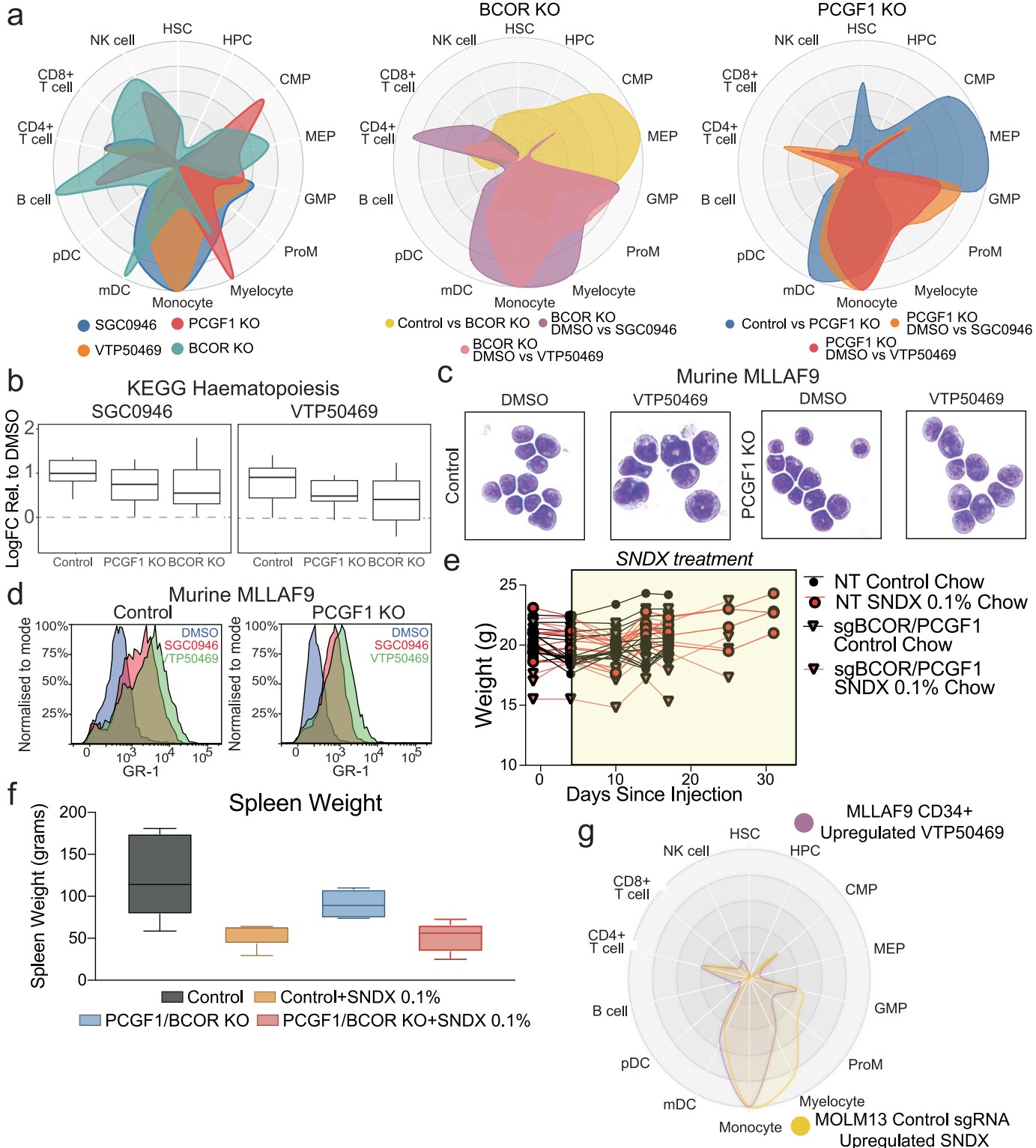

**Extended Data Fig. 3 | See next page for caption.**

**Extended Data Fig. 3 | PRC1.1 is required for efficient DOT1L and Menin inhibitor-induced differentiation in vitro and in vivo. a**. (Left) CellRadar plot of the top 100 upregulated genes (FDR < 0.05) after SGC0946 or VTP50469 in control cells (orange and blue, respectively) and upregulated genes following PCGF1 KO and BCOR KO relative to control MOLM13 cells (red and cyan, respectively). (Middle) CellRadar plot of the top 100 upregulated genes in BCOR KO cells relative to control (yellow), SGC0946 treatment in BCOR KO cells (purple), or VTP50469 treatment in BCOR KO cells (pink). (Right) Same as in middle but for PCGF1 KO cells. **b**. Box plots of genes upregulated during differentiation (KEGG haematopoiesis) following SGC0946 (left) or VTP50469 (right) in control, PCGF1 KO, or BCOR KO MOLM13 cells. Data from 3 biological replicates. **c**. Giemsa staining of murine MLLAF9 control or PCGF1 KO cells treated with either DMSO or VTP50469 (500 nM) for 3 days. **d**. MFI of GR-1 in murine MLLAF9 control and PCGF1 KO cells treated with DMSO, SGC0946, or VTP50469. **e**. Weights of mice transplanted with MOLM13 control cells fed control (n = 8) or SNDX 0.1% chow (n = 8) or PCGF1 BCOR KO cells fed control (n = 8) or SNDX 0.1% chow (n = 9). **f**. Box plot of the spleen weights measured at the endpoint. Mice transplanted with MOLM13 control cells fed control (n = 5 mice) or SNDX 0.1% chow (n = 5 mice) or PCGF1 BCOR KO cells fed control (n = 4 mice) or SNDX 0.1% chow (n = 5 mice). **g**. CellRadar plot showing the overlap of the upregulated genes in mice engrafted control MOLM13 (red) and murine MLLAF9 (brown) cells treated with Menin inhibitor. For all box plots centre line represents median, limits represent upper and lower quartiles, whiskers represent the maxima and minima if not exceeding 1.5*IQR. Source data available for **e** and **f**.

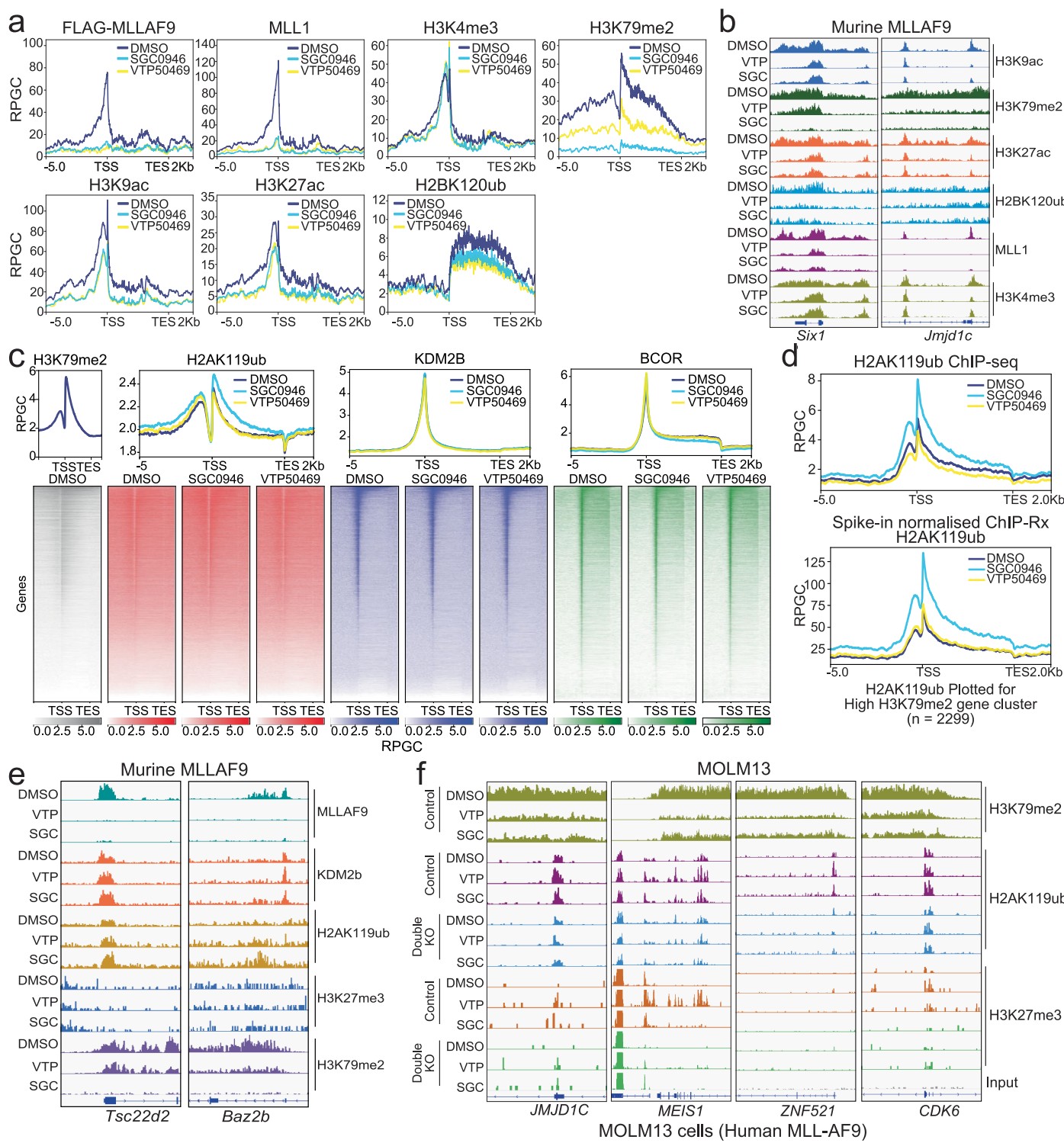

**Extended Data Fig. 4 | Menin and DOT1L inhibition induce repressive chromatin modifications at MLL-FP target genes. a.** Average profile plots of FLAG-MLLAF9, MLL1, H3K4me3, H3K9ac, H3K27ac, H2BK120ub, and H3K79me2 in murine MLLAF9 cells treated with DMSO, SGC0946, and VTP50469 for the region from 5 kb upstream of the TSS to 2 kb downstream of the TES. Genes plotted (n = 38) are those which are significantly downregulated (LogFC < −0.5 and FDR < 0.1) by both SGC0946 and VTP50469 and strongly bound by MLLAF9 (in the top 100 most highly bound genes by FLAG-MLLAF9 ChIPseq) as in main Fig. 4b, e. **b.** Genome browser snapshots of *Six1* and *Jmjd1c* in murine MLLAF9 cells with the indicated samples and treatments. **c.** Heatmap and average profile plots of H3K79me2, H2AK119ub, KDM2B, and BCOR ChIP-seq data for all genes

in murine MLLAF9 cells treated with DMSO or SGC0946 (5 μM for 48hrs) or VTP50469 (500 nM for 48hrs). Genes are ranked from highest to lowest by H3K79me2 occupancy in DMSO. **d.** Average profile plots of H2AK119ub in murine MLLAF9 cells treated with DMSO, SGC0946 (5 μM for 48hrs), or VTP50469 (500 nM for 48hrs) in a ChIP-Rx ChIP-seq experiment. Drosophila S2 chromatin was used as a spike-in control to normalise ChIP-seq data. Unnormalised (top) and normalised (bottom) average profile plots are shown. **e.** Genome browser snapshots of *Tsc22d2* and *Baz2b* in murine MLLAF9 cells with the indicated samples and treatments. **f.** Genome browser snapshot of *JMJD1C*, *MEIS1*, *ZNF521* and *CDK6* in control and PCGF1/BCOR KO (Double KO) MOLM13 cells with the indicated samples and treatments.

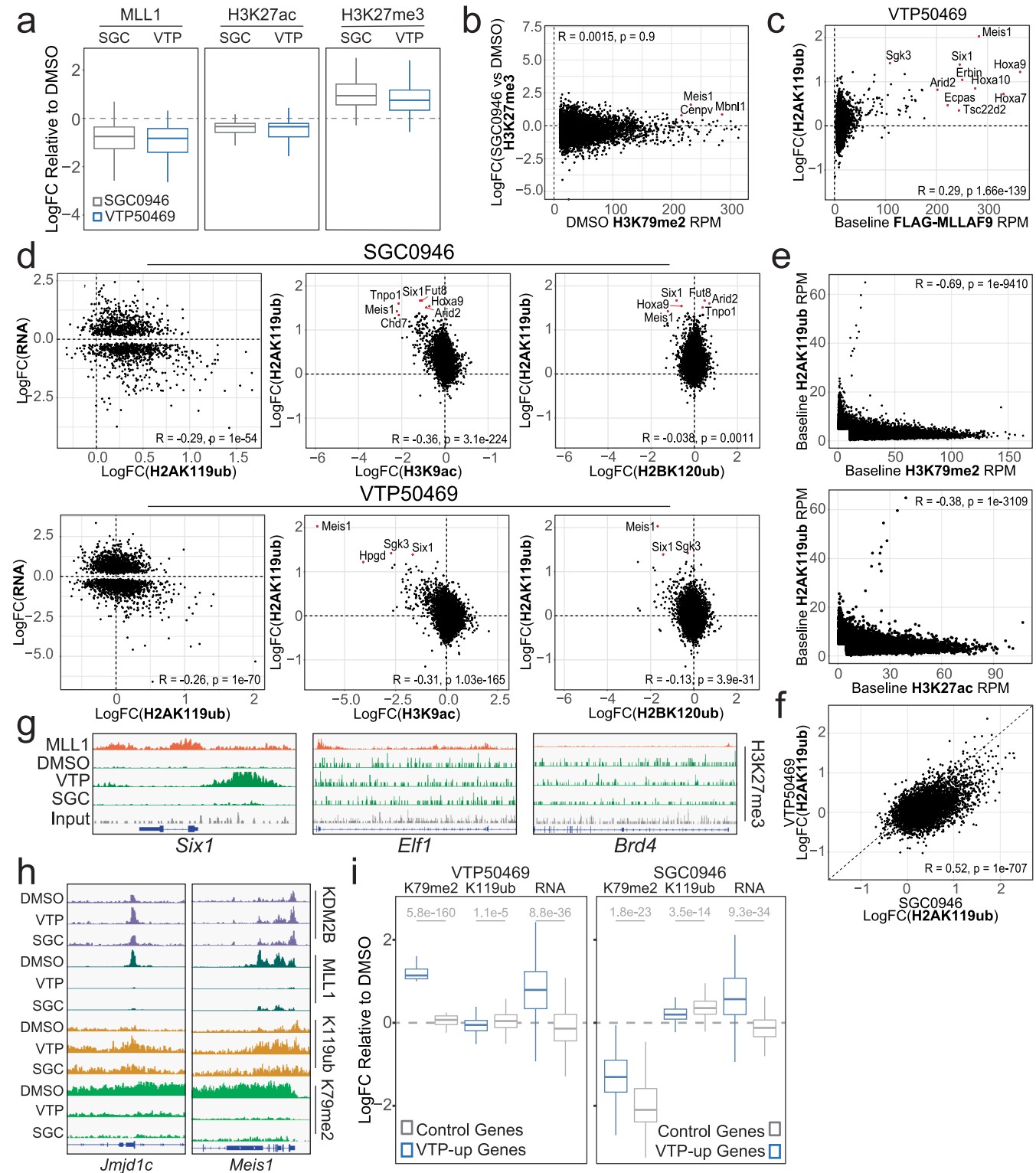

**Extended Data Fig. 5 | See next page for caption.**

**Extended Data Fig. 5 | H2AK119ub induction corresponds to the loss of H3K79me2 in human and mouse MLL-leukaemia. a.** Box plots of the LogFC of MLL1, H3K27ac, and H3K27me3 in SGC0946 or VTP50469 treatment relative to DMSO for commonly downregulated genes by SGC0946 (grey) and VTP50469 (blue, n = 176 genes). **b.** Global correlation plot of the baseline H3K79me2 RPM vs the LogFC of H3K27me3 (R = 0.0015 p = 0.9) in SGC0946 treatment. **c.** Correlation plots of LogFC of H2AK119ub after VTP50469 treatment vs baseline FLAG-MLLAF9 RPM (R = 0.29 p = 1.66e-139). **d.** Correlation plots of the LogFC of SGC0946 (top) or VTP50469 (bottom) treatment relative to DMSO for either: RNA vs H2AK119ub (R = −0.29 p = 1e-54 or R = −0.26 p = 1e-70), H2AK119ub vs H3K9ac (R = −0.36 p = 3.1e-224 or R = −0.31 p = 1.03e-165), or H2AK119ub vs H2BK120ub (R = −0.038 p = 1.1e-2 or R = −0.13 p = 3.9e-31). **e.** Correlation plots of the baseline H2AK119ub RPM (y-axis) against the baseline H3K79me2 RPM (top, R = −0.69 p = 1e-9410) or H3K27ac RPM (bottom, R = −0.38 p = 1e-3109) for 5 kb bins. **f.** Correlation plot of the LogFC of VTP50469 treatment vs the LogFC of SGC0946 treatment for H2AK119ub (R = 0.52 p = 1e-707). **g.** Genome browser snapshots showing H3K27me3 for the same genes as Fig. 5g. **h.** Genome browser snapshots of *Meis1* and *Jmjd1c* with the same ChIP-seq tracks as in Fig. 5g. **i.** Box plots of the LogFC of H3K79me2, H2AK119ub, and RNA in SGC0946 (grey) or VTP50469 (blue) treatment. LogFC for H3K79me2 and H2AK119ub calculated using the TSS ± 5 kb. P = 5.8e-160, p = 1.1e-5, p = 8.8e-36 for H3K79me2, H2AK119ub, and RNA, respectively, in control vs VTP-up genes after VTP50469 treatment. P = 1.8e-23, p = 3.5e-14, p = 9.3e-34 for H3K79me2, H2AK119ub, and RNA, respectively, in control vs VTP-up genes after SGC0946 treatment. P-values were calculated using a two-sided Welch's t-test with adjustment for multiple comparisons using the Holm method. For all box plots centre line represents median, limits represent upper and lower quartiles, whiskers represent the maxima and minima if not exceeding 1.5*IQR. All correlation coefficients were calculated using the Pearson method with no adjustment for multiple comparisons. ChIP-seq data from a representative replicate of 2 biological replicates and RNA-seq data from 3 biological replicates.

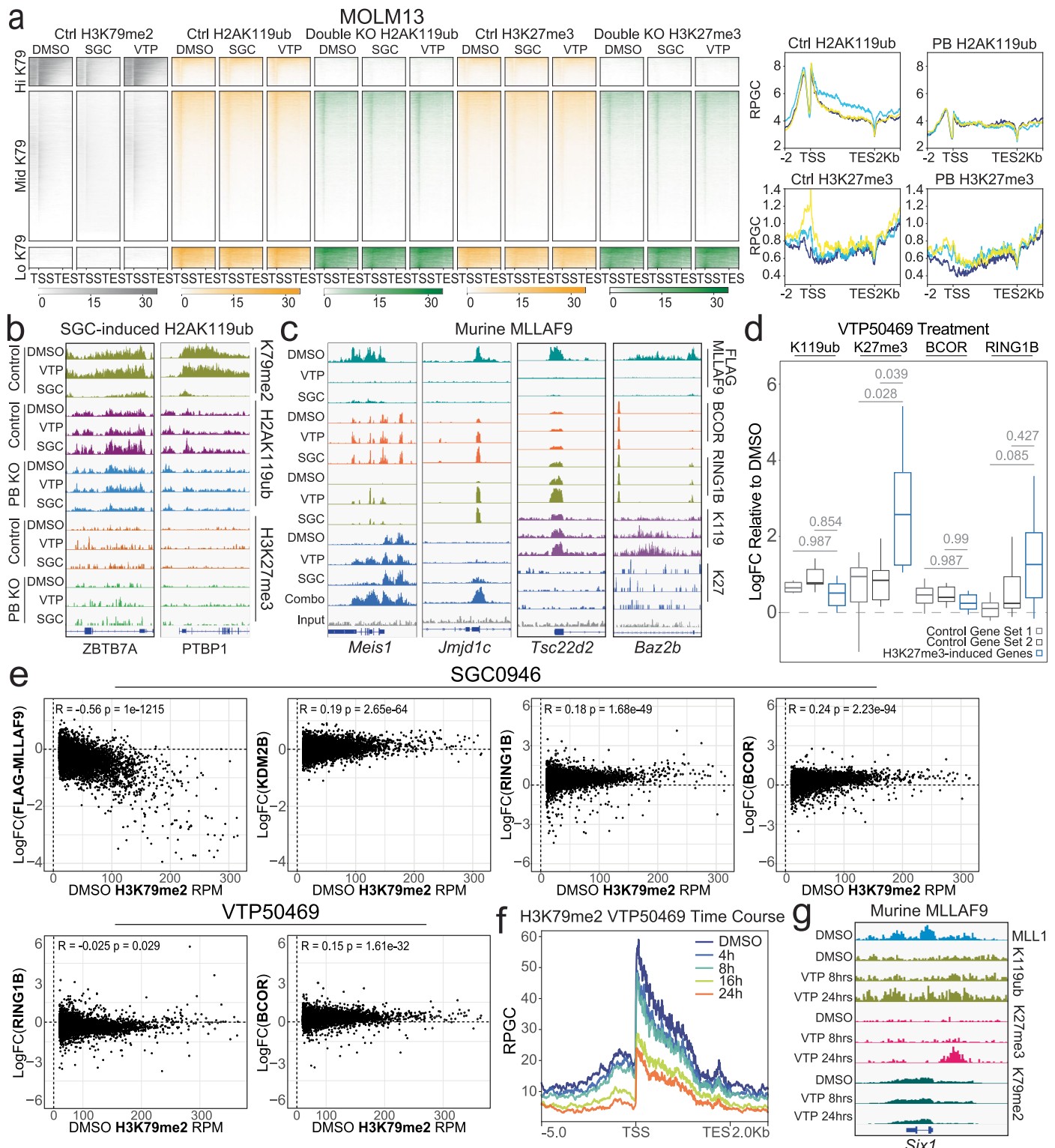

**Extended Data Fig. 6 | See next page for caption.**

**Extended Data Fig. 6 | H3K79me2 protects genes from PRC1.1-induced H2AK!19ub to control the kinetics of gene repression following Menin inhibition. a**. (Left) Global heatmap plots of H3K79me2, H2AK119ub, and H3K27me3 in control and/or PCGF1/BCOR KO MOLM13 cells treated with DMSO, SGC0946, or VTP50469. Genes were sorted into high, medium, and low H3K79me2 levels using k-means clustering based on control cells in DMSO. (Right) Average profile plots of the high H3K79me2 cluster for H2AK119ub and H3K27me3 in control and PCGF1/BCOR (PB) KO cells treated with DMSO, SGC0946, or VTP50469. **b-c**. Genome browser snapshots of *ZBTB7A* and *PTBP1* in MOLM13 cells (b) or *Meis1*, *Jmjd1c*, *Tsc22d2,* and *Baz2b* in murine MLLAF9 cells (c) with the indicated samples and treatments. **d.** Box plots of H2AK119ub, H3K27me3, BCOR, and RING1B ChIP-seq data in murine MLLAF9 cells treated with DMSO or VTP50469. Data presented as the LogFC of treatment relative to DMSO for n = 11 H3K27me3-induced or 2 subsets of n = 11 control genes that see H2AK119ub induction. P = 0.987, p = 0.854 for control 1 vs H3K27me3-induced and control 2 vs H3K27me3-induced genes for H2AK119ub, respectively. P = 0.028, p = 0.039 for control 1 vs H3K27me3-induced and control 2 vs H3K27me3-induced genes for H3K27me3, respectively. P = 0.987, p = 0.99 for

control 1 vs H3K27me3-induced and control 2 vs H3K27me3-induced genes for BCOR, respectively. P = 0.085, p = 0.427 for control 1 vs H3K27me3-induced and control 2 vs H3K27me3-induced genes for RING1B, respectively. P-values were calculated using a two-sided Welch's t-test with adjustment for multiple comparisons using the Holm method. Data from 1 replicate. **e.** (Top) Correlation plots of the LogFC of FLAG-MLLAF9 (R = −0.56 p = 1e-1215), KDM2B (R = 0.19 p = 2.65e-64), RING1B (R = 0.18 p = 1.68e-49), and BCOR (R = 0.24 p = 2.23e-94) in SGC0946 treatment relative to DMSO vs baseline H3K79me2 RPM. (Bottom) Correlation plots of the LogFC of RING1B (R = −0.025 p = 0.029) and BCOR (R = 0.15 p = 1.61e-32) in VTP50469 treatment vs baseline H3K79me2 RPM. Correlation coefficients were calculated using the Pearson method with no adjustment for multiple comparisons. Data from 1 representative replicate of 2 biological replicates. **f.** Average profile plot of H3K79me2 levels for murine MLLAF9 cells treated with DMSO or VTP50469 for 4, 8, 16 or 24 h. Genes plotted (n = 45) have a high baseline H3K79me2 RPM with profound loss after treatment and significant downregulation. **g.** Genome browser snapshot of *Six1* for the same ChIP-seq tracks from main Fig. 5j.

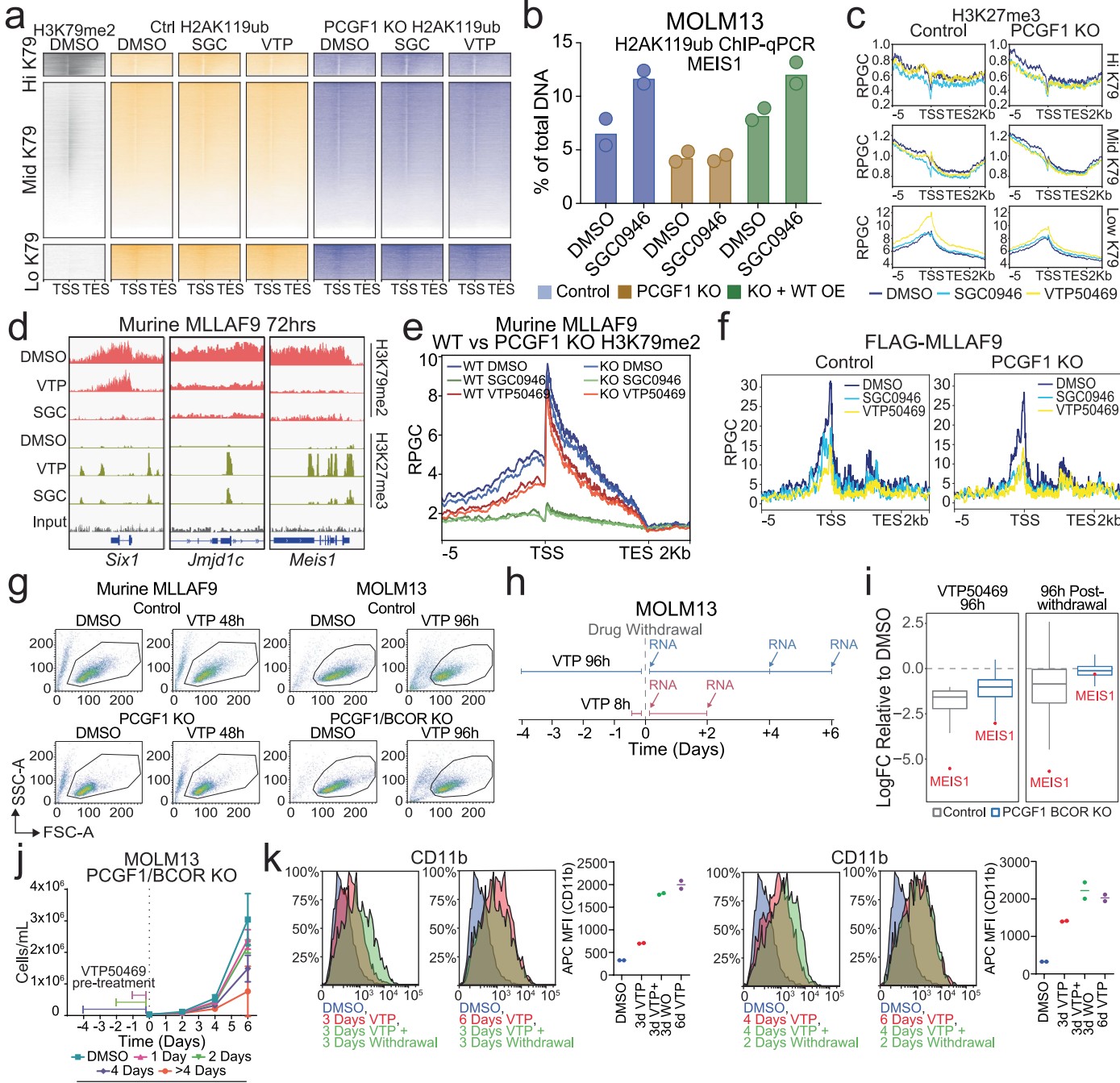

**Extended Data Fig. 7 | Menin inhibition induces irreversible Polycomb-mediated repression in MLL-rearranged and NPM1c-mutant leukaemia.**
**a.** Global heatmap plot of H3K79me2 and H2AK119ub in control and/or PCGF1 KO murine MLLAF9 cells treated with DMSO, SG0946 or VTP50469. Genes were sorted into 3 clusters (high, medium and low) using k-means clustering on the control cell H3K79me2 ChIP-seq data in DMSO. **b.** ChIP-qPCR analysis of control, PCGF1 KO, or PCGF1 KO expressing WT PCGF1 cDNA MOLM13 cells treated with DMSO or SGC0946 for H2AK119ub. Primers targeting the *MEIS1* promoter were used. Data from n = 2 independent experiments. **c.** Average profile plot of H3K27me3 in control and PCGF1 KO murine MLLAF9 cells treated with DMSO, SGC0946, or VTP50469. Genes were divided into 3 clusters as in (**a**) based on control cell H3K79me2 levels. **d.** Genome browser snapshots of *Six1*, *Jmjd1c,* and *Meis1* in control murine MLLAF9 cells with the indicated samples and treatments. **e.** Average profile plot of H3K79me2 in murine MLLAF9 control and PCGF1 KO cells treated with DMSO, SGC0946, or VTP50469. **f.** Average profile plot of FLAG-MLLAF9 in murine MLLAF9 control and PCGF1 KO cells treated with DMSO, SGC0946, or VTP50469 for the same gene subset is the same as in Fig. 4b, e.

**g.** FACS scatter plots of forward scatter (FSC-A) vs side scatter (SSC-A) for murine MLLAF9 control and PCGF1 KO cells treated with DMSO or VTP50469 (300 nM) for 48 h (left) or 96 h (right). **h.** Schematic of the VTP50469 drug withdrawal RNA sequencing experiment in MOLM13 cells. **i.** Box plots (centre line represents median, limits represent upper and lower quartiles, whiskers represent the maxima and minima if not exceeding 1.5*IQR) of the LogFC of RNA in MOLM13 control (grey) and PCGF1/BCOR KO (blue) under treatment and withdrawal conditions with *MEIS1* highlighted in red. Data from 3 biological replicates. **j.** Proliferation assay of MOLM13 PCGF1/BCOR KO cells treated with DMSO, VTP50469, or pretreated with VTP50469 (300 nM) for the indicated times before withdrawal. Data represents mean ± SD from n = 4 independent experiments. **k.** Histogram of the MFI of CD11b in control and PCGF1/BCOR KO MOLM13 cells treated with DMSO (blue), VTP50469 (red), or VTP50469 followed by drug withdrawal (green). Treatment times are indicated on each panel. MFI is also presented as a scatter dot plot (right). Data represents mean from n = 2 biological replicates. Source data available for **b**, **j**, and **k**.

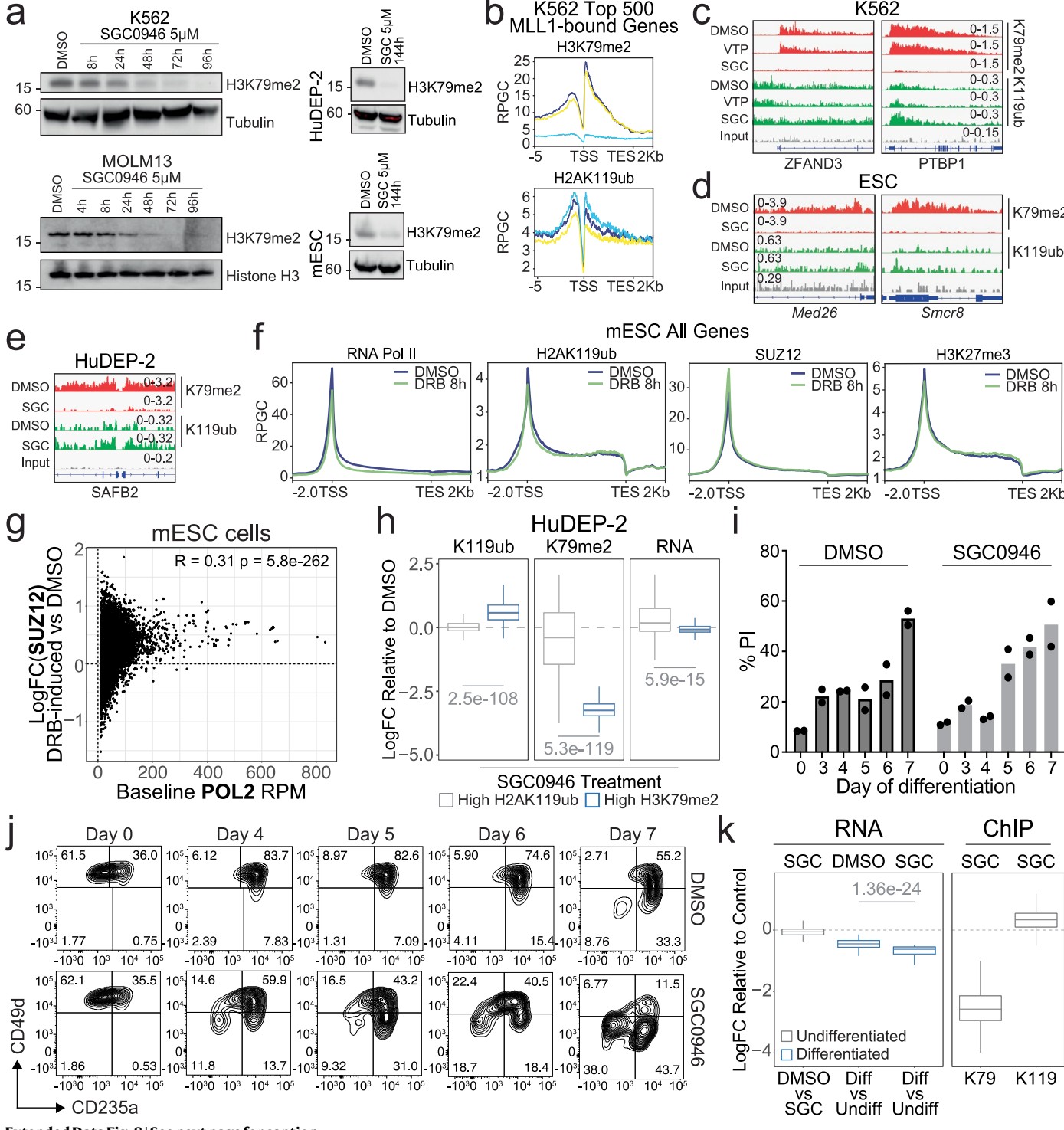

**Extended Data Fig. 8 | See next page for caption.**

**Extended Data Fig. 8 | DOT1L-PRC1 antagonism is conserved across difference cell types. a**. Western blot analysis of H3K79me2 in K562 (top), MOLM13 (middle), HuDEP-2, or mouse ES cells (bottom) treated with DMSO or SGC0946. Histone H3 or tubulin used as loading control. Data from 1 representative replicate of 2 biological replicates. **b**. Average profile plot of H3K79me2 and H2AK119ub in K562 cells treated with DMSO, SGC0946, or VTP50469 at the top 500 MLL1-bound genes[8]. **c**. Genome browser snapshots of ZFAND3 and PTBP1 in K562 treated with DMSO, VTP50469 (7 days), or SGC0946 (7 days). **d**. Genome browser snapshots of *Med26* and *Smcr8* in mouse ES cells treated with DMSO or SGC0946 (7 days). **e**. Genome browser snapshot of SAFB2 in HuDEP-2 cells treated with DMSO or SGC0946 (7 days). **f**. Global average profile plots of RNA Pol II, H2AK119ub, SUZ12, and H3K27me3 in mouse ES cells treated with DMSO or DRB (100 µM for 8hrs). **g**. Correlation plot of baseline RNA Pol II RPM against the LogFC of SUZ12 DRB treatment relative to DMSO (R = 0.31 p = 5.8e-262). Association tested using the Pearson method. **h**. Box plots of the LogFC of H3K79me2 (p = 2.5e-108), H2AK119ub (p = 5.3e-119), and RNA (p = 5.9e-15) in HuDEP-2 cells treated with SGC0946 relative to DMSO. **i**. Percentage of PI positive HuDEP-2 cells pre-treated with DMSO or SGC0946. **j**. Contour plots of CD49d and CD235a for HuDEP-2 cells pre-treated with DMSO or SGC0946. **k**. Box plots of RNA, H3K79me2, and H2AK119ub in HuDEP-2 cells. From left to right for the RNA boxplot: Undifferentiated SGC0946 treatment relative to undifferentiated DMSO, differentiated DMSO relative to undifferentiated DMSO, and differentiated SGC0946 pre-treatment relative to undifferentiated DMSO (p = 1.36e-24). The ChIP-seq boxplot (right) is the LogFC of SGC0946 treatment relative to DMSO for H3K79me2 (left) and H2AK119ub (right). Data from 3 biological replicates for RNA-seq and 1 replicate for ChIP-seq. For all box plots centre line represents median, limits represent upper and lower quartiles, whiskers represent the maxima and minima if not exceeding 1.5*IQR with p-values calculated using a two-sided Welch's t-test with no adjustment for multiple comparisons. Data from 3 biological replicates. Source data available for **a** and **i**.

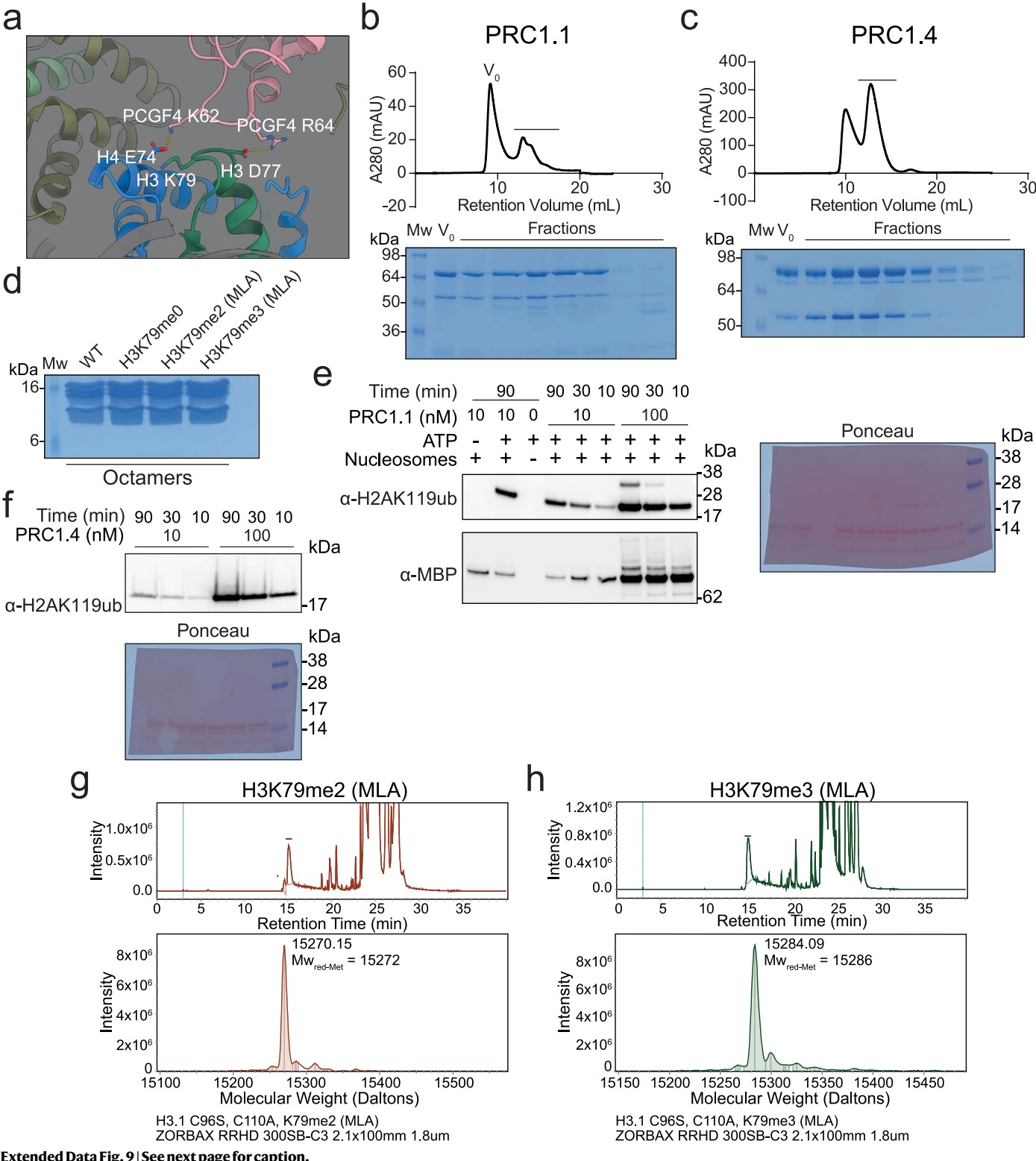

**Extended Data Fig. 9 | See next page for caption.**

**Extended Data Fig. 9 | H3K79me2/3 is sufficient to antagonize the histone ubiquitin ligase activity of PRC1. a**. Part of the interface between PCGF4, H3, and H4 from within the crystal structure of PRC1 in a complex with a nucleosome (PDB: 4R8P). Salt bridges with a distance of less than 3.5 Å are presented in dashed yellow lines. PCGF4, H3, and H4 are in pink, dark green and blue, respectively, and selected amino acids are presented. **b**. Size exclusion chromatography of RING1B-PCGF1 using Superdex 200 Increase 10/300 column and its fractions analysed using SDS-PAGE. PCGF1 is N-terminally tagged by MBP, and RING1B includes an N-terminal strep tag. Data from 1 replicate. **c**. Size exclusion chromatography of PCGF4-RING1B using HiLoad Superdex 200 16/600 columns, and its fractions analysed using SDS-PAGE. PCGF4 is N-terminally tagged by MBP, and RING1B includes an N-terminal strep tag. Data from 1 replicate. **d**. SDS-PAGE analysis of 5 µg of histone octamers made using unmodified or methyl-Lysine Analog (MLA) histones H3.1 as indicated. WT represents a wild-type H3.1 histone. H3K79me0 represents a non-MLA histone, without a K79C mutation, but with the C96S and C110A mutations that are included in the MLA histones. Data from 1 replicate. **e**. The activity of PRC1.1 (RING1B-PCGF1) on 1000 nM of unmodified wild-type nucleosomes was assayed at two PRC1.1 enzyme concentrations, as indicated. The reaction was stopped at three time points, as indicated. No ATP control, and E1/E2-ub only control were also performed as indicated. The experiment was carried out in two independent replicates, and a representative blot is presented. **f**. As (**e**), except that RING1B-PCGF4 was assayed, with samples as indicated. **g**. (Top) HPLC trace of H3K79me2 MLA histones separated on a ZORBAX RRHD 300SB-C3 2.1x100mm 1.8 µm column used for LC-MS. The horizontal line indicates the retention volume of the main product. An asterisk indicates a minor peak, corresponding to species with a nonspecific alkylation. (Bottom) Deconvoluted mass of species detected in the peak marked with a black horizontal line above. **h**. Same as (**g**), except that H3K79me3 MLA histone was analysed.

# Reporting Summary

## Statistics

For all statistical analyses, confirm that the following items are present in the figure legend, table legend, main text, or Methods section.

| n/a | Confirmed | |
|---|---|---|
| ☐ | ☒ | The exact sample size (*n*) for each experimental group/condition, given as a discrete number and unit of measurement |
| ☐ | ☒ | A statement on whether measurements were taken from distinct samples or whether the same sample was measured repeatedly |
| ☐ | ☒ | The statistical test(s) used AND whether they are one- or two-sided<br>*Only common tests should be described solely by name; describe more complex techniques in the Methods section.* |
| ☒ | ☐ | A description of all covariates tested |
| ☐ | ☒ | A description of any assumptions or corrections, such as tests of normality and adjustment for multiple comparisons |
| ☐ | ☒ | A full description of the statistical parameters including central tendency (e.g. means) or other basic estimates (e.g. regression coefficient) AND variation (e.g. standard deviation) or associated estimates of uncertainty (e.g. confidence intervals) |
| ☐ | ☒ | For null hypothesis testing, the test statistic (e.g. *F*, *t*, *r*) with confidence intervals, effect sizes, degrees of freedom and *P* value noted<br>*Give P values as exact values whenever suitable.* |
| ☒ | ☐ | For Bayesian analysis, information on the choice of priors and Markov chain Monte Carlo settings |
| ☒ | ☐ | For hierarchical and complex designs, identification of the appropriate level for tests and full reporting of outcomes |
| ☐ | ☒ | Estimates of effect sizes (e.g. Cohen's *d*, Pearson's *r*), indicating how they were calculated |

*Our web collection on statistics for biologists contains articles on many of the points above.*

## Software and code

Policy information about availability of computer code

| Data collection | Flow cytometry data was collected on either on BD LSRII or LSRFortessa (BD Biosciences), using DiVA V6.3.1.<br>Sequencing data was collected on either the NextSeq550 or NovaSeq6000 (Illumina) using Basespace to convert BCL files to FastQs |
|---|---|
| Data analysis | All murine sequencing data were aligned to mm10 and human sequencing data were aligned to hg38<br><br>CRISPR Screen<br>CRISPR screen data was analysed using MAGeCK ver. 0.5.9.4 and then plotted using Prism ver. 9<br><br>ChIP-seq<br>Reads were mapped using Bowtie2, BAM files were converted to TDFs using igvtools ver. 2.4.19 and bigwigs (BamCoverage), then heatmap and profile plots were made using deepTools ver. 3.1.3. Peaks were called using MACS2 ver. 2.1.1.2016309.<br><br>RNA-seq<br>Reads were trimmed using bbmap ver. 38.81 and aligned using STAR aligner ver. 2.7.9a and then fed into Degust for differential expression analysis.<br><br>RStudio<br>All RNA-seq and the non-deepTools ChIP-seq plots were made using RStudio ver. 4.0.2 with the following packages: biomaRt ver. 2.54.1, BioVenn ver. 1.1.3, BSgenome.Mmusculus.UCSC.mm10 ver. 1.4.3, ChIPpeakAnno ver. 3.32.0, csaw ver. 1.32.0, dplyr ver. 1.1.3, ggplot2 ver. 3.4.4, ggpubr ver. 0.6.0, ggrepel ver. 0.9.4, org.Hs.eg.db ver. 3.16.0, org.Mm.eg.db ver. 3.16.0, pheatmap ver. 1.0.12, profileplyR ver.. 1.14.1, reshape2 ver. 1.4.4, TxDb.Hsapiens.UCSC.hg38.knownGene ver 3.16.0, TxDb.Mmusculus.UCSC.mm10.ensGene ver. 3.4.0 |

Flow Cytometry
FlowLogic ver. 1.3 and FlowJo ver. 10

For manuscripts utilizing custom algorithms or software that are central to the research but not yet described in published literature, software must be made available to editors and reviewers. We strongly encourage code deposition in a community repository (e.g. GitHub). See the Nature Portfolio guidelines for submitting code & software for further information.

## Data

Policy information about availability of data

All manuscripts must include a data availability statement. This statement should provide the following information, where applicable:
- Accession codes, unique identifiers, or web links for publicly available datasets
- A description of any restrictions on data availability
- For clinical datasets or third party data, please ensure that the statement adheres to our policy

MLL1 ChIP-seq data in K562 cells (Extended Data Fig. 8b) GEO Accession Code GSE181829
RNA-seq data produced for this manuscript GEO Accession Code GSE260456
ChIP-seq data produced for this manuscript GEO Accession Code GSE260742

## Research involving human participants, their data, or biological material

Policy information about studies with human participants or human data. See also policy information about sex, gender (identity/presentation), and sexual orientation and race, ethnicity and racism.

| Reporting on sex and gender | N/A |
| --- | --- |
| Reporting on race, ethnicity, or other socially relevant groupings | N/A |
| Population characteristics | N/A |
| Recruitment | N/A |
| Ethics oversight | N/A |

Note that full information on the approval of the study protocol must also be provided in the manuscript.

# Field-specific reporting

Please select the one below that is the best fit for your research. If you are not sure, read the appropriate sections before making your selection.

☒ Life sciences    ☐ Behavioural & social sciences    ☐ Ecological, evolutionary & environmental sciences

For a reference copy of the document with all sections, see nature.com/documents/nr-reporting-summary-flat.pdf

# Life sciences study design

All studies must disclose on these points even when the disclosure is negative.

| Sample size | No statistical method was used to determine sample sizes and was selected prior to knowledge of the outcome. No power analysis was carried out. |
| --- | --- |
| Data exclusions | No data were excluded from this study except for those that were clear outliers due to technical problems in the assays. |
| Replication | Replicates are indicated in the figure legends and/or methods. Most assays were performed in at least biological triplicate. All experiments were able to be reliably reproduced. |
| Randomization | Mice were randomly assigned to cohorts at beginning of treatment. Cells were randomly split from the same pool of cells before being subject to treatment. Randomisation was applicable to other experiments. |
| Blinding | In vivo experiments utilized blinded technicians for assessing disease severity at end point. Investigators were not blinded to allocation during experiments and outcome assessment. |

# Reporting for specific materials, systems and methods

We require information from authors about some types of materials, experimental systems and methods used in many studies. Here, indicate whether each material, system or method listed is relevant to your study. If you are not sure if a list item applies to your research, read the appropriate section before selecting a response.

## Materials & experimental systems

| n/a | Involved in the study |
|-----|----------------------|
| ☐ | ☒ Antibodies |
| ☐ | ☒ Eukaryotic cell lines |
| ☒ | ☐ Palaeontology and archaeology |
| ☐ | ☒ Animals and other organisms |
| ☒ | ☐ Clinical data |
| ☒ | ☐ Dual use research of concern |
| ☒ | ☐ Plants |

## Methods

| n/a | Involved in the study |
|-----|----------------------|
| ☐ | ☒ ChIP-seq |
| ☐ | ☒ Flow cytometry |
| ☒ | ☐ MRI-based neuroimaging |

## Antibodies

| | |
|---|---|
| Antibodies used | DYKDDDDK Tag (D6W5B) Rabbit mAb (New England Biolabs 14793S)<br>Rabbit anti-MLL1 Antibody (Bio-Strategy BETHA300-086A-M)<br>Anti-trimethyl-Histone H3 (Lys4) (Merck 07-473)<br>Rabbit polyclonal to Histone H3 (di methyl K79) - ChIP Grade (ABCAM ab3594-100UG)<br>Rabbit polyclonal to Histone H3 (acetyl K9) (ABCAM ab4441-100UG)<br>Rabbit polyclonal to Histone H3 (acetyl K27) (ABCAM ab4729-100UG)<br>Ubiquityl-Histone H2B (Lys120) (D11) XP® Rabbit mAb (New England Biolabs 5546S)<br>RING1B (D22F2) XP® Rabbit mAb (New England Biolabs 5694S)<br>ChIPAb+™ JHDM1B Antibody (Merck 17-10264)<br>Ubiquityl-Histone H2A (Lys119) (D27C4) XP® Rabbit mAb (New England Biolabs 8240S)<br>Tri-Methyl-Histone H3 (Lys27) (C36B11) Rabbit mAb (New England Biolabs 9733S)<br>α-Tubulin (11H10) Rabbit mAb (HRP Conjugate) (New England Biolabs 9099S)<br>anti-mouse/human CD11b, APC-conjugated (BioLegend 301310)<br>Anti-human CD45, APC/Cy7-conjugated (BioLegend 304014)<br>Anti-mouse Ly6G Ly6C, FITC (Becton Dickinson 562060)<br>Anti-human CD11b, APC/Cy7 (Becton Dickinson 56103)<br>FITC Mouse Anti-Human CD235a BD Pharminigen 559943<br>PE/Cyanine7 Anti-Human CD49d BioLegend 304313<br>BCOR Rabbit Polyclonal antibody United BioResearch 12107-1-AP<br>Rabbit polyclonal Histone H3 ABCAM ab1791<br>H2AK119ub (for ubiquitination assay) CST D27C4<br>HRP-conjugated goat anti-rabbit Abcam A0545<br>Rabbit anti-H3 Abcam Ab1791<br>Mouse anti-MBP NEB E8032L<br>HRP-conjugated donkey anti-mouse Jackson Immuno Research 715-035-150 |
| Validation | All antibodies used have been validated in previous publications according to manufacturer's websites. |

## Eukaryotic cell lines

Policy information about cell lines and Sex and Gender in Research

| | |
|---|---|
| Cell line source(s) | MLLAF9 leukaemia cells were generated by magnetic bead selection (Miltenyi Biotec) of c-KIT+ cells from whole female mouse bone marrow and retroviral transduction with MLLAF9 constructs. MOLM13, MV4;11, K562, OCI-AML3 and HEK293T cells were obtained from ATCC or DSMZ. Murine embryonic stem cells and HuDEP-2 cells were obtained from collaborators. |
| Authentication | STR testing was performed to authenticate commercial cell lines |
| Mycoplasma contamination | Cell lines routinely tested negative for mycoplasma contamination. |
| Commonly misidentified lines (See ICLAC register) | No commonly misidentified cell lines were used in the study. |

## Animals and other research organisms

Policy information about studies involving animals; ARRIVE guidelines recommended for reporting animal research, and Sex and Gender in Research

| | |
|---|---|
| Laboratory animals | Female NOG Mice (NOD.Cg-Prkdcscidll2rgtm1Wjl/SzJ stock#005557) mice at 6-8 weeks of age |
| Wild animals | No wild animals were used. |

| | |
|---|---|
| Reporting on sex | To minimise variation in each mouse experiment, animals were all females. |
| Field-collected samples | No field samples were collected. |
| Ethics oversight | Animal experiments were approved by the Dana-Farber Cancer Institute's Institutional Animal Care and Use Committee. |

Note that full information on the approval of the study protocol must also be provided in the manuscript.

# Plants

| | |
|---|---|
| Seed stocks | N/A |
| Novel plant genotypes | N/A |
| Authentication | N/A |

# ChIP-seq

## Data deposition

☒ Confirm that both raw and final processed data have been deposited in a public database such as GEO.

☒ Confirm that you have deposited or provided access to graph files (e.g. BED files) for the called peaks.

| | |
|---|---|
| Data access links<br>*May remain private before publication.* | To review GEO accession GSE260742 |
| Files in database submission | GSM8123342 HUDEP-2 Control - DMSO input<br>GSM8123343 HUDEP-2 Control - DMSO H2KAK119ub<br>GSM8123344 HUDEP-2 Control - DMSO H3K79me2<br>GSM8123345 HUDEP-2 Control - SGC0946 5uM H2AK119ub<br>GSM8123346 HUDEP-2 Control - SGC0946 5uM H3K79me2<br>GSM8123347 K562 Control - DMSO input<br>GSM8123348 K562 Control - DMSO H2KAK119ub<br>GSM8123349 K562 Control - DMSO H3K79me2<br>GSM8123350 K562 Control - SGC0946 5uM H2AK119ub<br>GSM8123351 K562 Control - SGC0946 5uM H3K79me2<br>GSM8123352 K562 Control - VTP50469 500nM H2AK119ub<br>GSM8123353 K562 Control - VTP50469 500nM H3K79me2<br>GSM8123354 ESC Control - DMSO input<br>GSM8123355 ESC Control - DMSO H2KAK119ub<br>GSM8123356 ESC Control - DMSO H3K79me2<br>GSM8123357 ESC Control - SGC0946 5uM H2AK119ub<br>GSM8123358 ESC Control - SGC0946 5uM H3K79me2<br>GSM8123359 MLLAF9 Control - DMSO FLAG<br>GSM8123360 MLLAF9 Control - SGC0946 5uM FLAG<br>GSM8123361 MLLAF9 Control - VTP50469 500nM FLAG<br>GSM8123362 MLLAF9 Control - DMSO input<br>GSM8123363 MLLAF9 Control - DMSO input2<br>GSM8123364 MLLAF9 Control - SGC0946 5uM VTP50469 500nM H2AK119ub<br>GSM8123365 MLLAF9 Control - DMSO H2AK119ub<br>GSM8123366 MLLAF9 Control - SGC0946 5uM H2AK119ub<br>GSM8123367 MLLAF9 Control - VTP50469 500nM H2AK119ub<br>GSM8123368 MLLAF9 Control - SGC0946 5uM VTP50469 500nM H3K27me3<br>GSM8123369 MLLAF9 Control - DMSO H3K27me3<br>GSM8123370 MLLAF9 Control - SGC0946 5uM H3K27me3<br>GSM8123371 MLLAF9 Control - VTP50469 500nM H3K27me3<br>GSM8123372 MLLAF9 Control - DMSO H3K79me2<br>GSM8123373 MLLAF9 Control - SGC0946 5uM H3K79me2<br>GSM8123374 MLLAF9 Control - VTP50469 500nM H3K79me2<br>GSM8123375 MLLAF9 Control - DMSO input for ac<br>GSM8123376 MLLAF9 Control - DMSO input for RING1B<br>GSM8123377 MLLAF9 Control - DMSO H2AK119ub rep2<br>GSM8123378 MLLAF9 Control - SGC0946 5uM H2AK119ub rep2<br>GSM8123379 MLLAF9 Control - VTP50469 500nM H2AK119ub rep2<br>GSM8123380 MLLAF9 Control - DMSO H2AK120ub |

GSM8123381 MLLAF9 Control - SGC0946 5uM H2AK120ub
GSM8123382 MLLAF9 Control - VTP50469 500nM H2AK120ub
GSM8123383 MLLAF9 Control - DMSO H3K27ac
GSM8123384 MLLAF9 Control - SGC0946 5uM H3K27ac
GSM8123385 MLLAF9 Control - VTP50469 500nM H3K27ac
GSM8123386 MLLAF9 Control - DMSO H3K79me2 rep2
GSM8123387 MLLAF9 Control - SGC0946 5uM H3K79me2 rep2
GSM8123388 MLLAF9 Control - VTP50469 500nM H3K79me2 rep2
GSM8123389 MLLAF9 Control - DMSO H3K9ac
GSM8123390 MLLAF9 Control - SGC0946 5uM H3K9ac
GSM8123391 MLLAF9 Control - VTP50469 500nM H3K9ac
GSM8123392 MLLAF9 Control - DMSO KDM2B
GSM8123393 MLLAF9 Control - SGC0946 5uM KDM2B
GSM8123394 MLLAF9 Control - VTP50469 500nM KDM2B
GSM8123395 MLLAF9 Control - DMSO MLL1
GSM8123396 MLLAF9 Control - SGC0946 5uM MLL1
GSM8123397 MLLAF9 Control - VTP50469 500nM MLL1
GSM8123398 MLLAF9 Control - DMSO RING1B
GSM8123399 NTvKO PCGF1 KO DMSO H2AK119ub paired
GSM8123400 NTvKO PCGF1 KO DMSO input paired
GSM8123401 NTvKO PCGF1 KO SGC0946 5uM H2AK119ub paired
GSM8123402 NTvKO PCGF1 KO VTP50469 500nM H2AK119ub paired
GSM8123403 NTvKO Control - DMSO H2AK119ub paired
GSM8123404 NTvKO Control - DMSO input paired
GSM8123405 NTvKO Control - SGC0946 5uM H2AK119ub paired
GSM8123406 NTvKO Control - VTP50469 500nM H2AK119ub paired
GSM8123407 NTvKO PCGF1 KO DMSO H2AK119ub rep1
GSM8123408 NTvKO PCGF1 KO SGC0946 5uM H2AK119ub rep1
GSM8123409 NTvKO PCGF1 KO VTP50469 500nM H2AK119ub rep1
GSM8123410 NTvKO Control - DMSO H2AK119ub rep1
GSM8123411 NTvKO Control - SGC0946 5uM H2AK119ub rep1
GSM8123412 NTvKO Control - VTP50469 500nM H2AK119ub rep1
GSM8123413 NTvKO Control - DMSO input1
GSM8123414 NTvKO Control - DMSO input2
GSM8123415 NTvKO PCGF1 KO DMSO input
GSM8123416 NTvKO Control - DMSO input3
GSM8123417 NTvKO PCGF1 KO DMSO H2AK119ub rep2
GSM8123418 NTvKO PCGF1 KO SGC0946 5uM H2AK119ub rep2
GSM8123419 NTvKO PCGF1 KO VTP50469 500nM H2AK119ub rep2
GSM8123420 NTvKO Control - DMSO H2AK119ub rep2
GSM8123421 NTvKO Control - SGC0946 5uM H2AK119ub rep2
GSM8123422 NTvKO Control - VTP50469 500nM H2AK119ub rep2
GSM8123423 NTvKO PCGF1 KO DMSO H3K27me3
GSM8123424 NTvKO PCGF1 KO SGC0946 5uM H3K27me3
GSM8123425 NTvKO PCGF1 KO VTP50469 500nM H3K27me3
GSM8123426 NTvKO Control - DMSO H3K27me3
GSM8123427 NTvKO Control - SGC0946 5uM H3K27me3
GSM8123428 NTvKO Control - VTP50469 500nM H3K27me3
GSM8123429 NTvKO PCGF1 KO DMSO H3K4me3
GSM8123430 NTvKO PCGF1 KO SGC0946 5uM H3K4me3
GSM8123431 NTvKO PCGF1 KO VTP50469 500nM H3K4me3
GSM8123432 NTvKO Control - DMSO H3K4me3
GSM8123433 NTvKO Control - SGC0946 5uM H3K4me3
GSM8123434 NTvKO Control - VTP50469 500nM H3K4me3
GSM8123435 NTvKO PCGF1 KO DMSO H3K79me2
GSM8123436 NTvKO PCGF1 KO SGC0946 5uM H3K79me2
GSM8123437 NTvKO PCGF1 KO VTP50469 500nM H3K79me2
GSM8123438 NTvKO Control - DMSO H3K79me2
GSM8123439 NTvKO Control - SGC0946 5uM H3K79me2
GSM8123440 NTvKO Control - VTP50469 500nM H3K79me2
GSM8123441 Timecourse Control - DMSO H2AK119ub
GSM8123442 Timecourse Control - VTP50469 500nM 24h H2AK119ub
GSM8123443 Timecourse Control - VTP50469 500nM 8h H2AK119ub
GSM8123444 Timecourse Control - DMSO H3K27me3
GSM8123445 Timecourse Control - VTP50469 500nM 24h H3K27me3
GSM8123446 Timecourse Control - VTP50469 500nM 8h H3K27me3
GSM8123447 Timecourse Control - VTP50469 500nM 16h H3K79me2
GSM8123448 Timecourse Control - VTP50469 500nM 24h H3K79me2
GSM8123449 Timecourse Control - VTP50469 500nM 4h H3K79me2
GSM8123450 Timecourse Control - VTP50469 500nM 8h H3K79me2
GSM8123451 Timecourse Control - DMSO input
GSM8123452 Timecourse Control - DMSO H3K79me2
GSM8549820 MLLAF9 Control - DMSO H3K27me3 rep2
GSM8549821 MLLAF9 Control - VTP50469 500nM H3K27me3 rep2
GSM8549822 MLLAF9 Control - SGC0946 5uM H3K27me3 rep2
GSM8549823 MLLAF9 Control - input

GSM8549824 MLLAF9 Control - DMSO RING1B rep2
GSM8549825 MLLAF9 Control - VTP50469 500nM RING1B
GSM8549826 MLLAF9 Control - SGC0946 5uM RING1B
GSM8549827 MLLAF9 Control - DMSO BCOR
GSM8549828 MLLAF9 Control - VTP50469 500nM BCOR
GSM8549829 MLLAF9 Control - SGC0946 5uM BCOR
GSM8549830 MLLAF9 Control - DMSO H2AK119ub rep3
GSM8549831 MLLAF9 Control - VTP50469 500nM H2AK119ub rep3
GSM8549832 MLLAF9 Control - SGC0946 5uM H2AK119ub rep3
GSM8549833 mESC DRB - DMSO H3K27me3
GSM8549834 mESC DRB - DMSO H2AK119ub
GSM8549835 mESC DRB - DMSO POL2
GSM8549836 mESC DRB - DMSO SUZ12
GSM8549837 mESC DRB - DRB H3K27me3
GSM8549838 mESC DRB - DRB H2AK119ub
GSM8549839 mESC DRB - DRB POL2
GSM8549840 mESC DRB - DRB SUZ12
GSM8549841 mESC DRB - input
GSM8549842 MOLM13 NT - DMSO H3K27me3
GSM8549843 MOLM13 NT - VTP50469 500 nM H3K27me3
GSM8549844 MOLM13 NT - SGC0946 5uM H3K27me3
GSM8549845 MOLM13 PB - DMSO H3K27me3
GSM8549846 MOLM13 PB - VTP50469 500 nM H3K27me3
GSM8549847 MOLM13 PB - SGC0946 5uM H3K27me3
GSM8549848 MOLM13 NT - DMSO H2AK119ub
GSM8549849 MOLM13 NT - VTP50469 500 nM H2AK119ub
GSM8549850 MOLM13 NT - SGC0946 5uM H2AK119ub
GSM8549851 MOLM13 PB - DMSO H2AK119ub
GSM8549852 MOLM13 PB - VTP50469 500 nM H2AK119ub
GSM8549853 MOLM13 PB - SGC0946 5uM H2AK119ub
GSM8549854 MOLM13 NT - input
GSM8549855 MOLM13 NT - DMSO H3K79me2
GSM8549856 MOLM13 NT - SGC0946 5uM H3K79me2
GSM8549857 MOLM13 NT - VTP50469 500 nM H3K79me2
GSM8549858 MOLM13 PB - input
GSM8549859 MOLM13 PB - DMSO H3K79me2
GSM8549860 MOLM13 PB - SGC0946 5uM H3K79me2
GSM8549861 MOLM13 PB - VTP50469 500 nM H3K79me2
GSM8549862 K562 DRB - DMSO H2AK119ub
GSM8549863 K562 DRB - DRB H2AK119ub
GSM8549864 K562 DRB - DMSO H3K27me3
GSM8549865 K562 DRB - DRB H3K27me3
GSM8549866 K562 DRB - input

| | |
|---|---|
| Genome browser session<br>(e.g. UCSC) | *Provide a link to an anonymized genome browser session for "Initial submission" and "Revised version" documents only, to enable peer review. Write "no longer applicable" for "Final submission" documents.* |

## Methodology

| | |
|---|---|
| Replicates | Replicates are indicated in the figure legend, in some instances, only one ChIP-seq replicate was performed. |
| Sequencing depth | All ChIP-seq samples were sequenced to a depth of at least 10million single end reads on a NEXT-seq or NOVA-seq. |
| Antibodies | DYKDDDDK Tag (D6W5B) Rabbit mAb (New England Biolabs 14793S)<br>Rabbit anti-MLL1 Antibody (Bio-Strategy BETHA300-086A-M)<br>Anti-trimethyl-Histone H3 (Lys4) (Merck 07-473)<br>Rabbit polyclonal to Histone H3 (di methyl K79) - ChIP Grade (ABCAM ab3594-100UG)<br>Rabbit polyclonal to Histone H3 (acetyl K9) (ABCAM ab4441-100UG)<br>Rabbit polyclonal to Histone H3 (acetyl K27) (ABCAM ab4729-100UG)<br>Ubiquityl-Histone H2B (Lys120) (D11) XP® Rabbit mAb (New England Biolabs 5546S)<br>RING1B (D22F2) XP® Rabbit mAb (New England Biolabs 5694S)<br>ChIPAb+™ JHDM1B Antibody (Merck 17-10264)<br>Ubiquityl-Histone H2A (Lys119) (D27C4) XP® Rabbit mAb (New England Biolabs 8240S)<br>Tri-Methyl-Histone H3 (Lys27) (C36B11) Rabbit mAb (New England Biolabs 9733S) |
| Peak calling parameters | Broad peaks were called using MACS2 with the following parameters: --SPMR -B -f BAM -g mm --broad. The input non targeting sample bam file was provided as the control (-c). |
| Data quality | Quality control was performed on the raw sequencing data using FastQC, and the quality scores were found to be within acceptable ranges. |
| Software | Reads were mapped using Bowtie2, BAM files were converted to TDFs using igvtools ver. 2.4.19 and bigwigs (BamCoverage), then heatmap and profile plots were made using deepTools ver. 3.1.3. Peaks were called using MACS2 ver. 2.1.1.2016309. |

# Flow Cytometry

## Plots

Confirm that:

☒ The axis labels state the marker and fluorochrome used (e.g. CD4-FITC).

☒ The axis scales are clearly visible. Include numbers along axes only for bottom left plot of group (a 'group' is an analysis of identical markers).

☒ All plots are contour plots with outliers or pseudocolor plots.

☒ A numerical value for number of cells or percentage (with statistics) is provided.

## Methodology

| | |
|---|---|
| Sample preparation | Cell lines were collected and cell pellets were resuspended in 100 μL of FACS buffer (cell lines) |
| Instrument | Flow cytometry was performed using a BD LSRII or LSRFortessa (BD Biosciences), or a Cytek Northern Lights and cell sorting was performed using a BD Fusion3/5 (BD Biosciences) |
| Software | Analysis was performed using Flowlogic software (Inivai Technologies, Australia) and FlowJo (BD). |
| Cell population abundance | Sorts for mCherry, BFP and GFP were sorted to near 100% purity prior to experiments. |
| Gating strategy | Gating was done based on morphology (FSC, SSC) and then for the annexin experiments (Fig. 1i) a representative gating strategy was provided. |

☒ Tick this box to confirm that a figure exemplifying the gating strategy is provided in the Supplementary Information.

