## [Peer Review File · Nature Cell Biology]

DOT1L provides transcriptional memory through PRC1.1 antagonism

Corresponding Author: Dr Omer Gilan

Version 0:

Decision Letter:

*Please delete the link to your author homepage if you wish to forward this email to co-authors.

Dear Dr Gilan,

Please accept our sincere apologies for a delayed review process. Your manuscript, "DOT1L functions as a transcriptional memory component of the MLL-Polycomb axis", has now been seen by 3 referees, who are experts in cancer and genome integrity (referee 1); epigenetics and cancer (referee 2); and PRC in cancer (referee 3). As you will see from their comments (attached below) they find this work of potential interest, but have raised substantial concerns, which in our view would need to be addressed with considerable revisions before we can consider publication in Nature Cell Biology.

Nature Cell Biology editors discuss the referee reports in detail within the editorial team, including the chief editor, to identify key referee points that should be addressed with priority, and requests that are overruled as being beyond the scope of the current study. To guide the scope of the revisions, I have listed these points below. We are committed to providing a fair and constructive peer-review process, so please feel free to contact me if you would like to discuss any of the referee comments further.

In particular, it would be essential to:

- A- Perform further experiments to determine how DOT1L or H3K79me3/me2 prevent PRC1.1 binding (Referee #1)
- B- Determine how DOT1L or H3K79me3/me2 prevent PRC1.1 binding
- C- Clarify the CRISPR-based screen and validation and validation of ChIP-seq data in human cells
- D- Analyze the mechanisms underlying the proposed epigenetic changes, as well as their downstream effect.

E-All other referee concerns pertaining to strengthening existing data, providing controls, methodological details, clarifications and textual changes, as per all referees, should also be addressed.

F-Finally please pay close attention to our guidelines on statistical and methodological reporting (listed below) as failure to do so may delay the reconsideration of the revised manuscript. In particular please provide:

We would be happy to consider a revised manuscript that would satisfactorily address these points, unless a similar paper is published elsewhere, or is accepted for publication in Nature Cell Biology in the meantime.

- ensure that it conforms to our format instructions and publication policies (see below and <https://www.nature.com/nature/for-authors>).

- provide a point-by-point rebuttal to the full referee reports verbatim, as provided at the end of this letter.

- provide the completed Reporting Summary (found here https://www.nature.com/documents/nr-reporting-summary.pdf). This is essential for reconsideration of the manuscript will be available to editors and referees in the event of peer review. For more information see http://www.nature.com/authors/policies/availability.html or contact me.

When submitting the revised version of your manuscript, please pay close attention to our [href="https://www.nature.com/nature-portfolio/editorial-policies/image-integrity">Digital Image Integrity Guidelines. and to the following points below:](https://www.nature.com/nature-portfolio/editorial-policies/image-integrity)

Nature Cell Biology is committed to improving transparency in authorship. As part of our efforts in this direction, we are now requesting that all authors identified as 'corresponding author' on published papers create and link their Open Researcher and Contributor Identifier (ORCID) with their account on the Manuscript Tracking System (MTS), prior to acceptance. ORCID helps the scientific community achieve unambiguous attribution of all scholarly contributions. You can create and link your ORCID from the home page of the MTS by clicking on 'Modify my Springer Nature account'. For more information please visit please visit www.springernature.com/orcid.

This journal strongly supports public availability of data. Please place the data used in your paper into a public data repository, or alternatively, present the data as Supplementary Information. If data can only be shared on request, please explain why in your Data Availability Statement, and also in the correspondence with your editor. Please note that for some data types, deposition in a public repository is mandatory - more information on our data deposition policies and available repositories appears below.

Link Redacted

We would like to receive a revised submission within six months.

We hope that you will find our referees' comments, and editorial guidance helpful. Please do not hesitate to contact me if there is anything you would like to discuss.

Best wishes,

Sabrya Carim

Sabrya Carim, PhD
(she/her/hers)
Associate Editor, Nature Cell Biology
Nature Portfolio

Springer Nature
The Campus, 4 Crinan Street, London N1 9XW, UK
sabrya.carim@springernature.com
<https://orcid.org/0000-0001-9485-1938>

Reviewers' Comments:

Reviewer #1:

Remarks to the Author:

The authors sought to investigate how Menin and DOT1L contribute to MLL-FP function. They made some important findings through CRISPR screens, CRISPR knockouts (KOs), inhibitor treatments for Menin and DOT1L, and measuring various chromatin modifications and modifiers, such as the Polycomb Repressive system targeting MLL-FP targets and the mutual exclusion of H3K79me2 and H2AK119Ub. The authors also performed experiments to test the prevailing model in Polycomb repression. With solid evidence, they convincingly showed the effects of Pcgf1 and BCOR KO on Menin and DOT1L inhibition in various cell lines. I find this work novel, exciting, and a valuable contribution to the field of cancer epigenetics

I do have specific comments for the authors to address.

- The authors should evaluate their claims regarding PRC1.1's role in their system as follows:

- When assessing the role of PRC1.1, they measure H2AK119Ub. However, measuring H2AK119Ub does not necessarily indicate PRC1.1 activity; it is a measurement of RING1B activity (subunit common to all PRC1 complexes).
- In analyzing the role of PRC1.1 in repressing MLL-FP targets, they find that KDM2B shows little to no increase at MLL-FP targets (see Fig. 4e). However, they show that H2AK119Ub undergoes relatively more significant changes (Fig. 4e). This suggests that PRC1.1 is not the only complex contributing to the silencing of MLL-FP target genes. Other subcomplexes may be involved in this silencing. This possibility should be mentioned and, if possible, further explored, as previous data attribute this effect to PRC1.1.
- When assessing the role of PRC1.1 in repressing MLL-FP targets, why analyze all PRC1 complexes by ChIPing RING1B? Why not ChIP BCOR or Pcgf1?
- For the sgRNA screen, how is it that both UTX and PRC2 are identified as important for growth advantage? Shouldn't these have opposing effects?
- The authors observe that the three cell lines studied (MOLM13, MV4;11, Murine MLLAF9) have exceedingly rare overlaps in differentially expressed genes. While they focus on Meis1, an important gene with overlap between all cells, they don't discuss why there

is such little general overlap. Can the authors clearly explain this in the text and be consistent between different experiments with the different cell lines?

- Why aren't apoptosis measurements discussed in Figure 3F presented alongside Figure 1? It seems it would make more sense since the authors are discussing cell survival.
- "Remarkably, H3K79me2 loss and baseline levels were highly correlated ($r=0.51$ and $r=0.66$, respectively) with H2AK119ub induction after SGC0946 treatment (Fig. 5d)." A correlation of 0.51 is not considered high, but 0.66 is on the border of high correlation. It is an overstatement to say there is a high correlation.

Figure Comments:

- The cell nucleus in Figure 1G contains information that is uninterpretable unless highly zoomed in.
- In Figures 2B and C, it is hard to distinguish between PCGF1 KO and BCOR KO. Use different colors or data representation, such as dotted lines with no fill.

Specific Comments:

- The authors should evaluate their claims regarding PRC1.1's role in their system as follows:

- When assessing the role of PRC1.1, they measure H2AK119Ub. However, measuring H2AK119Ub does not necessarily indicate PRC1.1 activity; it is a measurement of RING1B activity (subunit common to all PRC1 complexes).
- In analyzing the role of PRC1.1 in repressing MLL-FP targets, they find that KDM2B shows little to no increase at MLL-FP targets (see Fig. 4e). However, they show that H2AK119Ub undergoes relatively more significant changes (Fig. 4e). This suggests that PRC1.1 is not the only complex contributing to the silencing of MLL-FP target genes. Other subcomplexes may be involved in this silencing. This possibility should be mentioned and, if possible, further explored, as previous data attribute this effect to PRC1.1.
- When assessing the role of PRC1.1 in repressing MLL-FP targets, why analyze all PRC1 complexes by ChIPing RING1B? Why not ChIP BCOR or Pcgf1?
- For the sgRNA screen, how is it that both UTX and PRC2 are identified as important for growth advantage? Shouldn't these have opposing effects?
- The authors observe that the three cell lines studied (MOLM13, MV4;11, Murine MLLAF9) have exceedingly rare overlaps in differentially expressed genes. While they focus on Meis1, an important gene with overlap between all cells, they don't discuss why there is such little general overlap. Can the authors clearly explain this in the text and be consistent between different experiments with the different cell lines?
- Why aren't apoptosis measurements discussed in Figure 3F presented alongside Figure 1? It seems it would make more sense since the authors are discussing cell survival.
- "Remarkably, H3K79me2 loss and baseline levels were highly correlated ($r=0.51$ and $r=0.66$, respectively) with H2AK119ub induction after SGC0946 treatment (Fig. 5d)." A correlation of 0.51 is not considered high, but 0.66 is on the border of high correlation. It is an overstatement to say there is a high correlation.

Figure Comments:

- The cell nucleus in Figure 1G contains information that is uninterpretable unless highly zoomed in.
- In Figures 2B and C, it is hard to distinguish between PCGF1 KO and BCOR KO. Use different colors or data representation, such as dotted lines with no fill.

Reviewer #2:

Remarks to the Author:

Menin and DOT1L inhibitors are in clinical trials for the treatment of AML. The authors have addressed the mechanism by which Menin and DOT1L as part of an MLL-Fusion oncoprotein complex contribute to drive MLL-FP leukemogenesis. By performing CRISPR based screens they identified components of the non-canonical PRC1.1 complex, BCOR and PCGF1 as being required for a sustained response to the DOT1L and Menin inhibitors. Interestingly the MLL3/MLL4 COMPASS as well as the PRC2 complex were also identified as hits, suggesting either that these proteins function together (as we know for PRC1-PRC2), or that parallel – and not explained pathways – are involved in the stable silencing of MLL-FP target genes in response to Menin and DOT1L inhibition. Validation studies were performed both in vivo and in vitro, showing the requirement for PRC1.1 in mediating a phenotypic effect to DOT1L and Menin inhibitors. This was further extended to transcriptional studies. Moreover, in agreement with H3K79me2 being lost and transcriptional downregulation, the authors observed an increase in deposition of H2AK119ub1 and H3K27me3. Interestingly, temporary DOT1L disruption led to irreversible transcription changes at target genes, potentially supporting some level of transcriptional memory. Finally, an antagonistic link between DOT1L and PRC1.1 was observed in many different cell types, suggesting a general role of DOT1L in balancing active and repressive chromatin states.

MLL-fusion proteins are believed to induce leukemia through a mechanism that involves the recruitment of DOT1L and the super elongation complex. This leads to increased transcriptional elongation of target genes and the development of leukemia. Previous studies from Scott Armstrong's lab, studying the role of MLL-FPs in AML (using a model like the one used in the submitted manuscript), have shown (see e.g. Deshpande et al, 2014 PMID 25464900) that H3K79me2/me3 (catalyzed by DOT1L) correlate with HOXA gene expression, as well as exclusion of the repressive H3K27me3 modification at the HOXA7-10 locus. Moreover, Deshpande et al showed that deletion of AF10, component of the super elongation complex leads to impaired DOT1L association with HOXA cluster genes, loss of H3K79me3/me2, and the spread of H3K27me3, which coincides with a decrease in gene expression. Therefore, the previous studies from the Armstrong lab have established a strong correlation between MLL-FPs, DOT1L, H3K79me3/me2 and transcriptional elongation, and shown that this is anticorrelated with H3K27me3. Moreover, they have shown that DOT1L loss leads to increased H3K27me3 levels. Despite the strong relevance of the Deshpande paper and other studies from the Armstrong lab, the papers are not cited in the submitted manuscript. While the authors may be correct that 'it is still unknown how MLL-FP, Menin and DOT1L influence the function of the Polycomb complexes', the submitted manuscript does not provide real novel insight into this question. In fact, most of the manuscript establishes what we know, i.e. an anticorrelation between MLL-FP fusion proteins and the binding of Polycomb group proteins, and an anticorrelation between H3K79me2/me3 and H3K27me3. How DOT1L or H3K79me3/me2 prevent PRC1.1 binding is not established, and it is still likely that the Polycomb group proteins are first bound, once transcription has stopped. In other words, a well-established mechanism for maintaining transcriptional repression by the Polycomb group proteins.

Other comments:

1. Several hits which were identified in CRISPR survival screen included BCOR, PCGF1 (PRC1.1), KMT2D and NCOA6 (MLL3/MLL4 COMPASS complex) and EED, EZH2, SUZ12 and JARID2 (PRC2). It would be great to understand how PRC2 was identified as one of the hits in survival screen but turned out to be essential for cell viability. This maybe important to understand/address a success/validity of CRISPR screen in these cell lines.

2. The authors used BCOR, PCGF1 or double KO cells throughout the study and claim that depletion of PRC1.1 confers resistance to DOT1L or Menin inhibitors in leukemic cells. It would be interesting to test if knockout of other PRC1.1 complex members, such as KDM2B leads to as similar phenotype. The authors should also test reversibility of the phenotypes and rule out clonal effects by re-introducing BCOR/PCGF1 in the KO cells. Chromatin analysis should be performed to establish if re-establishment of PRC1.1 can restore ubiquitination and hence H3K27me3 levels, and subsequently down-regulate transcription of key MLL target genes.

3. Previous studies by van den Boom et al 2016 (PMID 26748712) demonstrated an essential role for non-canonical PRC1.1 in leukemic cells. This conflicts with results reported in this study. Noticeably, both studies used MLL-AF9 cells. The authors should discuss potential explanations for these discrepancies.

4. The authors claim a potential role for DOT1L in transcriptional memory. However, the question is whether this is due to a slow turnover of H3K79me2 in response to the Menin inhibitor. Alternatively, the results could be explained by that transcriptional inhibition is required for the recruitment of the Polycomb group proteins, which agrees with the widely accepted concept that the Polycomb group proteins do not initiate transcriptional repression but are required for maintaining transcriptional repression.

Minor points:

1. Genome-wide and epigenetic focused CRISPR-screens were performed in both human and murine leukemia cells, but the manuscript only focuses on the epigenetic factors. For completion, the authors should provide some discussion on their findings from genome-wide CRISPR screen or some rationale behind for why needing this extra data.

2. For having a complete data set the authors may want to generate a BCOR KO MLL-AF9 cell line and compare it to a PCGF1 KO MLL-AF9 cell line in extended data figure 2a).

Reviewer #3:

Remarks to the Author:

This manuscript explores critical epigenetic events within the context of MLL fusion protein-related leukemias. Specifically, it investigates the role of DOT1L as a primary determinant in mitigating Polycomb associated repression. Remarkably, this mechanism appears to exhibit a broader, conserved impact beyond the hematopoietic system. While this concept presents intriguing insights into our understanding of molecular mechanisms driving oncohematological transformation, the experimental findings would benefit from a more systematic and transparent presentation to robustly substantiate the authors' conclusions.

Please find below our main concerns regarding each section/figure:

Figure 1

Authors should make an effort to explain the CRISPR-based screening of Figure 1. Please provide the details of the cell lines used and the design of the experiments in a way that it is easy to follow for the readers. For example, they mention the usage of two libraries, but in Methods there is just description of a chromatin library. Also, we could not find the data relative to the screening with the VTP50469 inhibitor in MV4;11 cells.

For how long was the experiment carried out? In the scheme of Figure 1a, 12 and 24 days are mentioned, but in the Extended Data Figure 1a and 1b it is written "13 days".

The initial results section, which forms the basis for the rest of the manuscript, should be enhanced and presented systematically for better clarity and coherence.

Validation of the screening results are also difficult to follow. Why when depicting results in MV4;11 cells the % of sgRNA is shown, but the results in MOLM13 refer to cell number? And why the authors validate the data in MOLM13 cells? But then in the supplementary material it seems the opposite (the cell lines used in Supp figures 1e and 1f are not indicated). Also, is there any validation in murine cells?

The experiment depicted in Fig 1g is difficult to follow. It is also unclear why the vectors used for the sgRNAs targeting PCGF1 and BCOR contains BFP.

The fact that the double KO is conferring survival specifically during Menin and DOT1L inhibition, but not in treatments with IBET-151 or ABT-199, is an interesting finding that and it should be included in the main figure.

Figure 2

Why BCOR KO is not shown in the Murine cell line MLLAF9 (Extended Figure 2e)?

Given that PRC1 functions as a repressor complex, it would be important to determine whether the behavior illustrated in Figure 2d is also observed for up-regulated genes.

Figure 3

With respect to the CellRadar RNA-seq panels, it is unclear whether the gene-set selected upon PCGF1 KO are in the presence of

inhibitors. No analysis has been shown before on the effect of transcriptomics of PCGF1 KO without MLL-AF inhibitors. The Cell Radar RNA-seq analysis should be performed upon treatment with MLL-AF inhibitors AND PCGF1 KO.

Upon analyzing the cytometer data, it appears that the loss of PRC1.1 alone has a significant impact on AML cells, as demonstrated in the CellRadar panels. However, this effect does not extend to markers such as Ly6G. How do the authors explain this?

Why transplants in mice are performed with human cells lines? Why not using the murine MLLAF9?

In transplant experiments, PRC1 KO alone does not cause MOLM13 to be more malignant, this seems counterintuitive taking into account the Cell Radar results. Please comment on this.

Indicate the number of mice used in the study directly in the figure or in the main text.

Figure 4

Since the ChIP experiments were performed using murine cells, it would be important to validate the results in human cells. To properly quantify the ChIP-seq data the authors should use spike-in controls.

After checking the global occupancy of MLL proteins, for histone modifications the authors selected a subset of genes to continue their analysis. How this subset is selected should be clearly stated in the main text and not only in the figure caption. For example, it is interesting that from the top-100 genes bound by MLL, only 38 are significantly downregulated upon inhibitors treatment. What is the effect of the treatment for the rest of the genes/loci?

The increase in repressive marks seems very modest, even in the selected subset of genes. Could the authors provide statistics for Figure 4e?

It seems that for the genes selected for Fig 4g, the changes observed are in line with the authors description, yet this is much less clear for the rest of genes, as plotted in Fig 4f. Can the authors provide statistical analysis for Fig 4f, or at least describe how those selected genes in Fig 4g were chosen.

Figure 5

The section starts with the suggestion that MLL-FPs may prevent Polycomb repression. To study this, the authors focused on all downregulated genes (n=176) after both SGC and VTP treatments. For consistency the authors should focus on the set of genes analyzed in figure 4, or at least justify the change.

By performing the analysis of genome analysis using 'bins', they observed differences in the dynamics of H2K79me3 mark. The authors suggest that this is because the enzymatic activity is abolished globally. Later, there are additional results on the dynamics of H3K79me2 loss. Are the regions depicted in Fig 5g losing the active mark after 48 hours? In other words, the authors should use the same loci in panel 5g and panel 5j.

What is depicted in Fig 5h? Transcription? Of which subset of genes? It is not clear neither in the figure nor the caption.

For the loci that are still decorated with H3K79me2 after Menin eviction treatment, the authors provide in the Discussion section a different explanation with respect to the data interpretation described in the results section of Figure 5. They claim this may be possible through the activity of the native DOT1L. If this is the case, even at longer time points, those loci will not be devoid of H3K79me3. The authors should clearly explain by which mechanism, under their point of view, Menin inhibitors behave differently than the DOT1L inhibitor. Are both mechanisms (delay and native DOT1L) possible at the same time?

Figure 6

ChIPseq data presented in panels 6a and 6d should be quantified (using boxplots, and including the corresponding statistical analysis) to appreciate the impact.

It looks that H3K27me3 is increased for the Top 1000 MLLAF genes upon Menin inhibition (VTP). How is this possible when the increase of H2AK119ub is supposed to be more limited, as reported by the authors in the previous section? And, why switching from Top100 to Top1000 genes?

Why is the increase in H3K27me3 observed only upon VTP treatment (as depicted in panel 6d), but not with SGC treatment?

In panel 6k, the authors included a set of data featuring OCIAML3 cells, which is not listed in the Materials and Methods section. Furthermore, there is no specific commentary on these cells in the Results section. Additionally, aside from the title, there is no further mention of NPM1 in the discussion. If experiments related to NPM1 are included, they should be properly explained and discussed to ensure coherence with the rest of the manuscript.

Finally, regarding the discussion, the authors should mention previous works (Van der Boom 2016, Cell Rep; Maat 2021, iScience; Schmidt 2022, Blood Cancer Discovery) that study the relation between MLLAF9 and PRC1.1 in the context of leukemia.

Methods should be written concisely, but should contain all elements necessary to allow interpretation and replication of the results. As a guideline, Methods sections typically do not exceed 3,000 words. The Methods should be divided into subsections listing reagents and techniques. When citing previous methods, accurate references should be provided and any alterations should be noted. Information must be provided about: antibody dilutions, company names, catalogue numbers and clone numbers for monoclonal antibodies; sequences of RNAi and cDNA probes/primers or company names and catalogue numbers if reagents are commercial; cell line names, sources and information on cell line identity and authentication. Animal studies and experiments involving human subjects must be reported in detail, identifying the committees approving the protocols. For studies involving human subjects/samples, a statement must be included confirming that informed consent was obtained. Statistical analyses and information on the reproducibility of experimental results should be provided in a section titled "Statistics and Reproducibility".

All Nature Cell Biology manuscripts submitted on or after March 21 2016 must include a Data availability statement as a separate section after Methods but before references, under the heading "Data Availability". For Springer Nature policies on data availability see <http://www.nature.com/authors/policies/availability.html>; for more information on this particular policy see <http://www.nature.com/authors/policies/data/data-availability-statements-data-citations.pdf>. The Data availability statement should include:

- Accession codes for primary datasets (generated during the study under consideration and designated as "primary accessions") and secondary datasets (published datasets reanalysed during the study under consideration, designated as "referenced accessions"). For primary accessions data should be made public to coincide with publication of the manuscript. A list of data types for which submission to community-endorsed public repositories is mandated (including sequence, structure, microarray, deep sequencing data) can be found here <http://www.nature.com/authors/policies/availability.html#data>.
- Unique identifiers (accession codes, DOIs or other unique persistent identifier) and hyperlinks for datasets deposited in an approved repository, but for which data deposition is not mandated (see here for details <http://www.nature.com/sdata/data-policies/repositories>).
- At a minimum, please include a statement confirming that all relevant data are available from the authors, and/or are included with the manuscript (e.g. as source data or supplementary information), listing which data are included (e.g. by figure panels and data types) and

mentioning any restrictions on availability.

- If a dataset has a Digital Object Identifier (DOI) as its unique identifier, we strongly encourage including this in the Reference list and citing the dataset in the Methods.

We recommend that you upload the step-by-step protocols used in this manuscript to the Protocol Exchange. More details can found at www.nature.com/protocolexchange/about.

All imaging data should be accompanied by scale bars, which should be defined in the legend.

Cropped images of gels/blots are acceptable, but need to be accompanied by size markers, and to retain visible background signal within the linear range (i.e. should not be saturated). The boundaries of panels with low background have to be demarked with black lines. Splicing of panels should only be considered if unavoidable, and must be clearly marked on the figure, and noted in the legend with a statement on whether the samples were obtained and processed simultaneously. Quantitative comparisons between samples on different gels/blots are discouraged; if this is unavoidable, it should only be performed for samples derived from the same experiment with gels/blots were processed in parallel, which needs to be stated in the legend.

Supplementary items should relate to a main text figure, wherever possible, and should be mentioned sequentially in the main

manuscript, designated as Supplementary Figure, Table, Video, or Note, and numbered continuously (e.g. Supplementary Figure 1, Supplementary Figure 2, Supplementary Table 1, Supplementary Table 2 etc.).

The total number of Supplementary Figures (not including the "unprocessed scans" Supplementary Figure) should not exceed the number of main display items (figures and/or tables (see our Guide to Authors and March 2012 editorial <http://www.nature.com/ncb/authors/submit/index.html#suppinfo>; <http://www.nature.com/ncb/journal/v14/n3/index.html#ed>). No restrictions apply to Supplementary Tables or Videos, but we advise authors to be selective in including supplemental data.

GUIDELINES FOR EXPERIMENTAL AND STATISTICAL REPORTING

REPORTING REQUIREMENTS – We are trying to improve the quality of methods and statistics reporting in our papers. To that end, we are now asking authors to complete a reporting summary that collects information on experimental design and reagents. The Reporting Summary can be found here <https://www.nature.com/documents/nr-reporting-summary.pdf>. If you would like to reference the guidance text as you complete the template, please access these flattened versions at <http://www.nature.com/authors/policies/availability.html>.

Version 1:

Decision Letter:

*Please delete the link to your author homepage if you wish to forward this email to co-authors.

Dear Dr Gilan,

Thank you for your patience with the peer review process. Your manuscript, "DOT1L functions as a transcriptional memory component of the MLL-Polycomb axis", has now been seen by all of our original referees, who are experts in cancer and genome integrity (referee 1); epigenetics and cancer (referee 2); and PRC in cancer (referee 3). As you will see from their comments (attached below) they find this work of interest, but have raised some remaining important points. Although we are also very interested in this study, we believe that their concerns should be addressed before we can consider publication in Nature Cell Biology.

Nature Cell Biology editors discuss the referee reports in detail within the editorial team, including the chief editor, to identify key referee points that should be addressed with priority, and requests that are overruled as being beyond the scope of the current study. We are committed to providing a fair and constructive peer-review process, so please feel free to contact me if you would like to discuss any of the referee comments further.

It would be essential to address all the reviewers' remaining concerns pertaining to strengthening existing data, providing controls, methodological details, clarifications and textual changes.

Finally please pay close attention to our guidelines on statistical and methodological reporting (listed below) as failure to do so may delay the reconsideration of the revised manuscript. In particular please provide:

In contrast, we would overrule reviewer#2's concerns (point 1) around the conceptual novelty provided by the work. Therefore, addressing this point experimentally would not be required for reconsideration of the study at the journal.

We therefore invite you to take these points into account when revising the manuscript. In addition, when preparing the revision please:

- ensure that it conforms to our format instructions and publication policies (see below and www.nature.com/nature/authors/).

- provide a point-by-point rebuttal to the full referee reports verbatim, as provided at the end of this letter.

- provide the completed Editorial Policy Checklist (found here <https://www.nature.com/authors/policies/Policy.pdf>), and Reporting Summary (found here <https://www.nature.com/authors/policies/ReportingSummary.pdf>). This is essential for reconsideration of the manuscript and these documents will be available to editors and referees in the event of peer review. For more information see <http://www.nature.com/authors/policies/availability.html> or contact me.

Nature Cell Biology is committed to improving transparency in authorship. As part of our efforts in this direction, we are now requesting that all authors identified as 'corresponding author' on published papers create and link their Open Researcher and Contributor Identifier (ORCID) with their account on the Manuscript Tracking System (MTS), prior to acceptance. ORCID helps the scientific community achieve unambiguous attribution of all scholarly contributions. You can create and link your ORCID from the home page of the MTS by clicking on 'Modify my Springer Nature account'. For more information please visit <http://www.springernature.com/orcid>.

Link Redacted

We would like to receive the revision within four weeks. If submitted within this time period, reconsideration of the revised manuscript will not be affected by related studies published elsewhere, or accepted for publication in Nature Cell Biology in the meantime. We would be happy to consider a revision even after this timeframe, but in that case we will consider the published literature at the time of resubmission when assessing the file.

We hope that you will find our referees' comments, and editorial guidance helpful. Please do not hesitate to contact me if there is anything you would like to discuss.

Best wishes,

Sabrya Carim

Sabrya Carim, PhD
(she/her/hers)
Associate Editor, Nature Cell Biology
Nature Portfolio

Springer Nature
The Campus, 4 Crinan Street, London N1 9XW, UK
sabrya.carim@springernature.com
<https://orcid.org/0000-0001-9485-1938>

Reviewers' Comments:

Reviewer #1 (Remarks to the Author):

The authors provided thorough responses to each of my concerns. They offered thoughtful feedback, made meaningful revisions, and conducted additional experiments to support their claims further and address my critiques. I believe the manuscript is suitable for publication however, it could be even stronger if the authors consider the following comments:

1. When the authors address choosing PRC1.1 over RING1B, they claim that KO of PCGF1 prevented a global increase in H2AK119Ub levels (Fig. 6a-c, Extended Data Figure 6a). While their claim is valid, I noticed that in medium and high H3K79me levels, there is a significant increase of H2AK119Ub even in PCGF1 KO (in the drug treatment conditions). While not as large of a change as wt, it is still a

change. This might be important when considering that H3K79me regions might be the critical regions for PCGF1 in this disease model. I wondered if other polycomb complexes might contribute to this increase of H2AK119Ub at H3K79me regions.

2. When the authors compare BCOR and KDM2B at MLLAF9 target genes, it is clear there is some overlap. But it isn't exactly clear how much overlap. This makes it difficult because the reader must compare a main figure to an extended figure at the same time. I suggest having the ChIP Seq tracks together so one can easily compare. I would also add that seeing the global levels of BCOR vs. KDM2B would be nice. Even if they may have other functions that explain why they may not have a perfect overlap, it would still be interesting to see how much of it is overlapping in their model systems.

3. Finally, I suggest that the authors tone down the paragraph below, where they propose mechanisms without any further biochemical and structural analyses, weakening their overall story. There is also a misstatement as the PCGF1 complex containing additional subunits is more active than the catalytic E3 ligase core (please see Rose et al., PMID: 27705745). PRC1 complexes containing PCGF4 and PCGF1 contain different subunits essential for localization, nucleosome-binding, and enzymatic activity. Therefore, the analyses that the authors propose are too speculative.

'The conservation of this functional interplay across different cell types suggests that a fundamental mechanism underpins this antagonism. To identify a potential explanation for how H3K79me2 could influence PRC1 activity downstream of recruitment, we sought to explore published structural data. Given that the PCGF proteins are the only non-enzymatic subunits of PRC1 that are critical for in vitro PRC1 activity [46], we reasoned that PCGF1 may influence PRC1.1 activity through specific interactions with the nucleosome. In support of this, available structural data of RING1B/PCGF4 complex bound to a nucleosome has shown that the PCGF4 K62 and R64 residues form salt bridges with H4E74 and H3D77, respectively, with D77 being two residues away from H3K79 [47] (Extended Data Fig. 9a). Importantly, alignment of PCGF1 and PCGF4 as well as structural modelling and AlphaFold3 analysis, suggests that these same salt bridges cannot be formed in the context of PCGF1, where the positively charged PCGF4 K62 is predicted to turn into the negatively charged PCGF1 E91 and the positively charged PCGF4 R64 is predicted to turn into the polar PCGF1 Q93 (Extended Data Fig. 9ac). Therefore, we reasoned that the PCGF1 E91 residue could in principle form a salt bridge with H3K79, which may be altered due to structural rearrangement by the presence of di trimethylation.'

Reviewer #2 (Remarks to the Author):

The authors have performed an extensive revision of the manuscript in response to the comments and concerns we raised, and they have pointed to the possibility that PCGF1/RING1B as part of the PRC1.1 complex interacts differently with nucleosomes than PCGF4/RING1B, thereby explaining its lack of interaction with H3K79me2.

The main question for us was (and is) whether the data in the manuscript supports the conclusions. Moreover, we also questioned the conceptual novelty of the findings reported.

The authors main conclusions are: We demonstrate that inhibition of transcription is not sufficient to induce Polycomb-mediated repressive histone modifications, especially at active genes. We also provide evidence that H3K79me2 mostly influences the activity of PRC1/PRC1.1, rather than its recruitment to chromatin.

Our comments:

1. The authors extend the many previous reported results showing an anti-correlation between H3K79me2 and H3K27me3 to H2AK119ub1. These published studies (e.g. PMID 33060580, 21741597) have not reported on an anti-correlation between H3K79me2 and H3K27me3 on a global scale but have focused on the genes that are required for development or leukemia, balancing activation (H3K79me2) and repression (H3K27me3). Although the demonstration that PRC1.1 is involved in setting up H2AK119ub1 and H3K27me3 has not been reported before, we do not consider this as a major conceptual novel finding, because of the known close link between PRC1 and PRC2 and therefore H2K119ub1 and H3K27me3 mediated transcriptional repression. The novelty of the manuscript is presumably that H3K79me2 directly inhibits the catalytic activity of PRC1.1, which, if demonstrated mechanistically (see below), is novel – however, perhaps better suited for another journal.

2. The demonstration that inhibition is not sufficient to induce Polycomb-mediated repressive histone modifications at active genes is interesting, however, we are not convinced that the results presented by the authors fully support this statement. Previously published results have demonstrated that Polycomb group proteins are recruited following transcriptional termination, and that they are required for maintaining transcriptional repression. The authors acknowledge this, and the results presented in Fig. 2i are also consistent with this notion. However, the authors concluded that transcription does not explain the observed increase in PRC1 activity. They pointed out that, unlike the Menin inhibitor, the DOT1L inhibitor not only down-regulated genes but also showed a marked difference in H2AK119ub1 deposition, indicating that the depletion of H3K79me2 affects PRC1 activity rather than transcriptional changes. Additionally, this effect did not result in a global increase in H3K27me3.

The authors present several experiments to support this claim. The authors observed absence of a global increase in H3K27me3 could be attributed to the relative short time points used and the fact that PRC2 will only work onto the genes it is recruited. Several published results have shown that H3K27me3 is established with slow kinetics in vitro and after prolonged PRC2 binding in vivo (see e.g. PMID 21078963; 19541851; 24561620). The authors analyzed published data and conducted their own experiments using ChIP-seq for SUZ12, POL2, H3K27me3, and H2AK119ub1 in mESCs, as well as H3K27me3 and H2AK119ub1 for K562 cells treated with DMSO or DRB for 8 hours. However, the data presented in Reviewer Figures 2 and 3 do not convincingly support the claim that transcription inhibition alone is insufficient to induce PRC1/2 activity, as these experiments were conducted with very short time points. The binding of SUZ12 or other PRC complexes would precede the establishment of H3K27me3 but requires a longer incubation time for the spreading of this mark due to the gradual recruitment of PRC2, its necessity for specific nucleation sites, positive feedback for spreading, and the coordinated chromatin changes that occur progressively over multiple cell cycles. In the analyses the authors also need to specifically analyze the increase of H3K27me3 at sites where SUZ12 is recruited in response to the treatment versus sites where SUZ12 is not bound.

3. The authors have provided some interesting experiments addressing the mechanism by which H3K79me2 potentially inhibits the

activity of PRC1.1. By structural modeling based on data from the RING1B/PCGF4 complex they suggest that E91 of PCGF1 should form a salt bridge with H3K79, which may be altered due to structural rearrangement by the presence of trimethylation (interestingly the authors do not measure H3K79me3 at any point in the manuscript). Therefore, they suggest that RING1B/PCGF1 activity is inhibited by H3K79me3. To understand whether this is the case they generate a PCGF1 E91K mutant. If we follow the rationale of the authors such mutant should not be affected by H3K79me3 and lead to the faster H2AK119ub1 of DOT1L inhibited genes to which PRC1.1 is bound. However, instead of testing this, the authors show that WT PCGF1 expression restores sensitivity to DOT1L and Menin inhibition, while the mutant restores sensitive a bit less (Fig. 7o). In the Extended figures 9f, g the authors show minor and non-significant effects between wt and E91K PCGF1 on MEIS1 expression, and the association of H2AK119ub1 and H3K27me3 with two DOT1L regulated genes. In other words, the experiments are not conclusive, perhaps partly because steady-state levels of H2AK119ub1 are measured.

To provide potentially more conclusive experiments the authors need to perform time-resolved experiments to test how mutant and wt PCGF1 differentially lead to H2AK119ub1 association with target genes in response to DOT1L inhibition. Moreover, they need to perform biochemical experiments to test PCGF1wt/RING1B and PCGF1mut/RING1B recombinant complexes for their activity on wt and H3K79me2 nucleosomes. Such data will directly test the authors hypothesis and potentially lead to experimental data supporting it.

4. The authors describe that a major difference between DOT1L inhibition and Menin inhibition is that Menin inhibition prevents MLL-FP binding, whereas DOT1L does not. However, in Fig. 4a, b they show that both compounds lead to reduction of MLL-AF9 binding. Moreover, all the significant downregulated genes show similar levels of downregulation using the two inhibitors (Figs. 4g, 5a, 5g, Extended Figs 4a, b, d, 5a). Why is MLL-AF9 lost upon DOT1L inhibition and what does this observation mean for the interpretation of the results? The authors should discuss this.

Additional comments:

There are still issues in data presentation and interpretation, particularly the following points:

- Figure 1h: Why do the DKO cells grow faster than the single KO and the wt cells?
- Extended Data Figure 1d: The RING1B sgRNA VTP and KDM2B sgRNA VTP do not show a significant increase in growth. Are these data statistically significant?
- Extended Data Figure 1i: The different graphs are difficult to separate, and interpretation is therefore difficult.
- Extended Data Figure 1m: The authors claim that "depletion of PRC1.1 did not affect the degree of H3K79me2 loss after DOT1L inhibition, indicating that resistance is not due to changes in DOT1L activity." However, the knockout of PRC1.1 (BCOR) leads to a greater loss of H3K79me2 at baseline levels compared to the parental line. Therefore, depletion of PRC1.1 may partially involve changes in DOT1L activity.
- Figures 2e and Extended Data Figure 2g: It appears that only MEIS1 is rescued by PCGF1 knockout. For genes such as Hoxa9, Jmjd1c (Figure 2e), and Runx2, MEF2C, there does not seem to be significant rescue, as log2FC values are similar in control and PCGF1 knockout cells. Thus, PCGF1 knockout does not appear to rescue many genes.
- Extended Data Figure 4d: The panels have not been labeled for which histone modifications have been tested.
- Figure 4g and Extended Data Figure 4e: In Figure 4g, JMJD1C and MEIS1 genes in MLL-AF9 cells treated with either VTP/SGC or both showed a consistent increase in H2AK119ub1 and H3K27me3. In contrast, Extended Data Figure 4e shows no signal of H3K27me3 for the JMJD1C gene in control MOLM13 cells treated with VTP/SGC, despite an increase in H2AK119ub1. Only the MEIS1 gene in control MOLM13 cells treated with VTP (not SGC) showed an increase in both H2AK119ub1 and H3K27me3. Why did SGC treatment not show an increase in these marks for this key gene? Such discrepancies challenge the authors' conclusion that "the induction of H2AK119ub was also accompanied by an increase in H3K27me3 deposition spread over the same regions of MLLAF9 target genes," and then "we also observed similar induction of repressive histone modifications at key MLL-AF9 target genes in human (MOLM13) leukaemia cells".
- Figure 5g: For completion the author should show H3K27me3 at these loci.
- Figure 6b: The order of the panels has been switched between the H2AK119ub1 (DMSO, VTP, SGC) and H3K27me3 (DMSO < SGC, VTP). This is slightly confusing
- Extended Data Figure 7a and b: The heatmaps show very weak signals of all the post translational modifications and it is not possible to assess the effects of SGC and VTP treatments. Are these differences statistically significant at all?

Reviewer #3 (Remarks to the Author):

We appreciate the effort the authors have made to address all our concerns regarding the original manuscript. While most of our specific points have been addressed, we still have some remaining concerns. Additionally, we would like to emphasize (once again) that any further efforts to enhance this manuscript's readability are highly appreciated, as it contains a significant amount of data that is difficult to follow. For instance, why is the genome-wide screen conducted in human cells, while the epigenetic screen is done in murine cells? Is there a specific scientific rationale behind this choice, or was it simply exploratory at this stage? We also fail to understand why MOLM13 cells are used for validations.

Regarding Cell Radar, if the knockout (KO) effects on transcription are so dramatic by themselves (such as a clear increase in genes related to, for example, ST-HSC), our previous comment remains relevant: why was this not considered in Figure 2? It is stated in the rebuttal and in several other sections that PCGF1 KO only blunts the transcriptional response to treatment, but this statement seems inaccurate, as it appears that PCGF1 KO by itself has significant effects, independent of the treatment. In any case, we suggest presenting the Cell Radar with the treatment alone, as well as with both PCGF1 KO and the treatment. While the KO may indeed blunt the transcriptional response to the treatment, there could be many other genes affected by the KO that are independent of the treatment. We believe this is an important set of data to include.

Again, the fact that the KO alone induces changes independently of the treatment makes it difficult to fully grasp the relevance of the cytometry and transplant experiments. While the authors have provided explanations in the rebuttal, the readers may still find this confusing. The data in the new panel, Fig. 3i, may help clarify some points, but once again, showing the Cell Radar for KO + treatment, compared to KO alone, is important. A clearer distinction between the effects caused by KO alone and those resulting from its interaction with the treatment should be explored, or at the very least, clearly discussed in the manuscript.

"Indicate the number of mice used in the study directly in the figure or in the main text."

"Thank you, we have now added this directly to the figure."

We are still not able to find the number of mice used in these experiments.

The ChIP experiments shown in Fig 4f seem unnecessary if they are not showing statistical significance. Just to clarify: are Mais1 and Jmjd1c part of the selected groups of genes (11), which are both MLLAF9 targets and rescued by PRC1.1? It is quite difficult to propose a general mechanism based on such a limited number of genes, especially one that occurs at the same loci. From panel 4e it seems clearer (although again for very few genes). The authors may be interested in the results of panel 4f because of the effect of double treatment on H3K27me3, but in general panel 4f seems to confuse the overall message. We found that these panels are not very well described in the results section, and the readers may be lost or may need further clarification when reaching this part of the manuscript.

We appreciate the 'Reviewer Fig. 4', as it provides consistency. We suggest including it in the manuscript.

GUIDELINES FOR SUBMISSION OF NATURE CELL BIOLOGY ARTICLES

ARTICLE FORMAT

ABSTRACT – should not exceed 150 words and should be unreferenced. This paragraph is the most visible part of the paper and should briefly outline the background and rationale for the work, and accurately summarize the main results and conclusions. Key genes, proteins and organisms should be specified to ensure discoverability of the paper in online searches.

TEXT – the main text consists of the Introduction, Results, and Discussion sections and must not exceed 3500 words including the abstract. The Introduction should expand on the background relating to the work. The Results should be divided in subsections with subheadings, and should provide a concise and accurate description of the experimental findings. The Discussion should expand on the findings and their implications. All relevant primary literature should be cited, in particular when discussing the background and specific findings.

REFERENCES – are limited to a total of 70 in the main text and Methods combined,. They must be numbered sequentially as they appear in the main text, tables and figure legends and Methods and must follow the precise style of Nature Cell Biology references. References only cited in the Methods should be numbered consecutively following the last reference cited in the main text. References only associated with Supplementary Information (e.g. in supplementary legends) do not count toward the total reference limit and do not need to be cited in numerical continuity with references in the main text. Only published papers can be cited, and each publication cited should be included in the numbered reference list, which should include the manuscript titles. Footnotes are not permitted.

Methods should be written concisely, but should contain all elements necessary to allow interpretation and replication of the results. As a guideline, Methods sections typically do not exceed 3,000 words. The Methods should be divided into subsections listing reagents and techniques. When citing previous methods, accurate references should be provided and any alterations should be noted. Information must be provided about: antibody dilutions, company names, catalogue numbers and clone numbers for monoclonal antibodies; sequences of RNAi and cDNA probes/primers or company names and catalogue numbers if reagents are commercial; cell line names,

sources and information on cell line identity and authentication. Animal studies and experiments involving human subjects must be reported in detail, identifying the committees approving the protocols. For studies involving human subjects/samples, a statement must be included confirming that informed consent was obtained. Statistical analyses and information on the reproducibility of experimental results should be provided in a section titled "Statistics and Reproducibility".

All Nature Cell Biology manuscripts submitted on or after March 21 2016, must include a Data availability statement as a separate section after Methods but before references, under the heading "Data Availability". For Springer Nature policies on data availability see <http://www.nature.com/authors/policies/availability.html>; for more information on this particular policy see <http://www.nature.com/authors/policies/data/data-availability-statements-data-citations.pdf>. The Data availability statement should include:

- Accession codes for primary datasets (generated during the study under consideration and designated as "primary accessions") and secondary datasets (published datasets reanalysed during the study under consideration, designated as "referenced accessions"). For primary accessions data should be made public to coincide with publication of the manuscript. A list of data types for which submission to community-endorsed public repositories is mandated (including sequence, structure, microarray, deep sequencing data) can be found here <http://www.nature.com/authors/policies/availability.html#data>.
- Unique identifiers (accession codes, DOIs or other unique persistent identifier) and hyperlinks for datasets deposited in an approved repository, but for which data deposition is not mandated (see here for details <http://www.nature.com/sdata/data-policies/repositories>).
- At a minimum, please include a statement confirming that all relevant data are available from the authors, and/or are included with the manuscript (e.g. as source data or supplementary information), listing which data are included (e.g. by figure panels and data types) and mentioning any restrictions on availability.
- If a dataset has a Digital Object Identifier (DOI) as its unique identifier, we strongly encourage including this in the Reference list and citing the dataset in the Methods.

We recommend that you upload the step-by-step protocols used in this manuscript to [protocols.io](http://www.protocols.io). More details can be found at <https://www.protocols.io/help/publish-articles>.

DISPLAY ITEMS – main display items are limited to 6-8 main figures and/or main tables. For Supplementary Information see below.

FIGURES – Colour figure publication costs \$395 per colour figure. All panels of a multi-panel figure must be logically connected and arranged as they would appear in the final version. Unnecessary figures and figure panels should be avoided (e.g. data presented in small tables could be stated briefly in the text instead).

All imaging data should be accompanied by scale bars, which should be defined in the legend.

Cropped images of gels/blots are acceptable, but need to be accompanied by size markers, and to retain visible background signal within the linear range (i.e. should not be saturated). The boundaries of panels with low background have to be demarked with black lines. Splicing of panels should only be considered if unavoidable, and must be clearly marked on the figure, and noted in the legend with a statement on whether the samples were obtained and processed simultaneously. Quantitative comparisons between samples on different gels/blots are discouraged; if this is unavoidable, it has to be performed for samples derived from the same experiment with gels/blots were processed in parallel, which needs to be stated in the legend.

Regardless of format, all figures must be vector graphic compatible files, not supplied in a flattened raster/bitmap graphics format, but should be fully editable, allowing us to highlight/copy/paste all text and move individual parts of the figures (i.e. arrows, lines, x and y axes, graphs, tick marks, scale bars etc). The only parts of the figure that should be in pixel raster/bitmap format are photographic images or 3D rendered graphics/complex technical illustrations.

Unprocessed scans of all key data generated through electrophoretic separation techniques need to be presented in a supplementary figure that should be labeled and numbered as the final supplementary figure, and should be mentioned in every relevant figure legend. This figure does not count towards the total number of figures and is the only figure that can be displayed over multiple pages, but should be provided as a single file, in PDF or TIFF format. Data in this figure can be displayed in a relatively informal style, but size markers and the figures panels corresponding to the presented data must be indicated.

The total number of Supplementary Figures (not including the “unprocessed scans” Supplementary Figure) should not exceed the number of main display items (figures and/or tables (see our Guide to Authors and March 2012 editorial <http://www.nature.com/ncb/authors/submit/index.html#suppinfo>; <http://www.nature.com/ncb/journal/v14/n3/index.html#ed>). No restrictions apply to Supplementary Tables or Videos, but we advise authors to be selective in including supplemental data.

GUIDELINES FOR EXPERIMENTAL AND STATISTICAL REPORTING

REPORTING REQUIREMENTS – To improve the quality of methods and statistics reporting in our papers we have recently revised the reporting checklist we introduced in 2013. We are now asking all life sciences authors to complete two items: an Editorial Policy Checklist (found here <https://www.nature.com/authors/policies/Policy.pdf>) that verifies compliance with all required editorial policies and a Reporting Summary (found here <https://www.nature.com/authors/policies/ReportingSummary.pdf>) that collects information on experimental design and reagents. These documents are available to referees to aid the evaluation of the manuscript. Please note that these forms are dynamic ‘smart pdfs’ and must therefore be downloaded and completed in Adobe Reader. We will then flatten them for ease of use by the reviewers. If you would like to reference the guidance text as you complete the template, please access these flattened versions at <http://www.nature.com/authors/policies/availability.html>.

We strongly recommend the presentation of source data for graphical and statistical analyses as a separate Supplementary Table, and request that source data for all independent repeats are provided when representative experiments of multiple independent repeats, or averages of two independent experiments are presented. This supplementary table should be in Excel format, with data for different figures provided as different sheets within a single Excel file. It should be labelled and numbered as one of the supplementary tables, titled “Statistics Source Data”, and mentioned in all relevant figure legends.

Version 2:

Decision Letter:

Dear Dr Gilan,

Thank you for your patience as we evaluated your revised manuscript. Your revised manuscript "DOT1L functions as a transcriptional memory component of the MLL-Polycomb axis", has now been seen by 3 referees, who are experts in cancer and genome integrity (referee 1); epigenetics and cancer (referee 2); and PRC in cancer (referee 3), and whose comments are pasted below. In light of their advice, we regret that we cannot offer to publish the study in Nature Cell Biology.

As you will see, although the reviewers find the work improved in revision, and while reviewers 1 and 3 offer positive comments, reviewer 2 has persisting concerns with regard to the mechanism underlying how H3K79me2 inhibits the catalytic activity of PRC1.1 and an antagonistic relationship between DOT1L and PRC1, and we consider these reservations sufficiently important to preclude publication of the study in Nature Cell Biology.

Although we cannot offer to publish your manuscript, I suggest that you consider Nature Communications as a suitable venue for this work. To transfer your manuscript, please use our manuscript transfer portal. You will not have to re-supply manuscript metadata and files, unless you wish to make modifications. For more information, please see our [manuscript transfer FAQ](http://www.nature.com/authors/author_resources/transfer_manuscripts.html?WT.mc_id=EMI_NPG_1511_AUTHORTRANSF&WT.ec_id=AUTHOR) page. Alternatively, I would be happy to consult with my colleagues at Nature Communications in mid-January 2024 to explore whether they would commit to take the revised manuscript forward with the existing peer review history.

We are very sorry that we could not be more positive on this occasion, but we thank you for the opportunity to consider this work.

With kind regards,
Sabrya Carim

Sabrya Carim, PhD
(she/her/hers)
Senior Editor, Nature Cell Biology
Nature Portfolio

Springer Nature
The Campus, 4 Crinan Street, London N1 9XW, UK
sabrya.carim@springernature.com
<https://orcid.org/0000-0001-9485-1938>

Reviewers' comments:

Reviewer #1 (Remarks to the Author):

While I still believe that the emphasis on structural modeling is somewhat excessive and the manuscript could benefit from further de-emphasizing this aspect, the authors present a very interesting and compelling story. I have no additional questions beyond those raised in my previous review concerning the structural modeling. Therefore, I see no reason to delay publication further.

Reviewer #2 (Remarks to the Author):

As written in our latest review of this manuscript: 'The novelty of the manuscript is presumably that H3K79me2 directly inhibits the catalytic activity of PRC1.1, which, if demonstrated mechanistically (see below), is novel – however, perhaps better suited for another journal.'

In the revised version of the manuscript, the authors have chosen not to provide any mechanistically based experiments to show that H3K79me2 inhibits the catalytic activity of PRC1.1 as requested in our overall comment and in detail in comment 3:

'To provide potentially more conclusive experiments the authors need to perform time-resolved experiments to test how mutant and wt PCGF1 differentially lead to H2AK119ub1 association with target genes in response to DOT1L inhibition. Moreover, they need to perform biochemical experiments to test PCGF1 wt/RING1B and PCGF1 mut/RING1B recombinant complexes for their activity on wt and H3K79me2 nucleosomes. Such data will directly test the authors hypothesis and potentially lead to experimental data supporting it.'

The authors have not experimentally addressed any of the requested experiments, and the main conclusion of the manuscript is therefore based on correlation and not mechanistic insights.

The authors claim that 'the manuscript reveals a novel antagonistic relationship between DOT1L and PRC1, with DOT1L functioning to establish transcriptional memory to protect genes from premature polycomb repression'. They do not. They show that the previously known anticorrelation between DOT1L and PRC1, established on several genes can be extended to a global level, but they do not provide the antagonistic relationship based on a mechanistic understanding. Our recommendation of the suitability of the publication of the manuscript in Nature Cell Biology has therefore not changed.

Reviewer #3 (Remarks to the Author):

Regarding my first point on clarifying the screening strategy, I now find the text and the author's narrative clear.

The analysis of gene expression prior to treatment has also been included. While transcriptomic changes caused by the KO alone are significant enough to impact cell identity (as shown by cell radar analysis), the authors now show that the top genes affected by the treatment were not previously influenced by the KO. This finding is, in fact, more relevant than the cell radar analysis itself. When considering all conditions, only minimal differences are observed between treatment with and without the KO. Therefore, if cell radar analysis is presented, it should include all conditions, as shown in the rebuttal. Otherwise, it should be omitted entirely.

Regarding the ChIP experiments in Fig. 4f, the authors should carefully decide how to present these results. As mentioned previously, profiles with such a small number of genes might be confusing for readers. However, the authors now provide a clearer explanation in the text about the relevance when presenting these results.

**For Nature Portfolio general information and news for authors, see <http://npg.nature.com/authors>.

Version 3:

Decision Letter:

Our ref: NCB-A53382C-Z

16th October 2025

Dear Dr. Gilan,

Thank you for submitting your revised manuscript "DOT1L provides memory of transcription through antagonism of PRC1" (NCB-A53382C-Z). It has now been seen by the original referees and their comments are below. The reviewers find that the paper has improved in revision, and therefore we'll be happy in principle to publish it in Nature Cell Biology, pending minor revisions to satisfy the referees' final requests and to comply with our editorial and formatting guidelines.

Please ensure to address the remaining reviewer#2 points ("additional comments") in a revised version of the manuscript.

Please ensure that all figures fit into a single standard page and adhere to a maximum page size of roughly 180mm wide x 200mm high. To ensure legibility once figures are re-sized, please use a font size of no smaller than 6pt throughout the figures. Please ensure that the figure panel labels are not too close to the page margins.

We are now performing detailed checks on your paper and will send you a checklist detailing our editorial and formatting requirements in about a week. **Please do not upload the final materials and make any revisions until you receive this additional information from us**.

Thank you again for your interest in Nature Cell Biology Please do not hesitate to contact me if you have any questions.

Sincerely,

Sabrya

Sabrya Carim, PhD
(she/her/hers)
Senior Editor, Nature Cell Biology
Nature Portfolio

Springer Nature
The Campus, 4 Crinan Street, London N1 9XW, UK
sabrya.carim@springernature.com
<https://orcid.org/0000-0001-9485-1938>

Reviewer #2 (Remarks to the Author):

This is the fourth version of the manuscript under review for Nature Cell Biology. The fundamental novelty of the work has not changed. Although the authors have added new data in this revision, we remain of the opinion that the results are incremental in the context of the literature, and that the manuscript would be more appropriate for another journal.

In our previous reviews, we asked the authors to provide mechanistic insights of how H3K79me2 directly inhibits the catalytic activity of PRC1.1. Specifically, in the last review we asked the authors: '... to perform time-resolved experiments to test how mutant and wt PCGF1

differentially lead to H2AK119ub1 association with target genes in response to DOT1L inhibition. Moreover, they need to perform biochemical experiments to test PCGF1wt/RING1B and PCGF1mut/RING1B recombinant complexes for their activity on wt and H3K79me2 nucleosomes. Such data will directly test the authors hypothesis and potentially lead to experimental data supporting it.'

In response to this request, the authors have removed the results related to the PCGF1 mutant and the final paragraph of the result section in the manuscript. Through additional experiments, they found that the previous results were not supported by the new data we had asked them to generate (fortunately, the previous results were not published). Instead, they now shown that di- and trimethylation of H3K79 inhibits the activity of recombinant PCGF1-RING1B and PCGF4-RING1B in vitro.

The main conceptual advance of the manuscript is that, in addition to MLL-FPs inhibiting PRC1.1 binding to their target genes, H3K79 methylation also inhibits PRC1.1 activity. In other words, H3K79 modification contributes to the inhibition H2AK119ub1 deposition.

Additional comments.

Upon re-reading the manuscript, we noted several statements in the introduction that appear biased and should be revised for clarity and neutrality.

Lines 99-100: "Thus, it not generally thought that the MLL-FP drives leukaemia initiation by antagonising Polycomb."

MLL fusion proteins (MLL-FPs) inevitably antagonize Polycomb group proteins, as they drive transcription at Polycomb target genes. While alternative models have been proposed for how MLL-FPs function, their activity ultimately counteracts Polycomb-mediated repression. To our knowledge, no scientist has proposed that knockout of the Polycomb genes would have the same effect as the expression of MLL fusion proteins.

Lines 111-115: 'Although Menin and DOT1L inhibitors have shown specific anti-leukaemic effects, DOT1L inhibitors have been disappointing in the clinic thus far while Menin inhibitors are highly promising in early-phase clinical trials [25, 26]. While both potently downregulate similar target genes, this discrepancy is likely related to the fact that these inhibitors target different aspects of MLL-FP function [27, 28].'

This may be true; however, what is certain is that the first generation of DOT1L inhibitors were highly unstable and displayed relatively poor PK-PD profiles, which limited their potency and effectiveness inhibiting DOT1L in vivo. The authors should discuss this possibility. Notably, Scott Armstrong, a co-author on the manuscript who was involved in of developing and testing of both classes of inhibitors will be aware of these limitations.

Reviewer #3 (Remarks to the Author):

The authors have addressed most of my points and I am satisfied with the revised version of the manuscript, which has overall improved.

Version 4:

Decision Letter:

Dear Dr Gilan,

I am pleased to inform you that your manuscript, "DOT1L provides transcriptional memory through PRC1.1 antagonism", has now been accepted for publication in Nature Cell Biology. Congratulations!

Once your paper has been scheduled for online publication, the Nature press office will be in touch to confirm the details. An online order

form for reprints of your paper is available at <https://www.nature.com/reprints/author-reprints.html>. All co-authors, authors' institutions and authors' funding agencies can order reprints using the form appropriate to their geographical region.

Please note that *Nature Cell Biology* is a Transformative Journal (TJ). Authors may publish their research with us through the traditional subscription access route or make their paper immediately open access through payment of an article-processing charge (APC). Authors will not be required to make a final decision about access to their article until it has been accepted. [Find out more about Transformative Journals](https://www.springernature.com/gp/open-research/transformative-journals)

Authors may need to take specific actions to achieve compliance with funder and institutional open access mandates. If your research is supported by a funder that requires immediate open access (e.g. according to [Plan S principles](https://www.springernature.com/gp/open-science/plan-s-compliance) or the [NIH public access policy](https://www.springernature.com/gp/open-science/us-federal-agency-compliance)) then you should select the gold OA route, and we will direct you to the compliant route where possible. Because authors warrant under our subscription licensing terms that they haven't committed to licensing any version of their article under a licence inconsistent with the terms of our agreement – including the applicable embargo period – publication under the subscription model isn't suitable for authors whose funders require no embargo.

If you have not already done so, we strongly recommend that you upload the step-by-step protocols used in this manuscript to protocols.io (<https://protocols.io>), an open online resource that allows researchers to share their detailed experimental know-how. All uploaded protocols are made freely available and are assigned DOIs for ease of citation. Protocols and Nature Portfolio journal papers in which they are used can be linked to one another, and this link is clearly and prominently visible in the online versions of both. Authors who performed the specific experiments can act as primary authors for the Protocol as they will be best placed to share the methodology details, but the Corresponding Author of the present research paper should be included as one of the authors. By uploading your Protocols onto protocols.io, you are enabling researchers to more readily reproduce or adapt the methodology you use, as well as increasing the visibility of your protocols and papers. You can also establish a dedicated workspace to collect your lab Protocols. Further information can be found at <https://www.protocols.io/help/publish-articles>.

Nature Cell Biology encourages authors presenting evidence for cell, biological, molecular, and genetic interactions to consider communicating these findings using Biofactoid (<https://biofactoid.org/>). This tool helps users share a searchable representation of interactions (e.g. binding, gene expression, post-translational modification) between genes, gene products, or chemicals. Information added to Biofactoid, with author attribution, is shared on social media and public databases, such as Pathway Commons, where it can be discovered and analyzed in the context of a large and growing corpus of knowledge.

With kind regards,

Sabrya.

Sabrya Carim, PhD
(she/her/hers)
Senior Editor, Nature Cell Biology
Nature Portfolio

Springer Nature
The Campus, 4 Crinan Street, London N1 9XW, UK
sabrya.carim@springernature.com
<https://orcid.org/0000-0001-9485-1938>

** Visit the Springer Nature Editorial and Publishing website at http://editorial-jobs.springernature.com?utm_source=ejp_NCB_email&utm_medium=ejp_NCB_email&utm_campaign=ejp_NCB for more information about our career opportunities. If you have any questions please click [here](mailto:editorial.publishing.jobs@springernature.com).

Author Response to Reviewers, NCB

Editor Sabrya Carim

Your manuscript, "DOT1L functions as a transcriptional memory component of the MLL-Polycomb axis", has now been seen by 3 referees, who are experts in cancer and genome integrity (referee 1); epigenetics and cancer (referee 2); and PRC in cancer (referee 3). As you will see from their comments (attached below) they find this work of potential interest, but have raised substantial concerns, which in our view would need to be addressed with considerable revisions before we can consider publication in Nature Cell Biology.

We would like to thank all 3 reviewers for their constructive comments about our manuscript. We have performed extensive revision experiments and additional analyses to respond to their concerns.

Reviewer #1: Remarks to the Author:

The authors sought to investigate how Menin and DOT1L contribute to MLL-FP function. They made some important findings through CRISPR screens, CRISPR knockouts (KOs), inhibitor treatments for Menin and DOT1L, and measuring various chromatin modifications and modifiers, such as the Polycomb Repressive system targeting MLL-FP targets and the mutual exclusion of H3K79me2 and H2AK119Ub. The authors also performed experiments to test the prevailing model in Polycomb repression. With solid evidence, they convincingly showed the effects of Pcgfl and BCOR KO on Menin and DOT1L inhibition in various cell lines. I find this work novel, exciting, and a valuable contribution to the field of cancer epigenetics

We are delighted that the reviewer finds this work novel, exciting and a valuable contribution to the field of cancer epigenetics. We thank the reviewer for their time in reviewing our work and for their expert insights and suggestions, which have helped to strengthen the manuscript. We have provided a point-by-point response to their specific comments below:

I do have specific comments for the authors to address.

- The authors should evaluate their claims regarding PRC1.1's role in their system as follows:
 - When assessing the role of PRC1.1, they measure H2AK119Ub. However, measuring H2AK119Ub does not necessarily indicate PRC1.1 activity; it is a measurement of RING1B activity (subunit common to all PRC1 complexes).

The reviewer makes a good point that H2AK119ub does not specifically read out PRC1.1 activity, but the activity of all PRC1 complexes. We focused on PRC1.1 for two reasons: 1) Previous studies have demonstrated that the majority of H2AK119ub at CpG islands is deposited by the PRC1.1 complex [1] and 2) knockout of PCGF1 prevented the global drug-induced increase in H2AK119ub levels (**Fig. 6a-c, Extended Data Figure 6a**). Together, this demonstrates that PRC1.1 is the primary complex contributing to the phenotypes observed in the manuscript.

- In analyzing the role of PRC1.1 in repressing MLL-FP targets, they find that KDM2B shows little to no increase at MLL-FP targets (see Fig. 4e). However, they show that H2AK119Ub undergoes relatively more significant changes (Fig. 4e). This suggests that PRC1.1 is not the only complex contributing to the silencing of MLL-FP target genes. Other subcomplexes may be involved in this silencing. This possibility should be mentioned and, if possible, further explored, as previous data attribute this effect to PRC1.1.
- When assessing the role of PRC1.1 in repressing MLL-FP targets, why analyze all PRC1 complexes by ChIPing RING1B? Why not ChIP BCOR or Pcgfl?

Here, the reviewer correctly points out that there is a discrepancy between occupancy of KDM2B and H2AK119ub levels after drug treatment, suggesting that either additional PRC1

complexes are regulating H2AK119ub deposition, or PRC1.1 activity is being regulated downstream of its recruitment.

One critical aspect of PRC1 function is that PRC1 recruitment can be uncoupled from its enzymatic activity. RING1B has little enzymatic activity in *in vitro* nucleosome ubiquitination assays and requires PCGF proteins for maximum activity [2]. Thus, the PCGF proteins play a critical role in mediating the ubiquitin ligase activity of PRC1 through activating PRC1 activity, independent of recruitment.

In the original manuscript we presented ChIP-seq data for the PRC1.1 subunit, KDM2B, in control or following drug treatment and RING1B ChIP-seq at baseline only. Our intention was to use KDM2B as a surrogate for PRC1.1 occupancy. However, considering the advice from the reviewer, we have performed additional ChIP-seq experiments for BCOR and RING1B in control (DMSO), Menin or DOT1L inhibitor treated MLLAF9 cells to better define the relative contribution of canonical vs non-canonical PRC1 complexes to H2AK119ub. Following drug treatment, BCOR recruitment was largely similar to KDM2B at MLLAF9 target genes (**Extended Data Fig. 6c**). In addition, global changes in RING1B and BCOR levels after DOT1L inhibition did not correlate with baseline levels of H3K79me2, suggesting that H3K79me2 does not function to regulate recruitment of PRC1.1 (**Extended Data Fig. 6e**). Interestingly, there was a very striking induction of RING1B occupancy at a small subset of MLL-FP target genes such as *Meis1* and *Jmjd1c* (**Extended Data Fig. 6c**). This was most notable at genes where a potent increase in H3K27me3 also occurs, which is likely due to triggering a positive feedback loop that results in further recruitment of canonical RING1B/PRC1 [3] (**Extended Data Fig. 6d**).

These data support the notion that instead of regulating PRC1.1 recruitment, H3K79me2 plays an important role in regulating PRC1.1 activity.

We have now also analysed structural data of RING1B-PCGF4 (canonical PRC1) complex bound to a nucleosome [4]. This analysis revealed that PCGF4 (BMI1) forms contacts with H4E74 and H3D77, which coincidentally, is two residues away from H3K79 (**Extended Data Fig. 9a-b**). Alignment of PCGF1 and PCGF4 as well as structural modelling and AlphaFold3 analysis (**Extended Data Fig. 9a-c**), suggests that these same salt bridges cannot be formed in the context of PCGF1. Instead, based on the positions of the residues, PCGF1 could, in principle, form a salt bridge with H3K79 to facilitate enzymatic activity, which may be altered due to structural rearrangement by the presence of di-trimethylation (**Extended Data Fig. 9a-c**). This modelling strongly reinforces the notion that H3K79me2 disproportionately impacts PRC1.1 activity, rather than recruitment.

To further support this hypothesis, we performed rescue experiments where we have mutated the putative PCGF1 residue that is likely to be required for contact with H3K79me (**Fig. 7n-o and Extended Data Fig. 9d-g**).

• For the sgRNA screen, how is it that both UTX and PRC2 are identified as important for growth advantage? Shouldn't these have opposing effects?

This is an interesting point and relates to the paradoxical observation that upon perturbation, opposing epigenetic regulators can converge on a similar phenotype. To address this question,

we validated the effects of depleting KMT2D and UTX on the efficacy of DOT1L and Menin inhibition using sgRNA competition assays. The results indicated that depletion of KMT2D and UTX clearly conferred resistance (**Extended Data Fig. 1f-g**).

To understand the molecular basis for the differences between UTX and PRC1.1, we performed RNA-seq in control, KMT2D KO, or UTX KO cells treated with either DMSO, Menin, or DOT1L inhibitor. These data have been made available as a reviewer only figure (**Reviewer Fig. 1**).

Analysis of key downregulated genes (MEIS1, ZNF521 etc) upon drug treatment revealed that their downregulation was unaffected by either UTX or KMT2D depletion, suggesting that resistance is conferred via a PRC1.1/PRC2-independent mechanism (**Reviewer Fig. 1**). Importantly, these findings are also consistent with a recent report on the role of the UTX/MLL3/4 complex in mediating the efficacy of Menin inhibition [5].

• The authors observe that the three cell lines studied (MOLM13, MV4;11, Murine MLLAF9) have exceedingly rare overlaps in differentially expressed genes. While they focus on Meis1, an important gene with overlap between all cells, they don't discuss why there is such little general overlap. Can the authors clearly explain this in the text and be consistent between different experiments with the different cell lines?

Apologies if this wasn't sufficiently clear. The Venn diagram referred to by the reviewer is in fact depicting the genes that are rescued by PRC1.1 loss, not simply genes that are downregulated in response to treatment. We have added labelling within the figure to make it clearer that this is referring to genes that are specifically rescued by PRC1.1 loss rather than an overlap in differentially expressed genes.

• Why aren't apoptosis measurements discussed in Figure 3F presented alongside Figure 1? It seems it would make more sense since the authors are discussing cell survival.

Thank you for this suggestion, we have moved the apoptosis measurements to Figure 1 to increase the clarity of the phenotypic data presented.

• "Remarkably, H3K79me2 loss and baseline levels were highly correlated ($r=0.51$ and $r=0.66$, respectively) with H2AK119ub induction after SGC0946 treatment (Fig. 5d)." A correlation of 0.51 is not considered high, but 0.66 is on the border of high correlation. It is an overstatement to say there is a high correlation.

The reviewer is correct that we overstated the degree of correlation, we have changed the wording to say, "well correlated" instead of "highly correlated", which is hopefully more accurate. It is difficult to accurately quantify changes in histone modifications, particularly for modifications with broad and complex patterns, such as H3K79me2 and H2AK119ub. The approach we took in the original version of the manuscript involved assessing the levels at the TSS +/- 5kb, which is limited in its ability to accurately capture the changes at each gene. Perhaps more importantly, the correlations are much higher than those observed with

transcriptional changes or other histone modifications such as H3K9ac where the r values observed were 0.29 and 0.36, respectively (**Extended Data Fig. 5c**).

Figure Comments: • The cell nucleus in Figure 1G contains information that is uninterpretable unless highly zoomed in.

Apologies for including this, it was an oversight and has now been removed.

• In Figures 2B and C, it is hard to distinguish between PCGF1 KO and BCOR KO. Use different colors or data representation, such as dotted lines with no fill.

We agree with the reviewer and have made the necessary changes to make these figures easier to interpret.

Reviewer #2: Remarks to the Author:

Menin and DOT1L inhibitors are in clinical trials for the treatment of AML. The authors have addressed the mechanism by which Menin and DOT1L as part of an MLL-Fusion oncoprotein complex contribute to drive MLL-FP leukemogenesis. By performing CRISPR based screens they identified components of the non-canonical PRC1.1 complex, BCOR and PCGF1 as being required for a sustained response to the DOT1L and Menin inhibitors. Interestingly the MLL3/MLL4 COMPASS as well as the PRC2 complex were also identified as hits, suggesting either that these proteins function together (as we know for PRC1-PRC2), or that parallel – and not explained pathways – are involved in the stable silencing of MLL-FP target genes in response to Menin and DOT1L inhibition. Validation studies were performed both in vivo and in vitro, showing the requirement for PRC1.1 in mediating a phenotypic effect to DOT1L and Menin inhibitors. This was further extended to transcriptional studies. Moreover, in agreement with H3K79me2 being lost and transcriptional downregulation, the authors observed an increase in deposition of H2AK119ub1 and H3K27me3. Interestingly, temporary DOT1L disruption led to irreversible transcription changes at target genes, potentially supporting some level of transcriptional memory. Finally, an antagonistic link between DOT1L and PRC1.1 was observed in many different cell types, suggesting a general role of DOT1L in balancing active and repressive chromatin states.

MLL-fusion proteins are believed to induce leukemia through a mechanism that involves the recruitment of DOT1L and the super elongation complex. This leads to increased transcriptional elongation of target genes and the development of leukemia. Previous studies from Scott Armstrong's lab, studying the role of MLL-FPs in AML (using a model like the one used in the submitted manuscript), have shown (see e.g. Deshpande et al, 2014 PMID 25464900) that H3K79me2/me3 (catalyzed by DOT1L) correlate with HOXA gene expression, as well as exclusion of the repressive H3K27me3 modification at the HOXA7-10 locus. Moreover, Deshpande et al showed that deletion of AF10, component of the super elongation complex leads to impaired DOT1L association with HOXA cluster genes, loss of H3K79me3/me2, and the spread of H3K27me3, which coincides with a decrease in gene expression. Therefore, the previous studies from the Armstrong lab have established a strong correlation between MLL-FPs, DOT1L, H3K79me3/me2 and transcriptional elongation, and shown that this is anticorrelated with H3K27me3. Moreover, they have shown that DOT1L loss leads to increased H3K27me3 levels. Despite the strong relevance of the Deshpande paper and other studies from the Armstrong lab, the papers are not cited in the submitted manuscript. While the authors may be correct that 'it is still unknown how MLL-FP, Menin and DOT1L influence the function of the Polycomb complexes', the submitted manuscript does not provide real novel insight into this question. In fact, most of the manuscript establishes what we know, i.e. an anticorrelation between MLL-FP fusion proteins and the binding of Polycomb group proteins, and an anticorrelation between H3K79me2/me3 and H3K27me3. How DOT1L or H3K79me3/me2 prevent PRC1.1 binding is not established, and it is still likely that the Polycomb group proteins are first bound, once transcription has stopped. In other words, a well-established mechanism for maintaining transcriptional repression by the Polycomb group proteins.

We thank the reviewer for their critical evaluation of our manuscript and providing expert insight and comments, which have helped to strengthen the manuscript. However, we believe

that the reviewer appears to have missed the major aspects of our findings and consequently their comments about the lack of novelty are poorly substantiated. Our manuscript details a specific and conserved antagonistic link between DOT1L and PRC1.1 (not PRC2) that to our knowledge, has never been shown. Our data also strongly support the notion that the relative stability of H3K79methylation (compared to most active histone modifications) ensures that it persists to provide a delay in PRC1.1-mediated repression following activator removal (e.g. drug treatment). This is a major insight that to our knowledge has never been reported in the literature.

The comments by the reviewer are sharply contrasted with the views of the other two expert reviewers.

Reviewer 1 states: “I find this work novel, exciting, and a valuable contribution to the field of cancer epigenetics.”

Reviewer 3 states: “Remarkably, this mechanism appears to exhibit a broader, conserved impact beyond the hematopoietic system. While this concept presents intriguing insights into our understanding...”

Nevertheless, we believe that some aspects of the reviewer’s comments were constructive and of value. We have therefore undertaken extensive additional experiments to address the reviewer’s two main concerns. We demonstrate that inhibition of transcription is not sufficient to induce Polycomb-mediated repressive histone modifications, especially at active genes. We also provide evidence that H3K79me2 mostly influences the activity of PRC1/PRC1.1, rather than its recruitment to chromatin. These new data are described in detail in a point-by-point response to the reviewer’s comments below.

In fact, most of the manuscript establishes what we know, i.e. an anticorrelation between MLL-FP fusion proteins and the binding of Polycomb group proteins, and an anticorrelation between H3K79me2/me3 and H3K27me3.

Here, if we understand the reviewer correctly, they are concerned that the anticorrelations between MLL-FP/DOT1L and Polycomb have already been reported. Whilst it is true that at ‘baseline’ active and repressive modifications are generally anti-correlated, steady state conditions are insufficient to ascribe or suggest causality. Instead, perturbations provide the opportunity to understand the causal nature of correlative relationships i.e. what happens when the specific modifications are removed.

They are correct that the Armstrong lab had previously shown that HOXA genes are marked by H3K27me3 following DOT1L deletion or inhibition, and we agree that this work should have been cited and have now included these references in the manuscript. Nevertheless, there are two critical aspects that distinguish those findings from what we are reporting in our manuscript.

Firstly, those studies observed an increase in H3K27me3 after DOT1L inhibition at a highly selective and restricted set of genes (e.g. MEIS1 and HOXA9), despite a global loss of

H3K79me2. Therefore, a complete loss of H3K79me2 at thousands of active genes has no impact on H3K27me3 levels (**Extended Data Fig. 5b**). This demonstrated lack of correlation between H3K79me2/me3 and H3K27me3 following DOT1L perturbation suggests the absence of a direct causal relationship and the selective changes at HOXA genes is likely due to other factors. In contrast, our manuscript clearly reports a global increase in H2AK119ub following DOT1L inhibition alone, regardless of whether genes undergo transcriptional changes (**Fig. 5**).

Secondly, the above-mentioned studies did not ascribe a functional role to Polycomb repression in the efficacy of DOT1L inhibition. On the other hand, our data strongly indicates that PRC1.1 plays a key functional role in the repression of MLL-FP target genes upon inhibition of Menin or DOT1L. This demonstrates that PRC1.1 is involved in gene repression and that this repression is likely critical for efficient leukaemia cell differentiation and cell death.

To summarise, we have uncovered a clear and striking relationship between H3K79me2 and H2AK119ub, and comprehensively demonstrated that this interplay is not influenced by MLL1/MLL-FP occupancy or transcriptional changes.

...it is still likely that the Polycomb group proteins are first bound, once transcription has stopped. In other words, a well-established mechanism for maintaining transcriptional repression by the Polycomb group proteins.

The reviewer makes an important point about the role of transcriptional activity and Polycomb function which has been previously reported in the literature. While we accept that there has been reported links between Polycomb and transcription, we believe that our data convincingly demonstrates that transcriptional effects do not explain the observed increase in PRC1 activity.

In our original version of the manuscript, we specifically ruled out the possibility that loss of transcription is resulting in the observed increase in H2AK119ub. Genes that were equally downregulated by both DOT1L and Menin inhibition showed marked differences in H2AK119ub deposition, suggesting that rather than transcriptional changes, PRC1 activity is regulated by changes in H3K79me2 (**Fig. 5b**). This is strongly supported by the fact that only DOT1L inhibition was able to induce global H2AK119ub deposition across 5 different cell types, with the degree of deposition being correlated with baseline levels of H3K79me2 (**Fig. 5d, Fig. 7a-f, Extended Data Fig. 6a**). Importantly, this was not accompanied by a global increase in H3K27me3 (**Extended Data Fig. 5b**).

To further strengthen this causal link and address previous reports linking Polycomb to transcription, we undertook two actions. We re-analysed the data from Riising et al. (2014), to determine whether H3K27me3 is induced at active genes upon transcriptional inhibition. We also performed our own experiments using the transcriptional inhibitor, DRB, in two independent cell lines, mESC and K562. We generated ChIP-seq data for SUZ12, POL2, H3K27me3 and H2AK119ub in mESC and H3K27me3 and H2AK119ub for K562 cells treated with DMSO or DRB for 8hrs.

Analyses of the published data and our own experiments demonstrate the following:

- Triptolide or DRB treatment in mESC cells resulted in globally increased SUZ12 chromatin binding without a corresponding global increase in H3K27me3, suggesting that transcriptional inhibition results in PRC2 recruitment but is not sufficient to induce PRC2 activity [6] (**Reviewer Fig. 2a-b**).
- SUZ12 binding was equally increased at active and repressed genes, suggesting that local gene activity does not influence the increase in PRC2 chromatin recruitment [6] (**Reviewer Fig. 2c**).
- Whilst H3K27me3 was increased at a small subset of genes, those loci had prebound SUZ12 and relatively low levels of POL2 occupancy prior to treatment (**Reviewer Fig. 2d**).
- Most importantly, DRB treatment (8hr) in mESC and K562 cells was not sufficient to induce H2AK119ub or H3K27me3 deposition, despite globally increased SUZ12 and decreased POL2 levels (**Fig. 7i, Extended Data Fig. 8d, and Reviewer Fig. 3**).

In summary, our data strongly indicates that transcriptional inhibition alone is not sufficient to induce PRC1 or PRC2-mediated repressive chromatin modifications. Furthermore, the fact that depletion of PRC1.1 effectively prevents drug-induced H3K27me3 deposition, blunts the transcriptional downregulation of MLL-FP target genes, and rescues the survival and proliferation of leukaemia cells strongly supports a role for PRC1.1 beyond simply maintaining gene repression as the reviewer suggested.

Together, our observations cannot be explained by changes in transcription and are consistent with a functional antagonism between DOT1L and PRC1.

How DOT1L or H3K79me3/me2 prevent PRC1.1 binding is not established

This is a very important question that the reviewer makes and is indeed something that we had been considering. We now provide additional data as well as structural modelling to propose a model for how H3K79methylation prevents H2AK119ub deposition.

In the original manuscript we included ChIP-seq data for KDM2B in leukaemia cells treated with a DOT1L or Menin inhibitor, which revealed a clear increase in binding only at selected MLL-FP target genes. As mentioned by Reviewer 1, there was a noticeable discrepancy between drug-induced recruitment of PRC1.1 (KDM2B binding) and increased H2AK119ub levels, suggesting that either additional PRC1 complexes are regulating H2AK119ub deposition, or PRC1.1 activity is being regulated downstream of its recruitment.

To rule out the possibility that additional PRC1 complexes are causing the gain in H2AK119ub, we performed ChIP-seq for BCOR and RING1B after treatment. Consistent with our results for KDM2B, we did not observe a clear and widespread H3K79me2-level dependent increase in BCOR and RING1B occupancy after DOT1L inhibition, suggesting that H3K79me2 does not function to regulate recruitment of PRC1.1 (**Extended Data Fig. 6e**).

One critical aspect of Polycomb biology is that recruitment can be uncoupled from enzymatic activity. In PRC1, RING1B has little enzymatic activity in *in vitro* nucleosome ubiquitination assays and requires PCGF proteins for maximum activity [2]. Thus, the PCGF proteins play a critical role in mediating the ubiquitin ligase activity of PRC1, independent of recruitment. In fact, we and others show that the PRC1.1 complex is often already pre-bound at the promoters of active genes prior to drug treatment (**Extended Data Fig. 6c**) [7, 8]. This observation together with the lack of correlation between PRC1.1 subunits and H3K79me2 (baseline levels or DOT1L inhibitor-loss), is consistent with the notion that loss of H3K79me2 disproportionately impacts the activity of PRC1.1 downstream of its recruitment (**Extended Data Fig. 6e**).

To identify a potential explanation for how H3K79me2 could influence PRC1 activity downstream of recruitment, we explored previous structural data of a PRC1 (RING1B-PCGF4) complex bound to a nucleosome [4]. Modelling this interaction reveals how PRC1 recognizes multiple histone surfaces to achieve substrate specificity. Specifically, we observed that PCGF4 K62 and R64 residues form salt bridges with H4E74 and H3D77, respectively, and that D77 is two residues away from H3K79 (**Extended Data Fig. 9a-b**). Importantly, alignment of PCGF1 and PCGF4, and structural modelling using AlphaFold3 suggest that these same salt bridges cannot be formed by PCGF1, where the positively charged PCGF4 K62 is predicted to be the negatively charged PCGF1 E91 and the positively charged PCGF4 R64 is predicted to be the polar PCGF1 Q93 (**Extended Data Fig. 9a-c**). Therefore, we reasoned that the PCGF1 E91 residue could in principle form a salt bridge with H3K79, which may be altered due to di-trimethylation of H3K79. This provides a plausible mechanistic model for how H3K79me could regulate the activity of PRC1.1.

To test this hypothesis, we performed rescue experiments with PCGF1 using either a wild type PCGF1 sequence (WT PCGF1) or an E91K PCGF1 mutant (MUT PCGF1) and over-expressed them in our PCGF1 KO or parental MOLM13 cells (**Fig. 7n, Extended Data Fig. 9a-c**). We included an empty vector as our negative control and western blot analysis showed equal expression of both WT and mutant PCGF1 (**Extended Data Fig. 9d**). Interestingly, rescue of WT PCGF1 expression in our PCGF1 KO cells completely restored sensitivity to both Menin and DOT1L inhibition while the effect of the drugs in the MUT PCGF1 was reduced compared to WT (**Fig. 7o**). In addition, overexpression of WT PCGF1, but not MUT PCGF1, in parental MOLM13 cells increased their sensitivity to both inhibitors (**Extended Data Fig. 9e**). Together, these analysis and data support the notion that since PRC1.1 is often pre-bound at the chromatin of active genes, its catalytic activity depends on direct contact with specific residues on the H3 surface, which may be inhibited by the presence of H3K79methylation.

Other comments:

1. Several hits which were identified in CRISPR survival screen included BCOR, PCGF1 (PRC1.1), KMT2D and NCOA6 (MLL3/MLL4 COMPASS complex) and EED, EZH2, SUZ12 and JARID2 (PRC2). It would be great to understand how PRC2 was identified as one of the hits in survival screen but turned out to be essential for cell viability. This maybe important to understand/address a success/validity of CRISPR screen in these cell lines.

The notion of gene essentiality refers to the gene's requirement for cell viability or proliferation. This is ultimately a continuum and not a binary description. Some genes may be extremely important for viability or proliferation, which means that their knockout results in the complete cessation of cellular proliferation or the induction of apoptosis. On the other hand, many genes have varying degrees of influence over cellular proliferation, in which case they will gradually drop out over time in a CRISPR screen. PRC2 subunits were only enriched in the screen at an early timepoint (day 14) in drug but were lost at later timepoints (now shown in **Fig. 1b**). This is likely due to the gradual depletion of sgRNAs against PRC2, which is supported by published data [9] and DepMap analysis of EZH2 and EED dependency in MLL-FP AML cell lines (**Extended Data Fig. 1j-k**).

2. The authors used BCOR, PCGF1 or double KO cells throughout the study and claim that depletion of PRC1.1 confers resistance to DOT1L or Menin inhibitors in leukemic cells. It would be interesting to test if knockout of other PRC1.1 complex members, such as KDM2B leads to as similar phenotype.

As requested by the reviewer, we have performed knockout of other PRC1.1 subunits to test if they also confer resistance to Menin and DOT1L inhibition. Similar to the modest enrichment of KDM2B and RING1B observed in the CRISPR screens, we found that knockout of RING1B resulted in modest but clear resistance to both DOT1L and Menin inhibition, while KDM2B depletion only conferred resistance to DOT1L inhibition. This data is now included in the revised manuscript (**Extended Data Fig. 1d**). Moreover, our findings are in agreement with a recent publication which also uncovered several subunits of the PRC1.1 complex in a CRISPR screen as being required for Menin inhibitor efficacy [5].

The authors should also test reversibility of the phenotypes and rule out clonal effects by re-introducing BCOR/PCGF1 in the KO cells. Chromatin analysis should be performed to establish if re-establishment of PRC1.1 can restore ubiquitination and hence H3K27me3 levels, and subsequently down-regulate transcription of key MLL target genes.

The reviewer makes a good point about the importance of performing rescue experiments to confirm that the KO phenotypes are not due to clonal effects. This is something that we always consider when performing knockouts and studying gene function. In part, we circumvent these concerns by employing multiple sgRNAs per gene and performing experiments with polyclonal KO cell populations to prevent clonal effects. Nevertheless, we carried out rescue experiments in our MOLM13 PCGF1 KO cells as it was also an opportunity to test the hypothesis that PCGF1 engages in direct interactions with the nucleosome surface around H3K79. These results confirm that rescue of PCGF1 expression completely restored sensitivity to Menin and DOT1L inhibition and the drug-induced downregulation of MEIS1 expression

as well as the accompanying induction of H2AK119ub and H3K27me3 (**Extended Data Fig. 9f-g**).

3. Previous studies by van den Boom et al 2016 (PMID 26748712) demonstrated an essential role for non-canonical PRC1.1 in leukemic cells. This conflicts with results reported in this study. Noticeably, both studies used MLL-AF9 cells. The authors should discuss potential explanations for these discrepancies.

We are indeed aware of these studies and have considered some potential explanations for the apparent discrepancies. The study by Van de Boom et al. mainly focuses on the effects of KDM2B depletion. While KDM2B is a bona-fide subunit of the PRC1.1 complex, there are many studies that have shown that it also has PRC1.1-independent functions, which could be contributing to the phenotypes observed [10-12]. They also extensively validate the RING1A and RING1B subunits which are the core catalytic subunits shared by all PRC1 complexes and are not PRC1.1 specific. Furthermore, data from the CRISPR database, DepMap, shows that the specific PRC1.1 subunits, BCOR and PCGF1, are not essential in AML cell lines. In contrast, knockout of these subunits confers a survival advantage in several leukaemia cell lines (e.g. MOLM13 and MV4;11) (**Extended Data Fig. 11**). Finally, a potential explanation for the discrepancy could also be due to the extensive use of shRNAs, which are known to have off-target effects [13]. In support of our findings, a recent study has also uncovered PRC1.1 subunits in a CRISPR screen for Menin inhibitor efficacy in MLLAF9 cells [5].

4. The authors claim a potential role for DOT1L in transcriptional memory. However, the question is whether this is due to a slow turnover of H3K79me2 in response to the Menin inhibitor. Alternatively, the results could be explained by that transcriptional inhibition is required for the recruitment of the Polycomb group proteins, which agrees with the widely accepted concept that the Polycomb group proteins do not initiate transcriptional repression but are required for maintaining transcriptional repression.

As mentioned in the earlier responses, our data suggests that the reviewer is only partially correct here. It is true that the maintenance of repression is largely orchestrated by PRC2 which is efficiently recruited once transcription has stopped. However, our data supports the view that PRC1.1 is actively involved in initiating gene repression and its activity is antagonised by DOT1L-dependent H3K79methylation.

The model of Polycomb-dependent maintenance of gene repression is largely attributed to PRC2, not PRC1. The hierarchical model of PRC1 and PRC2 recruitment now suggests that PRC1 and H2AK119ub occupancy can be achieved without PRC2 and that non-canonical PRC1.1-mediated H2AK119ub is required for recruitment of PRC2 (via JARID2) and deposition of H3K27me3 [1, 7]. The established dogma is that following PRC2-mediated H3K27me3, recruitment of canonical PRC1 occurs as part of a positive feedback loop to further reinforce and maintain the repressed state [3]. This is now supported by our additional data showing a dramatic increase in recruitment of RING1B, but not KDM2B or BCOR, to regions that gain H3K27me3 after treatment (**Extended Data Fig. 6d**). For selected genes that were

potently downregulated following Menin inhibition and that had also lost H3K79me2 following Menin inhibition (e.g. Meis1, JMJD1c), H3K27me3 was substantially increased only after the deposition of H2AK119ub (**Fig. 5j, Fig. 6b-c, Extended Data Fig. 4d**).

Therefore, while PRC2 is recruited to chromatin at genes that are silenced following Menin inhibition, the loss of activators (e.g. MLL-FP, MLL1) initially results in reduced transcription as well as eviction of DOT1L (via MLL-FP) leading to the selective loss of H3K79methylation. Consequently, PRC1.1-dependent H2AK119ub is deposited to further reduce transcription, silence the gene and serve as a platform for the efficient recruitment of PRC2 [1].

Ultimately, our data effectively elucidates the order of events during gene repression and untangles the distinct functions of DOT1L, PRC1.1 and PRC2 in the regulation of gene repression. DOT1L selectively restricts PRC1.1 activity by maintaining H3K79me2 such that loss of this modification enables PRC1.1 to repress genes via H2AK119ub. Subsequently, sufficient gene repression is then necessary to facilitate PRC2 recruitment and activity to maintain repression.

Since PRC2 activity is only increased at a small subset of genes following DOT1L inhibition, an interesting aspect of this model is that genes can gain H2AK119ub without a subsequent gain of H3K27me3 (**Extended Data Fig. 5b**). Therefore, most genes with an increase in H2AK119ub after DOT1L inhibition are not altered transcriptionally (**Fig. 7j, Extended Data Fig. 8f**). We therefore hypothesized that the increased H2AK119ub may function to prime genes for future repression. To address this, we took advantage of a non-leukaemia model of differentiation to determine whether DOT1L inhibitor pre-treatment influences cellular differentiation. Using the primary human immortalised erythroid progenitor cell line (HuDEP-2), we pre-treated cells with either DMSO or DOT1L inhibitor (SGC0946) for 7 days and induced their differentiation to erythrocytes over a subsequent 7 days. Through assessment of an established erythroid differentiation marker (CD49d), we found that while SGC0946 pre-treatment did not impact baseline differentiation state or cellular viability, it accelerated the induced differentiation of HuDEP-2 cells (**Figure 7k-m, Extended Data Figure 8g-h**). These results support the idea that H3K79methylation may function to regulate the rate of differentiation by controlling the rate of gene repression during differentiation. In agreement with this, RNA-seq analysis reveals that a subset of genes ‘primed’ for repression (increased H2AK119ub after DOT1L inhibition prior to differentiation), undergo greater silencing during differentiation compared to DMSO treated cells (**Extended Data Figure 8i**).

Minor points:

1. Genome-wide and epigenetic focused CRISPR-screens were performed in both human and murine leukemia cells, but the manuscript only focuses on the epigenetic factors. For completion, the authors should provide some discussion on their findings from genome-wide CRISPR screen or some rationale behind for why needing this extra data.

We agree with the reviewer that this should be discussed. Just for clarification, the genome-wide CRISPR screen specifically uncovered epigenetic factors in mediating resistance to DOT1L inhibition. There were surprisingly few hits falling under the category of non-

chromatin associated. Follow up CRISPR screens were then performed with an epigenetics-focused library in murine MLLAF9 cells using both Menin and DOT1L inhibitors to validate the findings from the whole genome CRISPR screen (**Extended Data Fig. 1b-c**). To improve the coherence and provide justification for the genome-wide screen, we have included additional explanations in the results section.

2. For having a complete data set the authors may want to generate a BCOR KO MLL-AF9 cell line and compare it to a PCGF1 KO MLL-AF9 cell line in extended data figure 2a).

Thank you for this comment, we agree that this was missing from the original manuscript. We have now added data to the revised manuscript for BCOR knockout in murine MLLAF9 cells. It confirms the requirement of BCOR for DOT1L and Menin inhibitor efficacy, similar to PCGF1 and what we also observed in the human leukaemia cell lines (**Extended Data Fig. 1e**).

Reviewer #3: Remarks to the Author:

This manuscript explores critical epigenetic events within the context of MLL fusion protein-related leukemias. Specifically, it investigates the role of DOT1L as a primary determinant in mitigating Polycomb associated repression. Remarkably, this mechanism appears to exhibit a broader, conserved impact beyond the hematopoietic system. While this concept presents intriguing insights into our understanding of molecular mechanisms driving oncohematological transformation, the experimental findings would benefit from a more systematic and transparent presentation to robustly substantiate the authors' conclusions. Please find below our main concerns regarding each section/figure:

We thank the reviewer for their time in reviewing our work and their expertise and are pleased that they consider the work to provide intriguing insights into our understanding of molecular mechanisms driving hematopoietic malignancies and its broader applicability. We have provided a point-by-point response to their comments, which have been very helpful in improving the manuscript.

Figure 1

Authors should make an effort to explain the CRISPR-based screening of Figure 1. Please provide the details of the cell lines used and the design of the experiments in a way that it is easy to follow for the readers. For example, they mention the usage of two libraries, but in Methods there is just description of a chromatin library. Also, we could not find the data relative to the screening with the VTP50469 inhibitor in MV4;11 cells.

Thank you for this comment, we have now rewritten the description of the CRISPR screens to improve readability. We apologise for not including the whole genome human CRISPR library in the methods section in the original version. For clarification, the whole genome CRISPR screen was performed ONLY with the DOT1L inhibitor, while the chromatin-focused CRISPR screens were performed with both DOT1L and Menin inhibitors. We have now made this much clearer in the text and figures.

For how long was the experiment carried out? In the scheme of Figure 1a, 12 and 24 days are mentioned, but in the Extended Data Figure 1a and 1b it is written "13 days". The initial results section, which forms the basis for the rest of the manuscript, should be enhanced and presented systematically for better clarity and coherence.

Apologies again for the confusion and lack of clarity around the experimental workflow of the CRISPR screens. We have now rectified this in the results section of the revised manuscript. In brief, as mentioned earlier, the whole genome CRISPR screen was performed in MV4;11 cells treated with the DOT1L inhibitor with samples collected at day 12 and day 24. In the murine MLLAF9 epigenetic-focused CRISPR screens, the time point analysed was day 13. We have removed the time points from the schematic and instead included them in the figure legend for each relevant plot.

Validation of the screening results are also difficult to follow. Why when depicting results in MV4;11 cells the % of sgRNA is shown, but the results in MOLM13 refer to cell number? And why the authors validate the data in MOLM13 cells? But then in the supplementary material it seems the opposite (the cell lines used in Supp figures 1e and 1f are not indicated). Also, is there any validation in murine cells?

Thank you for these comments and helping us to improve the clarity. We apologise that this is confusing, and in part it is due to subjecting several cell lines to a variety of different assays for validation. Firstly, % of sgRNA refers to sgRNA competition assays which involve mixing uninfected cells with sgRNA-infected cells and subjecting them to a growth 'competition' over time in the presence of drug. This is commonly used as a validation tool for these types of CRISPR screens and was performed in MV4;11 cells. Whilst competition assays are useful, they do not provide a sense for how well the KO cells are able to proliferate in the presence of drug. Therefore, we also performed a series of proliferation assays in both MOLM13 and MV4;11 cells which provide a much more complete picture of the effect of gene KO on drug response. Due to space constraints, the MOLM13 competition assays were not presented in the manuscript and instead, proliferation assays were used, which are more informative.

Apologies for the error in not labelling the proliferation assay plots (now in **Extended Data Fig. 1h-i** with the name of the cell line). We have moved the MV4;11 proliferation assays into the main figure (**Fig. 1f**) and the MOLM13 proliferation assays to the Extended Data Figure to keep things consistent.

We have also performed validation in the murine MLLAF9 cells (**Fig. 1i, Extended Data Fig. 1n-o**). As also requested by Reviewer 2 and for completeness, we have also included competition assay validation for BCOR and PCGF1 in the murine MLLAF9 cells (**Extended Data Fig. 1e**).

The experiment depicted in Fig 1g is difficult to follow. It is also unclear why the vectors used for the sgRNAs targeting PCGF1 and BCOR contains BFP.

Thank you for pointing this out, we have corrected this and updated the figure schematic. The PCGF1 and BCOR sgRNA lentiviral plasmids contain a BFP fluorescent reporter so our PCGF1 and BCOR KO cells are BFP+. However, the competition assay assessing DOT1L sgRNA depletion was done using the GFP fluorescent reporter contained in the DOT1L-sgRNA plasmid.

The fact that the double KO is conferring survival specifically during Menin and DOTL1 inhibition, but not in treatments with IBET-151 or ABT-199, is an interesting finding that and it should be included in the main figure.

Thank you for this suggestion, we agree that this is an important finding and should be included in the main figure and we have now moved it in the revised manuscript (**Fig. 1j**).

Figure 2

Why BCOR KO is not shown in the Murine cell line MLLAF9 (Extended Figure 2e)?

We originally did not validate BCOR in the MLLAF9 cells as its enrichment in the CRISPR screen was relatively low compared to PCGF1. However, for completeness, we have now validated it and have included this in the revised manuscript (**Extended Data Fig. 1e**).

Given that PRC1 functions as a repressor complex, it would be important to determine whether the behavior illustrated in Figure 2d is also observed for up-regulated genes.

This is a good question and in part something that is addressed in Figure 2b. Given that Menin and DOT1L are involved in gene activation, their inhibition initially results in the downregulation of specific genes. Upregulated genes are thus likely to be simply a result of indirect secondary effects following the initial gene downregulation. We thus also observe that upregulated genes are induced to a lesser degree in the KO cells. This data is now included in the revised manuscript (**Extended Data Fig. 2f**).

Figure 3

With respect to the CellRadar RNA-seq panels, it is unclear whether the gene-set selected upon PCGF1 KO are in the presence of inhibitors. No analysis has been shown before on the effect of transcriptomics of PCGF1 KO without MLL-AF inhibitors. The Cell Radar RNA-seq analysis should be performed upon treatment with MLL-AF inhibitors AND PCGF1 KO.

The genes in the CellRadar plots are those upregulated by treatment alone in control cells or those upregulated by PCGF1 KO alone, without treatment. This was intended to show the cell-type specific transcriptional programs that are activated by each perturbation (drug or KO).

The reviewer is correct that Cell Radar analysis on drug treatment in the PCGF1 KO would be interesting to perform, the issue is that Cell Radar merely uses gene lists and does not consider the level or direction of transcriptional changes. Given that PCGF1 KO only blunts the transcriptional response to treatment (**Fig. 2b**), treatment of the PCGF1 KO cells would have the same list of downregulated genes as the control cells.

Upon analyzing the cytometer data, it appears that the loss of PRC1.1 alone has a significant impact on AML cells, as demonstrated in the CellRadar panels. However, this effect does not extend to markers such as Ly6G. How do the authors explain this?

This is an interesting point by the reviewer and something we also noticed. We can only speculate that since Ly6G expression is already quite low in these cells, a further reduction in expression due to PCGF1 KO cells being more 'de-differentiated' is not possible. However, the RNA-seq shows that many stemness genes are derepressed in the PCGF1 KO cells, which is reflected in the CellRadar data. This is consistent with the FACS data suggesting the cells are taking on a more immature state, however, this is not specifically reflected by Ly6G expression.

Why transplants in mice are performed with human cells lines? Why not using the murine MLLAF9?

The reasons we had chosen to use the human cell line for the *in vivo* experiments are threefold. Firstly, as the PCGF1 KO cells are derived from clones, we were concerned about clonal effects manifesting *in vivo*, especially given the murine MLLAF9 cells are generated from a virally integrated MLLAF9 oncogene. Secondly, the MOLM13 BCOR/PCGF1 double KO cells showed a more dramatic phenotype; they are polyclonal and strongly resistant to Menin inhibition *in vitro*. Thirdly, the survival benefit of the SNDX chow (0.1% SNDX) in the murine MLLAF9 leukaemia model is not substantial and may hamper the readout of a resistant phenotype.

In transplant experiments, PRC1 KO alone does not cause MOLM13 to be more malignant, this seems counterintuitive taking into account the Cell Radar results. Please comment on this.

This is a valid point by the reviewer, and we can see why that would be confusing. It is worth noting, however, that the de-differentiation program (based on CellRadar) was largely seen in the murine MLLAF9 cells. In the MOLM13 cells, the CellRadar analysis did not clearly point to a de-differentiation state. We can't be certain of the reason for this, but we can speculate that perhaps MOLM13 cells don't maintain the self-renewal stem cell-like hierarchical structure that primary leukaemia cells have. Nevertheless, this was surprising to us given that the PCGF1/BCOR dual KO cell line proliferates significantly faster than the control cells and was expected to have a shorter disease latency. The fact that it did not, points to important differences between *in vitro* and *in vivo* survival and proliferation of cells.

To help alleviate the reviewer's concern, we also performed RNA-seq on CD45+ cells purified from bone marrow samples collected from mice sacrificed at day 18 to understand the transcriptional changes that occur *in vivo*. Interestingly, cell radar analysis of upregulated genes in *in vivo*-derived PRC1.1 KO MOLM13 cells revealed a more immature phenotype compared to *in vitro*-derived cells (**Fig. 3i**). Furthermore, intersecting the drug-induced transcriptional changes shows that the target genes of *in vivo* treated MOLM13 cells are highly similar to the direct targets of MLLAF9 in a primary human leukaemia model of CD34+ MLLAF9 transformed cells [14] (**Figure 3i, Extended Data Fig. 3g**).

Indicate the number of mice used in the study directly in the figure or in the main text.

Thank you, we have now added this directly to the figure.

Figure 4

Since the ChIP experiments were performed using murine cells, it would be important to validate the results in human cells. To properly quantify the ChIP-seq data the authors should use spike-in controls.

This is an important suggestion by the reviewer. In the original version of the manuscript, we had largely focused on the murine MLLAF9 cells for interrogation of the chromatin changes

since we could directly assess MLLAF9 occupancy (via the FLAG-tag). Moreover, we had previously generated extensive data on direct MLLAF9 target genes using degrons coupled with nascent transcriptomics [14, 15]. It is important to note, in the original version of the manuscript, we had also included ChIP-seq in two human (non-leukaemic) cell lines, K562 and HuDEP-2, to demonstrate the global antagonistic relationship between H3K79me2 and H2AK119ub.

Nevertheless, to confirm that similar changes occur in human leukaemia cells, we performed ChIP-seq for H3K79me2, H3K27me3 and H2AK119ub in control and BCOR/PCGF1 dual KO MOLM13 cells treated with either DMSO, Menin or DOT1L inhibitor. The results confirmed the selective and potent induction of PRC1.1 and PRC2 at canonical MLLAF9 target genes such as MEIS1 and JMJD1C following Menin inhibition (**Extended Data Fig. 4e**). Importantly, DOT1L inhibition in control cells also led to a clear and broad increase in H2AK119ub at genes with a high baseline H3K79me2 levels that was completely lost in the BCOR/PCGF1 KO cells (**Extended Data Fig. 6a-b**). Taken together, these ChIP-seq data are consistent with the data in murine MLLAF9 cells and support the importance of Polycomb-mediated repression in the clinical efficacy of Menin inhibitors.

At the request of the reviewer, we also repeated our ChIP-seq experiments in murine MLLAF9 cells using drosophila S2 spike-in chromatin to control for global changes and enable accurate data normalisation. We assessed both ChIP-Rx analysis of H2AK119ub as well as non-normalised ChIP-seq analysis, which clearly shows that a clear increase occurs after DOT1L inhibition, with an even greater increase after ChIP-Rx normalisation (**Extended Data Fig. 4c**). This data is now included in the revised manuscript.

After checking the global occupancy of MLL proteins, for histone modifications the authors selected a subset of genes to continue their analysis. How this subset is selected should be clearly stated in the main text and not only in the figure caption. For example, it is interesting that from the top-100 genes bound by MLL, only 38 are significantly downregulated upon inhibitors treatment. What is the effect of the treatment for the rest of the genes/loci?

This is an important point and a question that relates to how we identify bona-fide MLLAF9 target genes. Since MLLAF9 overexpression results in widespread occupancy, many genes are not functionally dependent on MLLAF9 binding. Hence why we previously coupled degrons with nascent transcriptomics to identify the direct bona-fide MLLAF9 targets, with many of these being in the top 100 MLLAF9 bound genes [15]. We have thus changed the main text to specifically describe how we selected the subset of MLLAF9 targets for analysis in Figure 4. We focused on the top 100 MLLAF9-bound genes which contain the genes that are directly regulated by MLLAF9 (based on our previous work). We then determined the subset of those that are downregulated by both Menin and DOT1L inhibition ($p < 0.05$, $LFC < -0.5$, $n = 38$ genes). The 62 genes that are not downregulated by Menin and DOT1L inhibition is partly due to the cutoffs used but also due to many of them either not being functionally dependent on MLLAF9 or not specifically downregulated by both Menin and DOT1L inhibition in our model.

The increase in repressive marks seems very modest, even in the selected subset of genes. Could the authors provide statistics for Figure 4e?

The increase is indeed quite modest, and we thank the reviewer for the suggestion to include statistics for the box plots in Figure 4E. The analysis in Figure 4E was intended to largely demonstrate that specifically H2AK119ub is increased after treatment with both Menin and DOT1L inhibitors at MLLAF9 targets. We have now added statistics to show that this is the case. Conversely, H3K27me3 is only increased at a highly restricted subset of genes and thus would not be expected to show a clear increase at the 38 MLLAF9 target genes, especially at the timepoint used (48hrs). Similarly, KDM2B also shows a relatively modest increase at MLLAF9 targets compared to the set of control genes.

It seems that for the genes selected for Fig 4g, the changes observed are in line with the authors description, yet this is much less clear for the rest of genes, as plotted in Fig 4f. Can the authors provide statistical analysis for Fig 1f, or at least describe how those selected genes in Fig 4g were chosen.

As mentioned and discussed above, the direct MLLAF9 target genes have previously been described by us and others. Figure 4F is an average of the selected key MLLAF9 target genes (n = 38) that are also rescued in PCGF1 KO (n =11), of which not all are direct Polycomb targets. Therefore, the observed average increase in H2AK119ub is perhaps not as dramatic as expected. The profile plot is also depicting very few MLLAF9 targets and so it appears jagged and spiky. In addition, we attempted to present the data as a boxplot quantifying the levels at the TSS, but the changes were not statistically significant. However, it's worth noting that the nature and pattern of this chromatin modification is such that even a clear and large increase in this histone mark would appear modest when presented as an average profile plot and that 11 genes does not yield substantial statistical power. Finally, the chosen examples, *Meis1* and *Jmjd1c* are well described canonical MLLAF9 targets previously identified by us and others. We have made changes to the manuscript text to clarify this point [15].

Figure 5

The section starts with the suggestion that MLL-FPs may prevent Polycomb repression. To study this, the authors focused on all downregulated genes (n=176) after both SGC and VTP treatments. For consistency the authors should focus on the set of genes analyzed in figure 4, or at least justify the change.

This is a good point raised by the reviewer and something that we should've made clearer in the results section. As we sought to investigate how MLL-FPs may be preventing Polycomb repression, we identified three potential mechanistic explanations; transcription, MLL eviction or H3K79methylation. In figure 4 we focused specifically on MLL-FP target genes that are the likely drivers of leukemogenesis to determine if canonical MLLAF9 targets are decorated with repressive modifications following Menin and/or DOT1L inhibition. However, to understand the mechanistic basis for Polycomb antagonism by MLLAF9 we leveraged the differences between the two inhibitors and the fact that MLL1 and MLL-FP eviction occurs more broadly. Therefore, in Figure 5 we switch from investigating the effects on distinct MLLAF9 target

genes to the broader transcriptional changes induced by DOT1L and Menin inhibition. By selecting similarly downregulated genes shared by both treatments, we were able to perform a comparative analysis to reveal the likely mechanism/s driving changes in Polycomb activity. Even though Menin inhibition resulted in more potent eviction of MLLAF9/MLL1, the reduction in H3K79me2 was substantially less than with the DOT1L inhibitor and thus a significantly lower induction of H2AK119ub was observed (**Fig. 5b-e**).

By performing the analysis of genome analysis using 'bins', they observed differences in the dynamics of H2K79me3 mark. The authors suggest that this is because the enzymatic activity is abolished globally. Later, there are additional results on the dynamics of H3K79me2 loss. Are the regions depicted in Fig 5g losing the active mark after 48 hours? In others words, the authors should use the same loci in pane 5g and panel 5j.

To clarify, different loci were represented in 5g and 5j to demonstrate the differences in dynamics that we observed with DOT1L and Menin inhibitor treatment. *Six1* (shown in 5g) and *Meis1* and *JMJD1c* (shown in 5j) are all genes that show a loss in H3K79me2 with both inhibitors (Figure 5g left panel). *BRD4* and the 3'-end region of *Elf1* show a loss of H3K79me2 only after DOT1L inhibition. Therefore, the gene loci used in Figure 5i-j are similar to the regions used in Figure 5g that correspond with a loss of H3K79me2 with both inhibitors.

The intention of Figure 5g was to compare the effects of the two inhibitors on the relationship between H3K79me2 and H2AK119ub. Figure 5i, on the other hand, depicts the temporal dynamics and kinetics of H3K79me2 loss and H2AK119ub/H3K27me3 gain in response to Menin inhibition.

To alleviate the reviewer's concern, we have added IGV snapshots of *Meis1* and *Jmjd1c* in the same form displayed in Figure 5g and *Six1* as depicted in Figure 5j as a Reviewer only figure (**Reviewer Fig. 4**).

What is depicted in Fig 5h? Transcription? Of which subset of genes? It is not clear neither in the figure nor the caption.

Apologies for this. Figure 5h does show transcription, we have labelled this in the figure and changed the figure legend.

For the loci that are still decorated with H3K79me2 after Menin eviction treatment, the authors provide in the Discussion section a different explanation with respect to the data interpretation described in the results section of Figure 5. They claim this may be possible through the activity of the native DOT1L. If this is the case, even at longer time points, those loci will not be devoid of H3K79me3. The authors should clearly explain by which mechanism, under their point of view, Menin inhibitors behave differently than the DOT1L inhibitor. Are both mechanism (delay and native DOT1L) possible at the same time?

Thank you for highlighting this, we have updated the text to more accurately reflect our view on the mechanistic basis for these observations. There are indeed likely to be two potential mechanisms that are taking place. One mechanism is related to the transcriptional state of the

gene such that upon gene downregulation, H3K79me2 levels will reduce to correspond with the new level of expression. However, the extent to which this complex relationship between transcriptional output and H3K79me is true in every context remains unclear. The second mechanism is one that we recently published [15] and that we included in the discussion. It shows that, in MLL-FP leukaemia, there are two functionally distinct DOT1L complexes. The native DOT1L complex deposits H3K79me at all genes while the MLL-FP-DOT1L complex deposits H3K79me only at MLL-FP target genes. Since Menin inhibition potentially evicts MLL-FP, it will result in a dramatic loss of H3K79me2 only at genes that are predominantly dependent on MLL-FP for H3K79me2. The native DOT1L complex mechanism potentially explains how Menin inhibition can potentially evict MLL without a corresponding decrease in H3K79me2 due to the low contribution of MLL-FP towards the total H3K79me2 at a particular gene. It's important to note that the delay in H3K79me2 loss (due to passive dilution) still occurs whether you inhibit the native DOT1L or MLL-FP DOT1L complex.

In the results section we explain that H2AK119ub induction is only seen with Menin inhibition at genes where we also see loss of H3K79me2. In other words, at genes where we see potent loss of MLL1/MLL-FP but no corresponding loss of H3K79me2, there is no increase in H2AK119ub levels. Nevertheless, Menin inhibitor can induce transcriptional downregulation of distinct genes in other cell types, which would consequently lead to reduced H3K79me2 over time and thus the eventual induction of H2AK119ub.

Figure 6

ChIPseq data presented in panels 6a and 6d should be quantified (using boxplots, and including the corresponding statistical analysis) to appreciate the impact.

Thank you, we have added the necessary statistical analysis and quantified the data using boxplots.

It looks that H3K27me3 is increased for the Top 1000 MLLAF genes upon Menin inhibition (VTP). How is this possible when the increase of H2AK119ub is supposed to be more limited, as reported by the authors in the previous section? And, why switching from Top100 to Top1000 genes?

We agree with the reviewer that this plot appears to show a small increase in H3K27me3 after treatment with VTP50469. However, there are a few reasons why this apparent increase is not meaningful. Firstly, the increase is only observed in a short window (~2-3kb) upstream of the TSS and does not spread into the gene body. This increase does not correlate with any other specific chromatin change associated with Menin inhibition. We have now reanalysed the data and separated it into clusters based on the levels of H3K79me2 (high, medium and low). The modest increase in H3K27me3 in the PCGF1 KO cells was only observed in the cluster with little to no H3K79me2 (**Extended Data Fig. 7a-b**).

We apologise for the confusion with the profile plot analysis. The analysis of the top 1000 genes was intended to demonstrate the changes in repressive modifications across a broad set

of genes. We have now replaced it with the analysis mentioned above for both H2AK119ub and H3K27me3 ChIP-seq (**Fig. 6a, Extended Data Fig. 7a-b**).

Why is the increase in H3K27me3 observed only upon VTP treatment (as depicted in panel 6d), but not with SGC treatment?

The reviewer is correct in pointing out that H3K27me3 is noticeably increased after Menin inhibition but not DOT1L inhibition. The main reason for this is that at the timepoint analysed (48hrs for both Menin and DOT1L inhibitor), Menin inhibition more potently downregulates transcription which is necessary to induce H3K27me3. Conversely, DOT1L inhibition is much slower as it is dependent on passive dilution and therefore cellular division to remove H3K79me2 and downregulate transcription. Importantly, in both cases, the induction of H2AK119ub is required for the deposition of H3K27me3.

To confirm that longer treatment with the DOT1L inhibitor will facilitate greater induction of H3K27me3, we performed ChIP-seq in MLLAF9 cells treated with either DMSO, DOT1L or Menin inhibitor for 3 days and performed ChIP-seq for H3K27me3. As expected, Menin inhibitor potently induced H3K27me3 and in agreement with our hypothesis, DOT1L inhibitor also efficiently induced H3K27me3 at critical MLL-FP target genes at this later time point (**Fig. 6c**).

In panel 6k, the authors included a set of data featuring OCIAML3 cells, which is not listed in the Materials and Methods section. Furthermore, there is no specific commentary on these cells in the Results section. Additionally, aside from the title, there is no further mention of NPM1 in the discussion. If experiments related to NPM1 are included, they should be properly explained and discussed to ensure coherence with the rest of the manuscript.

Apologies for the error in not including the OCI-AML3 cell line in the materials and methods section, we have now corrected this. The experiments with the OCI-AML3 cells were mentioned in the results section of the original manuscript in line 411.

Our intention with the OCI-AML3 cells was to determine whether short term Menin inhibition could induce sustained repression (a similar phenotype to our washout experiments) in the context of AML with wild-type MLL1. We have now amended the discussion to mention these experiments more clearly.

Finally, regarding the discussion, the authors should mention previous works (Van der Boom 2016, Cell Rep; Maat 2021, iScience; Schmidt 2022, Blood Cancer Discovery) that study the relation between MLLAF9 and PRC1.1 in the context of leukemia.

We thank the reviewer for this comment, which was also pointed out by reviewer 2. We have now included these references in the results/discussion of the revised manuscript.

References:

1. Blackledge, N.P., et al., *Variant PRC1 complex-dependent H2A ubiquitylation drives PRC2 recruitment and polycomb domain formation*. *Cell*, 2014. **157**(6): p. 1445-1459.
2. Buchwald, G., et al., *Structure and E3-ligase activity of the Ring-Ring complex of polycomb proteins Bmi1 and RING1B*. *Embo j*, 2006. **25**(11): p. 2465-74.
3. Moussa, H.F., et al., *Canonical PRC1 controls sequence-independent propagation of Polycomb-mediated gene silencing*. *Nature Communications*, 2019. **10**(1): p. 1931.
4. McGinty, R.K., R.C. Henrici, and S. Tan, *Crystal structure of the PRC1 ubiquitylation module bound to the nucleosome*. *Nature*, 2014. **514**(7524): p. 591-6.
5. Soto-Feliciano, Y.M., et al., *Molecular switch between mammalian MLL complexes dictates response to Menin-MLL inhibition*. *bioRxiv*, 2021: p. 2021.10.22.465184.
6. Riising, E.M., et al., *Gene silencing triggers polycomb repressive complex 2 recruitment to CpG islands genome wide*. *Mol Cell*, 2014. **55**(3): p. 347-60.
7. Cooper, S., et al., *Jarid2 binds mono-ubiquitylated H2A lysine 119 to mediate crosstalk between Polycomb complexes PRC1 and PRC2*. *Nature Communications*, 2016. **7**: p. 1-8.
8. Fursova, N.A., et al., *Synergy between Variant PRC1 Complexes Defines Polycomb-Mediated Gene Repression*. *Mol Cell*, 2019. **74**(5): p. 1020-1036.e8.
9. Neff, T., et al., *Polycomb repressive complex 2 is required for MLL-AF9 leukemia*. *Proc Natl Acad Sci U S A*, 2012. **109**(13): p. 5028-33.
10. Spangler, C.J., et al., *Structural basis of paralog-specific KDM2A/B nucleosome recognition*. *Nat Chem Biol*, 2023. **19**(5): p. 624-632.
11. Zhu, S., et al., *RIPK3 deficiency blocks R-2-hydroxyglutarate-induced necroptosis in IDH-mutated AML cells*. *Sci Adv*, 2024. **10**(16): p. eadi1782.
12. Weiss, R.J., et al., *Genome-wide screens uncover KDM2B as a modifier of protein binding to heparan sulfate*. *Nat Chem Biol*, 2021. **17**(6): p. 684-692.
13. Jackson, A.L., et al., *Widespread siRNA "off-target" transcript silencing mediated by seed region sequence complementarity*. *Rna*, 2006. **12**(7): p. 1179-87.
14. Olsen, S.N., et al., *MLL::AF9 degradation induces rapid changes in transcriptional elongation and subsequent loss of an active chromatin landscape*. *Mol Cell*, 2022. **82**(6): p. 1140-1155.e11.
15. Gilan, O., et al., *CRISPR-ChIP reveals selective regulation of H3K79me2 by Menin in MLL leukemia*. *Nat Struct Mol Biol*, 2023.

Response to reviewers

Reviewer #1 (Remarks to the Author):

The authors provided thorough responses to each of my concerns. They offered thoughtful feedback, made meaningful revisions, and conducted additional experiments to support their claims further and address my critiques. I believe the manuscript is suitable for publication however, it could be even stronger if the authors consider the following comments:

Thank you to the reviewer for their positive comments and for supporting the publication of our manuscript. The reviewer has been very constructive throughout the review process, and we appreciate their additional comments to further strengthen the manuscript.

1. When the authors address choosing PRC1.1 over RING1B, they claim that KO of PCGF1 prevented a global increase in H2AK119Ub levels (Fig. 6a-c, Extended Data Figure 6a). While their claim is valid, I noticed that in medium and high H3K79me levels, there is a significant increase of H2AK119Ub even in PCGF1 KO (in the drug treatment conditions). While not as large of a change as wt, it is still a change. This might be important when considering that H3K79me regions might be the critical regions for PCGF1 in this disease model. I wondered if other polycomb complexes might contribute to this increase of H2AK119Ub at H3K79me regions.

We attributed the persistence of increased H2AK119ub observed in the PCGF1 KO cells to a reduction in PRC1.1 activity. This notion is supported by our ChIP-seq experiments in the human (MOLM13) leukaemia cells with combined KO of PCGF1/BCOR, which demonstrated both the reduced baseline levels of H2AK119ub prior to treatment and the near complete suppression of H2AK119ub induction following treatment (**Extended Data Figure 6a**). However, we cannot completely rule out the role of other PRC1 complexes and have therefore added a sentence to highlight this possibility in the discussion (line 623).

2. When the authors compare BCOR and KDM2B at MLLAF9 target genes, it is clear there is some overlap. But it isn't exactly clear how much overlap. This makes it difficult because the reader must compare a main figure to an extended figure at the same time. I suggest having the ChIP Seq tracks together so one can easily compare. I would also add that seeing the global levels of BCOR vs. KDM2B would be nice. Even if they may have other functions that explain why they may not have a perfect overlap, it would still be interesting to see how much of it is overlapping in their model systems.

We thank the reviewer for this comment and agree that it would be interesting to compare BCOR and KDM2B global occupancy and have thus generated global heatmaps of KDM2B, BCOR and H2AK119ub occupancy (**Extended Data Figure 4e**). We have also performed a correlation analysis of BCOR and KDM2B levels, which revealed a close relationship between the two factors, as expected. Despite this, some genes had disproportionately higher levels of BCOR or KDM2B (**Reviewer Figure 1**), we are happy to include this as part of an Extended Data Figure if the reviewer thinks it would be helpful. We have now also combined the genome browser tracks together to make comparison easier.

Reviewer Figure 1: ChIP-seq correlation of BCOR and KDM2B levels at the TSS +/-5kb of all genes .

3. Finally, I suggest that the authors tone down the paragraph below, where they propose mechanisms without any further biochemical and structural analyses, weakening their overall story. There is also a misstatement as the PCGF1 complex containing additional subunits is more active than the catalytic E3 ligase core (please see Rose et al., PMID: 27705745). PRC1 complexes containing PCGF4 and PCGF1 contain different subunits essential for localization, nucleosome-binding, and enzymatic activity. Therefore, the analyses that the authors propose are too speculative.

‘The conservation of this functional interplay across different cell types suggests that a fundamental mechanism underpins this antagonism. To identify a potential explanation for how H3K79me2 could influence PRC1 activity downstream of recruitment, we sought to explore published structural data. Given that the PCGF proteins are the only non-enzymatic subunits of PRC1 that are critical for in vitro PRC1 activity [46], we reasoned that PCGF1 may influence PRC1.1 activity through specific interactions with the nucleosome. In support of this, available structural data of RING1B/PCGF4 complex bound to a nucleosome has shown that the PCGF4 K62 and R64 residues form salt bridges with H4E74 and H3D77, respectively, with D77 being two residues away from H3K79 [47] (Extended Data Fig. 9a). Importantly, alignment of PCGF1 and PCGF4 as well as structural modelling and AlphaFold3 analysis, suggests that these same salt bridges cannot be formed in the context of PCGF1, where the positively charged PCGF4 K62 is predicted to turn into the negatively charged PCGF1 E91 and the positively charged PCGF4 R64 is predicted to turn into the polar PCGF1 Q93 (Extended Data Fig. 9ac). Therefore, we reasoned that the PCGF1 E91 residue could in principle form a salt bridge with H3K79, which may be altered due to structural rearrangement by the presence of di trimethylation.’

Apologies for the misstatement regarding the PCGF subunits, the reviewer is correct in saying that other subunits have been shown to be required for the in vitro activity of PRC1. We have now amended that statement.

Our intention with the structural modelling analysis was to simply propose a *potential* biochemical explanation for the relationship between H3K79me and H2AK119ub. However, we agree that this can only definitively be proven using structural and biochemical experiments and so we have now added a sentence to moderate the paragraph and reflect the speculative nature of these results (line 586).

Reviewer #2 (Remarks to the Author):

The authors have performed an extensive revision of the manuscript in response to the comments and concerns we raised, and they have pointed to the possibility that PCGF1/RING1B as part of the PRC1.1 complex interacts differently with nucleosomes than PCGF4/RING1B, thereby explaining its lack of interaction with H3K79me2.

The main question for us was (and is) whether the data in the manuscript supports the conclusions. Moreover, we also questioned the conceptual novelty of the findings reported.

The authors main conclusions are: We demonstrate that inhibition of transcription is not sufficient to induce Polycomb-mediated repressive histone modifications, especially at active genes. We also provide evidence that H3K79me2 mostly influences the activity of PRC1/PRC1.1, rather than its recruitment to chromatin.

Our comments:

1. The authors extend the many previous reported results showing an anti-correlation between H3K79me2 and H3K27me3 to H2AK119ub1. These published studies (e.g. PMID 33060580, 21741597) have not reported on an anti-correlation between H3K79me2 and H3K27me3 on a global scale but have focused on the genes that are required for development or leukemia, balancing activation (H3K79me2) and repression (H3K27me3). Although the demonstration that PRC1.1 is involved in setting up H2AK119ub1 and H3K27me3 has not been reported before, we do not consider this as a major conceptual novel finding, because of the known close link between PRC1 and PRC2 and therefore H2K119ub1 and H3K27me3 mediated transcriptional repression. The novelty of the manuscript is presumably that H3K79me2 directly inhibits the catalytic activity of PRC1.1, which, if demonstrated mechanistically (see below), is novel – however, perhaps better suited for another journal.

Thank you to the reviewer for appreciating the substantial work that we put into our revisions and recognising the novel aspect of the manuscript. We have addressed their additional comments below.

Just a brief comment about the anti-correlation. As the reviewer correctly recognises, the published studies mentioned do not report on a global anti-correlation. This is a crucial distinction with our work, where we have for the first time demonstrated a global anti-correlation and functional link between H3K79me2 and **PRC1-deposited** H2AK119ub (**Figure 5D**). Importantly, our findings do not extend to PRC2-mediated H3K27me3, as induction of this modification is governed by other mechanisms, which are not the focus of our manuscript.

Instead, the manuscript reveals a novel antagonistic relationship between DOT1L and PRC1, with DOT1L functioning to establish transcriptional memory to protect genes from premature polycomb repression.

2. The demonstration that inhibition is not sufficient to induce Polycomb-mediated repressive histone modifications at active genes is interesting, however, we are not convinced that the results presented by the authors fully support this statement. Previously published results have demonstrated that Polycomb group proteins are recruited following transcriptional termination,

and that they are required for maintaining transcriptional repression. The authors acknowledge this, and the results presented in Fig. 2i are also consistent with this notion. However, the authors concluded that transcription does not explain the observed increase in PRC1 activity. They pointed out that, unlike the Menin inhibitor, the DOT1L inhibitor not only down-regulated genes but also showed a marked difference in H2AK119ub1 deposition, indicating that the depletion of H3K79me2 affects PRC1 activity rather than transcriptional changes. Additionally, this effect did not result in a global increase in H3K27me3.

If we understand the reviewer correctly, the crux of the confusion is that they are trying to reconcile the reported link between transcription and Polycomb with our findings that transcriptional changes do not explain the increase in PRC1 activity.

Firstly, it is important to establish an important point that may have been overlooked. The transcription of genes can be reduced through the loss of an activator (e.g. MLL-FP eviction) without any immediate changes in the activity of Polycomb complexes. Indeed, this is what we observed (**Figure 5h**). At an early timepoint, transcriptional downregulation of MLLAF9 target genes is similar in both control and PRC1 depleted cells. These results suggest that transcription can be reduced independently of PRC1 activity.

We agree with the reviewer that in a physiological non perturbed context, transcriptional changes may eventually lead to increased PRC1 activity (e.g. during differentiation), however, our data indicates that this is likely due to the gradual dilution or loss of H3K79me2 that occurs at downregulated genes as the cells divide. This is in fact the central message of the manuscript, that DOT1L mediated memory, via the long-lived H3K79me2, prevents premature PRC1-mediated repression following transcriptional downregulation.

The point of confusion may have arisen due to not adequately explaining the link between transcription and H3K79me2. In mammalian cells, H3K79me2 is established co-transcriptionally and ubiquitously marks active genes [1]. Hence, H3K79me2 is often a consequence of transcriptional activation, and its global loss does not always impact transcription. However, transcriptional downregulation will eventually result in the dilution of H3K79me2 through cell division (due to the absence of a demethylase). Importantly, our results clearly demonstrate that loss of transcription is not sufficient to drive PRC1 activity **until** H3K79me2 is diluted through cell division. Our experiments bypass the normal processes where H3K79me2 is gradually lost due to a reduction in transcription during development. By dramatically abolishing H3K79me2 without a concomitant change in transcription, we were able to specifically isolate H3K79me2, rather than the process of transcription, as the causative factor preventing PRC1-mediated repression (**Figure 7j**).

The reviewer's interpretation that termination of transcription **alone** is sufficient for increased Polycomb activity, would require much of our data to be ignored. For example, our results clearly demonstrate that Menin inhibitor induced transcriptional changes do not lead to a corresponding increase in H2AK119ub at all downregulated genes (**Figure 5b, g**), since H3K79me2 remains at most of these genes until later timepoints when it is diluted through cell division (**Figure 5b, g-j**). Furthermore, our results using the DOT1L inhibitor across 5 different cell types demonstrates that loss of H3K79me2 is sufficient to cause a widespread increase in H2AK119ub (PRC1 activity) in the absence of widespread transcriptional changes (**Figure 5d, 7a-c, j, Extended Data Figure 6a, 8f**).

The authors present several experiments to support this claim. The authors observed absence of a global increase in H3K27me3 could be attributed to the relative short time points used and the fact that PRC2 will only work onto the genes it is recruited. Several published results have shown that H3K27me3 is established with slow kinetics in vitro and after prolonged PRC2 binding in vivo (see e.g. PMID 21078963; 19541851; 24561620). The authors analyzed published data and conducted their own experiments using CHIP-seq for SUZ12, POL2, H3K27me3, and H2AK119ub1 in mESCs, as well as H3K27me3 and H2AK119ub1 for K562 cells treated with DMSO or DRB for 8 hours. However, the data presented in Reviewer Figures 2 and 3 do not convincingly support the claim that transcription inhibition alone is insufficient to induce PRC1/2 activity, as these experiments were conducted with very short time points.

Whilst we agree that the link between transcription and PRC2 probably exists, it remains largely peripheral to the central findings of our study. Our manuscript details the antagonistic relationship between DOT1L and PRC1.1, not PRC2. Nonetheless, we systematically addressed it in the revised manuscript largely because the reviewer had focussed on this relationship.

It is important to reiterate that in our leukaemia cell data, the activity of PRC2 occurs at a highly restricted set of genes (e.g. Meis1, JMJD1c, Six1) following treatment with a Menin or DOT1L inhibitor. The level of PRC2 activity appears to be related to the degree of loss of H3K79me2 AND transcription. Conversely, PRC1 activity increases globally at TSSs upon loss of H3K79me2 with the degree of increase in H2AK119ub correlating with the baseline levels of H3K79me2, independent of transcriptional changes.

Whilst it is true that PRC2 kinetics are slow, the 8hr time point we had used is similar to the timepoints previously published by the Helin Lab [2]. Importantly, the kinetics of PRC1 activity are relatively fast and thus are unlikely to explain the absence of increased H2A119ub levels following DRB treatment (**Figure 7i**) [3]. We agree that it would be interesting to look at the effects of longer treatments, however, this is not possible as these transcriptional inhibitors are profoundly cytotoxic at later timepoints.

The binding of SUZ12 or other PRC complexes would precede the establishment of H3K27me3 but requires a longer incubation time for the spreading of this mark due to the gradual recruitment of PRC2, its necessity for specific nucleation sites, positive feedback for spreading, and the coordinated chromatin changes that occur progressively over multiple cell cycles. In the analyses the authors also need to specifically analyze the increase of H3K27me3 at sites where SUZ12 is recruited in response to the treatment versus sites where SUZ12 is not bound.

We agree with the reviewer that the full establishment of H3K27me3 is a multistep process that spans multiple cell divisions. This fact is consistent with our finding that H3K79me2 functions as transcriptional memory that contributes to the delay in Polycomb activity. We would again like to reiterate that our manuscript does not attempt to address the purported link between transcription and PRC2, but instead is focussed on the novel antagonism between H3K79me2 and PRC1.

Nevertheless, as requested by the reviewer, we have analysed the top 500 loci where SUZ12 is increased following DRB treatment (**Reviewer Figure 2a**). This analysis reveals that there is a small subset of genes where H3K27me3 is increased following DRB treatment, as reported

in the Helin paper, even after 6h. Importantly, these findings do not extend to H2AK119ub, where no changes were seen at the same loci after DRB treatment (**Reviewer Figure 2a**).

Critically, when we analysed the chromatin state of these genes, we found that they harboured low levels of H3K79me2 and RNA Pol II at baseline (**Reviewer Figure 2b-d**). These results suggest that the subset of genes most responsive to DRB treatment in terms of SUZ12 recruitment are marked by low levels of H3K79me2 and RNA Pol II. This is entirely consistent with our data and further confirms that transcriptional inhibition is not sufficient to induce PRC2 activity at highly active genes. We are happy to include some of this data in the manuscript if the reviewer thinks that it would strengthen our findings.

Reviewer Figure 2: a) analysis of the top 500 genes with increased SUZ12 binding after DRB treatment at 12h. Corresponding changes in H3K27me3 and H2AK119ub are shown. b) Baseline levels of H3K79me2 (RPM) are shown for genes with ‘high’, ‘mid’ and ‘low’ H3K79me2 levels and compared with the levels of H3K79me2 for SUZ12-induced or H3K27me3 induced genes following DRB treatment. The SUZ12-induced genes are the same 500 from (a) and the H3K27me3-induced genes are the subset of the 500 SUZ12 genes with a LogFC > 0.3 (n=241) c) Baseline levels of RNA Pol 2 (RPM) are shown for genes with ‘high’, ‘mid’ and ‘low’ RNA Pol 2 levels and compared with the levels of RNA Pol 2 for SUZ12-induced or H3K27me3-induced genes following DRB treatment (same gene subsets as in b). d) Genome browser snapshots of two genes with high (KLF9) or low (ZFP3612) baseline H3K79me2 levels.

3. The authors have provided some interesting experiments addressing the mechanism by which H3K79me2 potentially inhibits the activity of PRC1.1. By structural modeling based on data from the RING1B/PCGF4 complex they suggest that E91 of PCGF1 should form a salt bridge with H3K79, which may be altered due to structural rearrangement by the presence of trimethylation (interestingly the authors do not measure H3K79me3 at any point in the manuscript). Therefore, they suggest that RING1B/PCGF1 activity is inhibited by H3K79me3.

To understand whether this is the case they generate a PCGF1 E91K mutant. If we follow the rationale of the authors such mutant should not be affected by H3K79me3 and lead to the faster H2AK119ub1 of DOT1L inhibited genes to which PRC1.1 is bound. However, instead of testing this, the authors show that WT PCGF1 expression restores sensitivity to DOT1L and Menin inhibition, while the mutant restores sensitive a bit less (Fig. 7o). In the Extended figures 9f, g the authors show minor and non-significant effects between wt and E91K PCGF1 on MEIS1 expression, and the association of H2AK119ub1 and H3K27me3 with two DOT1L regulated genes. In other words, the experiments are not conclusive, perhaps partly because steady-state levels of H2AK119ub1 are measured.

We understand the reviewer's concern and we agree that we did not adequately explain the structural modelling. Our intention was to harness structural data to model the interaction between PCGF proteins and nucleosomes, especially in relation to H3K79. This analysis demonstrated that PCGF4 forms salt bridges with H3D77 and H4E74, which are in the vicinity of H3K79 (**Extended Data Figure 9a**). Since these salt bridges cannot form in the context of PCGF1 (**Extended data Figure 9b**), our modelling suggests that instead, a potential direct interaction could occur between PCGF1 E91 and H3K79 (in its unmethylated form). Therefore, we hypothesized that H3K79me2/3 could disrupt the interaction of PCGF1 with the nucleosome necessary for activity. As correctly stated by the reviewer, in such a model, the mutant PCGF1 E91K could be predicted to modify nucleosomes either if H3K79 is methylated or not, given the resemblance of the mutant with PCGF4 (**Extended data Figure 9c**). However, there is also the formal possibility that the mutant may fail to interact properly with unmethylated H3K79, resulting in a mutant that is less active. Our rescue experiments clearly showed that unlike WT PCGF1, the mutant PCGF1 was unable to completely rescue the phenotype of PCGF1 KO cells, suggesting that this residue may be required for interaction with H3K79. Moreover, the changes in Meis1 expression after treatment between the WT and mutant PCGF1 were statistically significant after the addition of another independent replicate (n=4) (**Extended Data Figure 9f**).

These data suggest that the PCGF1 E91 residue is required for the activity of PCGF1 following loss of H3K79me2, consistent with the proposed biochemical mechanism. However, deciphering the precise mechanism at the atomic level would require extensive biochemical and structural experiments which are beyond the scope of this study.

To provide potentially more conclusive experiments the authors need to perform time-resolved experiments to test how mutant and wt PCGF1 differentially lead to H2AK119ub1 association with target genes in response to DOT1L inhibition.

While our results indicate that amino acid E91 of PCGF1 at the vicinity of H3K79 is required for its function, a thorough structure-function investigation would be required to unambiguously decipher the exact biochemical mechanism. We agree with the reviewer that kinetic experiments after DOT1L inhibition may result in larger differences between the mutant and wildtype. However, given that the potent loss of H3K79me2 is required to maximise the induction of H2AK119ub and the reduced activity of the mutant is captured in our single time point experiments, we cannot think of a reason why it would change the trend. Without changing the trend between the mutant and wildtype, kinetic experiments will only incrementally extend our analysis without further supporting the mechanism.

Moreover, thee need to perform biochemical experiments to test PCGF1wt/RING1B and PCGF1mut/RING1B recombinant complexes for their activity on wt and H3K79me2 nucleosomes. Such data will directly test the authors hypothesis and potentially lead to experimental data supporting it.

As mentioned above, we agree with the reviewer that biochemical experiments are required to elucidate the exact mechanism of PRC1.1 association with H3K79-methylated nucleosomes. However, these experiments are very complex, time consuming and are not guaranteed to reflect the mechanisms that occur in an endogenous context. Critically, it is possible that precise assessment of the mechanism, **including the design of ideal mutations to test it**, would require structural determination of PRC1.1 in a complex with H3K79-methylated nucleosomes. This is far beyond the scope of this study.

However, we do agree with the reviewer that our structural modelling and functional validation are not definitive. Our intention was to simply propose a potential biochemical explanation, and we have toned down the paragraph describing these results, as was also requested by Reviewer 1 (point 3). We have now moderated the statements in line 586 of the results section.

4. The authors describe that a major difference between DOT1L inhibition and Menin inhibition is that Menin inhibition prevents MLL-FP binding, whereas DOT1L does not. However, in Fig. 4a, b they show that both compounds lead to reduction of MLL-AF9 binding. Moreover, all the significant downregulated genes show similar levels of downregulation using the two inhibitors (Figs. 4g, 5a, 5g, Extended Figs 4a, b, d, 5a). Why is MLL-AF9 lost upon DOT1L inhibition and what does this observation mean for the interpretation of the results? The authors should discuss this.

The major difference between DOT1L and Menin inhibition is that DOT1L inhibition results in the global loss of H3K79me, whilst Menin inhibitors disrupt the interaction between Menin and MLL1/2 and MLL-FP. The consequences of Menin inhibition are that MLL-FP is evicted along with DOT1L (as DOT1L is recruited by MLL-FP), which leads to loss of H3K79me2 at genes with high levels of MLL-FP. DOT1L inhibition also leads to eviction of MLL-FP as previously shown by us and others [4, 5]. The proposed mechanism for this is that Menin contains a reader domain required for recognising and binding H3K79me2 [6]. Therefore, the similar displacement of MLLAF9 seen with both inhibitors suggests that MLL-FP eviction is not the major factor contributing to the differences in H2AK119ub induction (**Figure 5b, g**).

Additional comments:

There are still issues in data presentation and interpretation, particularly the following points:

- Figure 1h: Why do the DKO cells grow faster than the single KO and the wt cells?

The double knockout cells (BCOR and PCGF1) and to a lesser extent the single knockout cells, grow faster than the wildtype cells. The likely reason for this is that disruption of PRC1.1 leads to modest de-repression of several MLL-FP target genes in the human leukaemia cell lines (MOLM-13 and MV4;11) (**Figure 3b**). We have included additional analysis to highlight the effects of BCOR and/or PCGF1 loss on gene expression prior to treatment (**Extended Data Figure 2e**).

- Extended Data Figure 1d: The RING1B sgRNA VTP and KDM2B sgRNA VTP do not show a significant increase in growth. Are these data statistically significant?

We have now added additional independent replicates to the competition assays. Our updated analysis shows that while there is a statistically significant increase in RING1B sgRNA+ cells, the KDM2B sgRNA+ cells did not reach a statistically significant increase.

- Extended Data Figure 1i: The different graphs are difficult to separate, and interpretation is therefore difficult.

Thank you for bringing this to our attention, it has now been corrected.

- Extended Data Figure 1m: The authors claim that “depletion of PRC1.1 did not affect the degree of H3K79me2 loss after DOT1L inhibition, indicating that resistance is not due to changes in DOT1L activity.” However, the knockout of PRC1.1 (BCOR) leads to a greater loss of H3K79me2 at baseline levels compared to the parental line. Therefore, depletion of PRC1.1 may partially involve changes in DOT1L activity.

It does appear by western blot in the BCOR KO cells that baseline levels of H3K79me2 are lower than in the control cells. While we cannot explain these results, given that this was not the case in the PCGF1 KO cells, we believe our conclusion that PRC1.1 depletion did not affect the degree of H3K79me2 loss after DOT1L inhibition still stands. We also overlapped the ChIP-seq in the MLL-AF9 cells, which similarly shows largely comparable levels of H3K79me2 in the control and PCGF1 KO cells (**Extended Data Figure 7d**).

- Figures 2e and Extended Data Figure 2g: It appears that only MEIS1 is rescued by PCGF1 knockout. For genes such as *Hoxa9*, *Jmjd1c* (Figure 2e), and *Runx2*, *MEF2C*, there does not seem to be significant rescue, as log2FC values are similar in control and PCGF1 knockout cells. Thus, PCGF1 knockout does not appear to rescue many genes.

The effects of PCGF1 and/or BCOR KO are to effectively blunt the Menin/DOT1L inhibitor-induced transcriptional downregulation of genes rather than completely prevent their downregulation. The box plots in **Figure 2d** illustrate the coordinated rescue of downregulated genes following treatment with both inhibitors across the three cell lines. However, the impact on specific genes is variable between cell lines and is also dependent on the time point at which the RNA-seq was performed. Using degrons coupled with nascent transcriptomics, we have previously shown that there are only approximately 20-50 bona-fide direct MLLAF9 target genes [4, 5]. Whilst the canonical MLL-FP target gene, *Meis1*, is reproducibly rescued by PCGF1 or BCOR KO in both the human leukaemias and primary murine leukaemia cells following both treatments, other well described and critical MLL-FP target genes are also clearly rescued in either the human or murine cells. For example, in the murine MLLAF9 cells, *Six1*, *Tsc22d2* and *Jmjd1c* were rescued in both drugs, while *Hoxa9* and *Arid1b* were more specific for the DOT1L inhibitor (**Figure 2g**). Similarly, in the human leukaemia cell lines, *MEF2C* and *JMJD1C* were also consistently rescued.

- Extended Data Figure 4d: The panels have not been labelled for which histone modifications have been tested.

Apologies for this, it has now been amended.

- Figure 4g and Extended Data Figure 4e: In Figure 4g, *JMJD1C* and *MEIS1* genes in MLL-

AF9 cells treated with either VTP/SGC or both showed a consistent increase in H2AK119ub1 and H3K27me3. In contrast, Extended Data Figure 4e shows no signal of H3K27me3 for the JMJD1C gene in control MOLM13 cells treated with VTP/SGC, despite an increase in H2AK119ub1. Only the MEIS1 gene in control MOLM13 cells treated with VTP (not SGC) showed an increase in both H2AK119ub1 and H3K27me3.

Here, the reviewer is highlighting differences in chromatin changes between the murine and human leukaemia cell lines. Whilst there are differences in the MLL-FP target genes between the two cell lines, the main reason for the variable induction of H3K27me3 is related to the time points used and the degree of gene downregulation. For example, in the murine MLLAF9 cells, *Meis1*, *Six1* and *Jmjd1c* are potentially downregulated and show a clear increase in H3K27me3 after 3 days of treatment. This is also related to the short doubling time (12h) of the primary murine MLLAF9 cells compared to the longer doubling time (24h) of MOLM13 cells. At the time point when ChIP-seq was performed, DOT1L inhibitor treatment had not sufficiently reduced H3K79me2 to cause enough gene downregulation to induce H3K27me3. This is clearly illustrated by the differences in H3K79me2 reduction between Menin and DOT1L inhibition. Menin inhibition more potently reduced H3K79me2 at *Meis1*, which is why it led to a greater induction in H2AK119ub and H3K27me3 (**Extended Data Figure 4f**).

Why did SGC treatment not show an increase in these marks for this key gene? Such discrepancies challenge the authors' conclusion that “the induction of H2AK119ub was also accompanied by an increase in H3K27me3 deposition spread over the same regions of MLLAF9 target genes,” and then “we also observed similar induction of repressive histone modifications at key MLL-AF9 target genes in human (MOLM13) leukaemia cells”.

Our focus is primarily on the induction of H2AK119ub, which was clearly seen with both Menin and DOT1L inhibition. We have amended the sentence to more clearly focus on H2AK119ub changes and emphasize the restricted effects on H3K27me3.

Figure 5g: For completion the author should show H3K27me3 at these loci.

For presentation reasons, we have included this in Extended Data Figure.

- Figure 6b: The order of the panels has been switched between the H2AK119ub1(DMSO, VTP, SGC) and H3K27me3 (DMSO< SGC, VTP). This is slightly confusing

Thank you for drawing our attention to this, we have now corrected it.

- Extended Data Figure 7a and b: The heatmaps show very weak signals of all the post translational modifications and it is not possible to assess the effects of SGC and VTP treatments. Are these differences statistically significant at all?

The changes in histone modifications are much better illustrated by the corresponding profile plots in **Figure 6a**, where we see widespread induction of H2AK119ub with maximal induction observed at genes with high levels of H3K79me2. On the other hand, H3K27me3 was much more selective with no changes observed even in the high H3K79me2 cluster. Furthermore, our intention with **Extended Data Figure 7a** was to show the levels of H3K79me2 at baseline in the high, medium and low H3K79me2 clusters rather than to illustrate the changes in H2AK119ub after treatment.

Reviewer #3 (Remarks to the Author):

We appreciate the effort the authors have made to address all our concerns regarding the original manuscript. While most of our specific points have been addressed, we still have some remaining concerns.

Thank you to the reviewer for their constructive and helpful comments throughout the review process. The responses to their remaining comments are described below.

Additionally, we would like to emphasize (once again) that any further efforts to enhance this manuscript's readability are highly appreciated, as it contains a significant amount of data that is difficult to follow. For instance, why is the genome-wide screen conducted in human cells, while the epigenetic screen is done in murine cells? Is there a specific scientific rationale behind this choice, or was it simply exploratory at this stage? We also fail to understand why MOLM13 cells are used for validations.

Apologies for the lack of clarity in some parts, we have made substantial efforts to further improve the readability. To address the examples given by the reviewer, the genome-wide CRISPR screen was performed with the initial intention of broadly uncovering the regulators of DOT1L inhibitor efficacy. Since this analysis largely revealed genes with chromatin-based functions, we then performed follow up screens using an epigenetics-focussed library in murine MLLAF9 cells and included Menin inhibition. The murine MLLAF9 model is a powerful primary leukaemia model that we have characterised extensively at the transcriptional and chromatin level. Since there are both advantages and disadvantages to the human AML cell lines vs the primary murine leukaemia model, we performed the additional screens in the murine model to strengthen the relevance of the findings. We have restructured the paragraph and added a sentence to clarify the rationale for why the screens were performed in that order.

Whilst initial validation was performed in both MOLM13, MV4;11 and murine MLLAF9 cells, the primary reason for using the MOLM13 cells for most of the subsequent validation was largely due to the phenotype of PRC1.1 depletion in the MOLM13 cells being more pronounced than in the MV4;11 cells, especially in the cells with BCOR and PCGF1 double knockout. However, other advantages of the MOLM13 cells include the slower kinetics of their response to DOT1L and Menin inhibition, which enables the assessment of differentiation more easily. We have added a brief justification for why we used MOLM13 cells for follow-up experiments in line 159.

Regarding Cell Radar, if the knockout (KO) effects on transcription are so dramatic by themselves (such as a clear increase in genes related to, for example, ST-HSC), our previous comment remains relevant: why was this not considered in Figure 2? It is stated in the rebuttal and in several other sections that PCGF1 KO only blunts the transcriptional response to treatment, but this statement seems inaccurate, as it appears that PCGF1 KO by itself has significant effects, independent of the treatment. In any case, we suggest presenting the Cell Radar with the treatment alone, as well as with both PCGF1 KO and the treatment. While the KO may indeed blunt the transcriptional response to the treatment, there could be many other genes affected by the KO that are independent of the treatment. We believe this is an important set of data to include.

The reviewer makes an important point about the impact of PRC1.1 KO prior to drug treatment. When quantifying gene downregulation in response to treatment, the expression level is normalised to baseline levels prior to treatment. Therefore, the transcriptional response in the PCGF1 KO cells is blunted independently of any de-repression caused by PCGF1 KO at baseline.

We agree with the reviewer that we didn't adequately address the transcriptional effects of PRC1.1 depletion at baseline in Figure 2. We had performed an analysis of RNA-seq data from BCOR and PCGF1 KO cells showing a good correlation of both upregulated and downregulated genes, however we failed to explore the effects on MLL-FP target genes directly (**Extended Data Figure 2d**). To complement the cell radar data and de-repression of distinct MLL-AF9 target genes shown in **Figure 3a-b**, we are now including a more unbiased analysis of the baseline de-repression of genes that are also downregulated after treatment.

The analysis focussed on the top 100 genes that are downregulated by Menin or DOT1L inhibitor in MOLM13 cells, which showed no significant de-repression of these genes prior to treatment in either BCOR or PCGF1 KO cells. However, specific MLL-FP target genes (e.g. Meis1) showed modest de-repression (**Extended Data Figure 2e**).

As requested by the reviewer, we have also included the Cell radar plots (shown below) for the KO plus treatment vs treatment alone. We would be happy to include these in Extended data Figure 3 if the reviewer feels that it would add clarity.

Reviewer Figure 3: Cell radar plot of murine MLLAF9 (left) and human MOLM13 (right) cells for upregulated genes in the indicated samples.

Again, the fact that the KO alone induces changes independently of the treatment makes it difficult to fully grasp the relevance of the cytometry and transplant experiments. While the authors have provided explanations in the rebuttal, the readers may still find this confusing. The data in the new panel, Fig. 3i, may help clarify some points, but once again, showing the Cell Radar for KO + treatment, compared to KO alone, is important. A clearer distinction between the effects caused by KO alone and those resulting from its interaction with the treatment should be explored, or at the very least, clearly discussed in the manuscript.

The KO alone certainly induces some changes at baseline that are enriched for genes that belong to a more immature progenitor cell state. However, many of these changes are relatively modest and Cell Radar does not integrate the degree of transcriptional change. Therefore, the expected phenotypic changes are likely to be relatively subtle. Whilst the flow cytometry analysis indeed shows some changes in morphology, there was no clear change in the levels of the Ly6G marker in the PCGF1 KO cells at baseline (**Figure 3d**).

We have made additional changes in the manuscript to better explain the connection between transcriptional changes associated with PRC1 depletion at baseline and the phenotypes observed using flow cytometry and in vivo transplant experiments.

"Indicate the number of mice used in the study directly in the figure or in the main text."

"Thank you, we have now added this directly to the figure."

We are still not able to find the number of mice used in these experiments.

We sincerely apologise for this oversight; we have now confirmed that it is included.

The ChIP experiments shown in Fig 4f seem unnecessary if they are not showing statistical significance. Just to clarify: are *Mais1* and *Jmjd1c* part of the selected groups of genes (11), which are both MLLAF9 targets and rescued by PRC1.1? It is quite difficult to propose a general mechanism based on such a limited number of genes, especially one that occurs at the same loci. From panel 4e it seems clearer (although again for very few genes). The authors may be interested in the results of panel 4f because of the effect of double treatment on H3K27me3, but in general panel 4f seems to confuse the overall message. We found that these panels are not very well described in the results section, and the readers may be lost or may need further clarification when reaching this part of the manuscript.

We understand the reviewer's concern with this figure as was raised in the first submission. Our intention was to highlight the genes that undergo PRC2-mediated silencing, and we thus focused on genes that are MLLAF9 targets, respond to both treatments and are rescued by PCGF1 loss. Indeed, *Meis1* and *JMJD1c* are both part of the subset of 11 genes that we focused on. The reason for the small number of genes is primarily due to MLLAF9 directly regulating only a small set of genes that we previously identified using degrons coupled to nascent transcriptomics [4, 5].

We did not intend to propose a general mechanism from these analyses but only to demonstrate that Polycomb acts directly at MLLAF9 target genes, thereby providing an explanation for the phenotypic and transcriptional data. Whilst we think the profile plot has an important qualitative value, especially when seeing the pattern of increased H3K27me3 and H2AK119ub in the combination treatment, we accept the reviewer's point about the small number of genes and the inability to show statistical significance.

Therefore, we have added the names of the genes used in the profile plot to the figure legend. Alternatively, we can replace the profile plots with additional genome browser snapshots of genes that show induction of H3K27me3. Either way, we agree with the reviewer that these panels were not adequately explained, and we have now improved the description of these analyses in the results section.

We appreciate the 'Reviewer Fig. 4', as it provides consistency. We suggest including it in the manuscript.

Thank you, we have added it to the extended data figures. Genome browser snapshot of *Jmjd1c* and *Meis1* (**Extended Data Figure 5g**) genome browser snapshot of *Six1* VTP time course (**Extended Data Figure 6g**).

References:

1. Steger, D.J., et al., *DOT1L/KMT4 recruitment and H3K79 methylation are ubiquitously coupled with gene transcription in mammalian cells*. Mol Cell Biol, 2008. **28**(8): p. 2825-39.
2. Riising, E.M., et al., *Gene silencing triggers polycomb repressive complex 2 recruitment to CpG islands genome wide*. Mol Cell, 2014. **55**(3): p. 347-60.
3. Dobrinić, P., A.T. Szczurek, and R.J. Klose, *PRC1 drives Polycomb-mediated gene repression by controlling transcription initiation and burst frequency*. Nat Struct Mol Biol, 2021. **28**(10): p. 811-824.
4. Gilan, O., et al., *CRISPR-ChIP reveals selective regulation of H3K79me2 by Menin in MLL leukemia*. Nat Struct Mol Biol, 2023.
5. Olsen, S.N., et al., *MLL::AF9 degradation induces rapid changes in transcriptional elongation and subsequent loss of an active chromatin landscape*. Mol Cell, 2022. **82**(6): p. 1140-1155.e11.
6. Lin, J., et al., *Menin "reads" H3K79me2 mark in a nucleosomal context*. Science, 2023. **379**(6633): p. 717-723.

Reviewers' comments:

We thank the reviewers for dedicating their time to review our manuscript. We believe that their comments allowed us to improve the manuscript substantially. Changes to the manuscript have been highlighted in red and the changes to figures are listed below:

New data/Figures

- *In vitro* ubiquitination experiment **Fig. 7n-p**
- PRC1 complex purification and assembly, ubiquitination time course, and nucleosome reconstitution **Extended Data Fig. 9b-h**
- Replicates of the *in vitro* ubiquitination experiment **Raw_blot.pdf**

Minor Changes

- Wild type PCGF1 rescue experiments **Extended Data Fig. 1i, Extended Data Fig. 2i, Extended Data Fig. 7b**
- New CellRadar analysis **Fig. 3a, Extended Data Fig. 3a**
- Removed the AlphaFold structural modelling and associated mutational analysis

Reviewer #1 (Remarks to the Author):

While I still believe that the emphasis on structural modeling is somewhat excessive and the manuscript could benefit from further de-emphasizing this aspect, the authors present a very interesting and compelling story. I have no additional questions beyond those raised in my previous review concerning the structural modeling. Therefore, I see no reason to delay publication further.

We thank the reviewer for their constructive feedback and appreciate that they “see no reason to delay publication further”. We agree with the reviewer that the structural modelling was not sufficiently tempered given its speculative nature. Accordingly, we have removed the structural modelling and associated mutational analysis from the revised manuscript.

Instead, we now present compelling biochemical experiments demonstrating that H3K79me_{2/3} alone is sufficient to inhibit PRC1 activity *in vitro* (**Fig. 7n-p, Extended Data Fig. 9, Figure in response to Reviewer #2 below**). This direct antagonism of PRC1 activity supports our broader mechanism where H3K79 methylation protects active genes from Polycomb-mediated repression. It also presents an elegant explanation for why DOT1L is specifically hijacked by MLL-Fusion proteins in the development and maintenance of leukaemia: to enhance the epigenetic switch from repression (Polycomb) to activation (MLL) at leukaemogenic genes.

Reviewer #2 (Remarks to the Author):

As written in our latest review of this manuscript: ‘The novelty of the manuscript is presumably that H3K79me₂ directly inhibits the catalytic activity of PRC1.1, which, if demonstrated mechanistically (see below), is novel – however, perhaps better suited for another journal.’ In the revised version of the manuscript, the authors have chosen not to provide any mechanistically based experiments to show that H3K79me₂ inhibits the catalytic activity of PRC1.1 as requested in our overall comment and in detail in comment 3: ‘To provide potentially more conclusive experiments the authors need to perform time-resolved experiments to test how mutant and wt PCGF1 differentially lead to H2AK119ub1 association with target genes in response to DOT1L inhibition. Moreover, they need to perform

biochemical experiments to test PCGF1wt/RING1B and PCGF1mut/RING1B recombinant complexes for their activity on wt and H3K79me2 nucleosomes. Such data will directly test the authors hypothesis and potentially lead to experimental data supporting it.’

The authors have not experimentally addressed any of the requested experiments, and the main conclusion of the manuscript is therefore based on correlation and not mechanistic insights.

The authors claim that ‘the manuscript reveals a novel antagonistic relationship between DOT1L and PRC1, with DOT1L functioning to establish transcriptional memory to protect genes from premature polycomb repression’. They do not. They show that the previously known anticorrelation between DOT1L and PRC1, established on several genes can be extended to a global level, but they do not provide the antagonistic relationship based on a mechanistic understanding. Our recommendation of the suitability of the publication of the manuscript in Nature Cell Biology has therefore not changed.

We thank the reviewer for their thorough assessment and agree with their statement that it would constitute a novel finding if we had shown that “H3K79me2 directly inhibits the catalytic activity of PRC1.1”. We believe our new data (**Fig. 7n-p**) now provide direct evidence for this.

We would also like to apologise to the reviewer for not fully addressing this primary concern in our previous response. At the time, we believed our data provided strong support for a functional relationship between DOT1L and PRC1 and that we had rigorously excluded alternative explanations. However, we acknowledge that our manuscript did not offer evidence of a direct biochemical mechanism to explain how DOT1L antagonises PRC1.

While we accept that we had not demonstrated direct biochemical evidence to support the interplay between DOT1L and PRC1, we respectfully disagree with the reviewer’s assertion that there is a “known anti-correlation between DOT1L and PRC1 at several genes” and that our study merely extends this observation globally. A specific link between DOT1L and PRC1 has not been previously reported and a limited correlation or anti-correlation with PRC2 at a few loci does not constitute evidence of a broad functional relationship. Our manuscript included extensive **functional** experiments that went beyond simply demonstrating an anti-correlation between DOT1L and PRC1. These data highlighted a previously undescribed functional relationship between these two key chromatin proteins, as well as providing important insights into its molecular, cellular and therapeutic implications. Together with our new results (below), our study is thus the first to demonstrate a direct functional relationship between DOT1L and PRC1 activity.

The reviewer’s critical feedback proved highly constructive, motivating us over the past six months to generate definitive biochemical evidence for a direct functional link between DOT1L and PRC1. Using *in vitro* histone ubiquitination experiments, we now demonstrate that H3K79me2/3 alone is sufficient to antagonise PRC1 activity, as summarised below:

- We performed an *in vitro* ubiquitination assay using recombinant human PRC1.1 (PCGF1-RING1B) and PRC1.4 (PCGF4-RING1B) core complexes (**Fig. 7n**). The substrate consisted of either unmodified nucleosome core particles (H3K79me0) or nucleosome core particles that were reconstituted using octamers that contain an H3 di- or tri-methyl-lysine analogue (MLA) at position 79 (H3K79me2 MLA or H3K79me3 MLA, respectively; **Fig. 7n**).

- The rate of the ubiquitination of H3K79me2 MLA nucleosomes is about 2-fold lower than that of the unmodified nucleosomes for PRC1.1 and ~4-fold lower for PRC1.4 (**Fig. 7o-p**).

- H3K79me3 MLA nucleosomes show an even more pronounced decrease in the rate of ubiquitination compared to unmodified nucleosomes for PRC1.1 (~4-fold) and PRC1.4 (~10-fold) (**Fig. 7o-p**).

- These results conclusively indicate that H3K79me2/3 is sufficient to inhibit the core PRC1 complex. This is consistent with a model where the antagonism between DOT1L and PRC1 we observed across diverse cell types is direct and attributed to the catalytic function of DOT1L.

In light of these new findings — particularly our observation that both core PRC1.1 and PRC1.4 complexes are antagonised by H3K79me2/3 — we've withdrawn our prior suggestion of differential nucleosome engagement by PCGF1 and PCGF4. Accordingly, we have also removed the AlphaFold structural modelling and associated mutational analysis from the manuscript.

We hope the reviewer agrees that we have now conclusively addressed their remaining concern regarding the lack of evidence for direct regulation. With this key issue resolved, we believe our findings represent a significant advance for the field and would like to thank the reviewer for highlighting this important issue, which has ultimately strengthened the manuscript and enhanced its overall impact.

For the reviewer's convenience, we have also included this new data along with PRC1 complex purification and assembly, and the ubiquitination time course as separate figures below. This data, as well as the uncropped blots of the biological replicates can be found in **Fig. 7n-p**, **Extended Data Fig. 9** and source data **Raw_blots.pdf**.

The H3K79me2/3 mark is sufficient to inhibit H2AK119 ubiquitination of nucleosomes by the core PRC1 complex. (Top) A schematic representation of the *in vitro* histone ubiquitination assay. E1 and E2 represent UBA1 and UbcH5c, respectively. E3 represents the PRC1.1 or PRC1.4 protein complex heterodimers consisting of PCGF1-RING1B or PCGF4-RING1B, respectively. (Left) An *in vitro* ubiquitination assay comparing the activity of RING1B-PCGF1 (PRC1.1) or RING1B-PCGF4 (PRC1.4) on mononucleosomes that were reconstituted with an unmodified H3 histone or methyl-lysine analogue (MLA) H3 histones, as indicated. Ubiquitination was detected by immunoblotting using anti-H2AK119ub antibodies. H3 and MBP immunoblots were used as loading controls to detect the nucleosome substrate and MBP-tagged PCGF1/4 proteins, respectively. (Right) Ubiquitination activity results from (Left) are shown as a bar plot. The means represent the relative ubiquitination activities presented in the form of H2AK119ub densitometry values normalised to the average signal of the same blot. Error bars indicate the standard deviation over three independent replicates of ubiquitination reactions that were carried out on three different days and were then subjected to immunoblotting simultaneously. Values for each replicate are shown as dots.

PRC1 complex purification, assembly and time course. **a.** Size exclusion chromatography of RING1B-PCGF1 using Superdex 200 Increase 10/300 column and its fractions analysed using SDS-PAGE. PCGF1 is N-terminally tagged by MBP, and RING1B includes an N-terminal strep tag. **b.** Size exclusion chromatography of PCGF4-RING1B using HiLoad Superdex 200 16/600 columns, and its fractions analysed using SDS-PAGE. PCGF4 is N-terminally tagged by MBP, and RING1B includes an N-terminal strep tag. **c.** SDS-PAGE analysis of 5 μ g of histone octamers made using unmodified or methyl-Lysine Analog (MLA) histones H3.1 as indicated. WT represents a wild-type H3.1 histone. H3K79me0 represents a non-MLA histone, without a K79C mutation, but with the C96S and C110A mutations that are included in the MLA histones. **d.** The activity of PRC1.1 (RING1B-PCGF1) on 1000 nM of unmodified wild-type nucleosomes was assayed at two PRC1.1 enzyme concentrations, as indicated. The reaction was stopped at three time points, as indicated. No ATP control, and E1/E2-ub only control were also performed as indicated. The experiment was run in two independent replicates, and a representative blot is presented. **e.** As (**d**), except that RING1B-PCGF4 was assayed, with samples as indicated. **f.** (Top) HPLC trace of H3K79me2 MLA histones separated on a ZORBAX RRHD 300SB-C3 2.1x100mm 1.8 μ m column used for LC-MS. The horizontal line indicates the retention volume of the main product. An asterisk indicates a minor peak, corresponding to species with a nonspecific alkylation. (Bottom) Deconvoluted mass of species detected in the peak marked with a black horizontal line above. **g.** Same as (**f**), except that H3K79me3 MLA histone was analysed.

Reviewer #3 (Remarks to the Author):

Regarding my first point on clarifying the screening strategy, I now find the text and the author's narrative clear.

We thank the reviewer for their persistence on this point and agree that the manuscript has now improved due to these changes to readability.

The analysis of gene expression prior to treatment has also been included. While transcriptomic changes caused by the KO alone are significant enough to impact cell identity (as shown by cell radar analysis), the authors now show that the top genes affected by the treatment were not previously influenced by the KO. This finding is, in fact, more relevant than the cell radar analysis itself. When considering all conditions, only minimal differences are observed between treatment with and without the KO. Therefore, if cell radar analysis is presented, it should include all conditions, as shown in the rebuttal. Otherwise, it should be omitted entirely.

Thank you for the suggestions. We also agree that it is useful to have the context provided by the whole suite of conditions in CellRadar and so they have replaced **Fig. 3a** and the knockout conditions in MOLM13 cells have been included in **Extended Data Fig. 3a**.

Regarding the ChIP experiments in Fig. 4f, the authors should carefully decide how to present these results. As mentioned previously, profiles with such a small number of genes might be confusing for readers. However, the authors now provide a clearer explanation in the text about the relevance when presenting these results.

To further increase clarity, we have also added the number of genes directly to the figure.

Reviewers' comments:

Reviewer #2 (Remarks to the Author):

This is the fourth version of the manuscript under review for Nature Cell Biology. The fundamental novelty of the work has not changed. Although the authors have added new data in this revision, we remain of the opinion that the results are incremental in the context of the literature, and that the manuscript would be more appropriate for another journal.

In our previous reviews, we asked the authors to provide mechanistic insights of how H3K79me2 directly inhibits the catalytic activity of PRC1.1. Specifically, in the last review we asked the authors: ‘.. to perform time-resolved experiments to test how mutant and wt PCGF1 differentially lead to H2AK119ub1 association with target genes in response to DOT1L inhibition. Moreover, they need to perform biochemical experiments to test PCGF1wt/RING1B and PCGF1mut/RING1B recombinant complexes for their activity on wt and H3K79me2 nucleosomes. Such data will directly test the authors hypothesis and potentially lead to experimental data supporting it.’

In response to this request, the authors have removed the results related to the PCGF1 mutant and the final paragraph of the result section in the manuscript. Through additional experiments, they found that the previous results were not supported by the new data we had asked them to generate (fortunately, the previous results were not published). Instead, they now shown that di- and trimethylation of H3K79 inhibits the activity of recombinant PCGF1-RING1B and PCGF4-RING1B in vitro.

The main conceptual advance of the manuscript is that, in addition to MLL-FPs inhibiting PRC1.1 binding to their target genes, H3K79 methylation also inhibits PRC1.1 activity. In other words, H3K79 modification contributes to the inhibition H2AK119ub1 deposition.

Additional comments.

Upon re-reading the manuscript, we noted several statements in the introduction that appear biased and should be revised for clarity and neutrality.

Lines 99-100: “Thus, it not generally thought that the MLL-FP drives leukaemia initiation by antagonising Polycomb.”

MLL fusion proteins (MLL-FPs) inevitably antagonize Polycomb group proteins, as they drive transcription at Polycomb target genes. While alternative models have been proposed for how MLL-FPs function, their activity ultimately counteracts Polycomb-mediated repression. To our knowledge, no scientist has proposed that knockout of the Polycomb genes would have the same effect as the expression of MLL fusion proteins.

We thank the reviewer for highlighting this point. The statement has now been removed to enhance clarity.

Lines 111-115: ‘Although Menin and DOT1L inhibitors have shown specific anti-leukaemic effects, DOT1L inhibitors have been disappointing in the clinic thus far while Menin inhibitors are highly promising in early-phase clinical trials [25, 26]. While both potently downregulate

similar target genes, this discrepancy is likely related to the fact that these inhibitors target different aspects of MLL-FP function [27, 28].’

This may be true; however, what is certain is that the first generation of DOT1L inhibitors were highly unstable and displayed relatively poor PK-PD profiles, which limited their potency and effectiveness inhibiting DOT1L in vivo. The authors should discuss this possibility. Notably, Scott Armstrong, a co-author on the manuscript who was involved in of developing and testing of both classes of inhibitors will be aware of these limitations.

We agree with the reviewer regarding the poor pharmacokinetic properties of first-generation DOT1L inhibitors. We have now clarified in the manuscript that DOT1L inhibitors not only differ mechanistically from Menin inhibitors but also exhibit poor PK/PD characteristics, which likely contributed to their limited clinical success.

Line 83 “Although DOT1L inhibitors have suffered from poor PK/PD properties, the differences in clinical success between the two drugs likely also reflects their distinct mechanism of action [26, 28] [29, 30].”

Reviewer #3:

Remarks to the Author:

The authors have addressed most of my points and I am satisfied with the revised version of the manuscript, which has overall improved.

We thank the reviewer for their positive feedback and appreciate their support for the publication of this manuscript.